# On the Expressive Power of Permutation-Equivariant Weight-Space Networks

**Adir Dayan** [* 1] **Yam Eitan** [* 1] **Haggai Maron** [1 2]

## Abstract

Weight-space learning studies neural architectures that operate directly on the parameters of other neural networks. Motivated by the growing availability of pretrained models, recent work has demonstrated the effectiveness of weight-space networks across a wide range of tasks. SOTA weight-space networks rely on permutation-equivariant designs to improve generalization. However, this may negatively affect expressive power, warranting theoretical investigation. Importantly, unlike other structured domains, weight-space learning targets maps operating on both weight and function spaces, making expressivity analysis particularly subtle. While a few prior works provide partial expressivity results, a comprehensive characterization is still missing. In this work, we address this gap by developing a systematic theory for expressivity of weight-space networks. We first prove that all prominent permutation-equivariant networks are equivalent in expressive power. We then establish universality in both weight- and function-space settings under mild, natural assumptions on the input weights, and characterize the edge-case regimes where universality no longer holds. Guided by our theoretical results, we show that slight modifications to existing weight-space models yield a 34% improvement over prior SOTA, demonstrating the practical relevance of our framework.

## 1. Introduction

Weight-space learning studies neural architectures that operate directly on the parameters of other neural networks (Eilertsen et al., 2020; Unterthiner et al., 2020; Schürholt et al., 2021; Navon et al., 2023; Zhou et al., 2023a; Kofinas et al.,

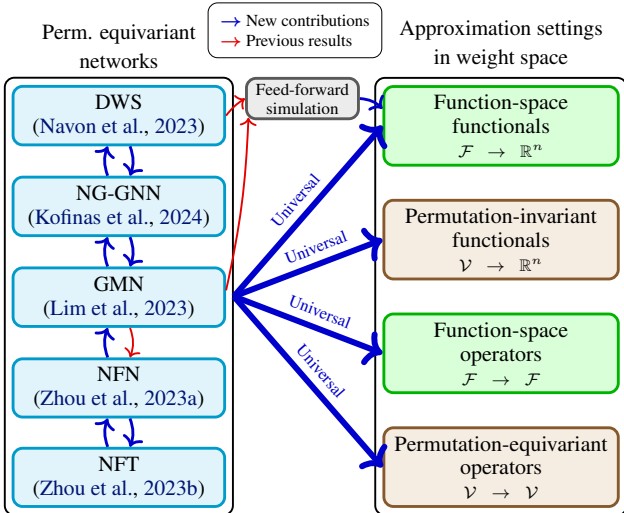

*Figure 1.* Expressivity landscape for permutation-equivariant weight-space networks on MLPs (blue arrows: new contributions; red arrows: previous results). $\mathcal{V}$ and $\mathcal{F}$ denote weight-space and function-space, respectively. **Left:** Equivalence of permutation-equivariant networks. **Center:** Assuming general position, all weight-space networks are universal across all approximation settings, strengthening prior feed-forward simulation results for DWS and GMN. **Right:** Approximation settings in weight space.

2024; Lim et al., 2023). Rather than treating trained models as black-box objects, weight-space methods view their parameters as structured data for downstream tasks such as accuracy prediction (Lim et al., 2023), meta-optimization (Zhou et al., 2023a; Kofinas et al., 2024), and editing implicit neural representations (Navon et al., 2023; Lim et al., 2023; Zhou et al., 2023a). A key property of MLP weight spaces is their inherent symmetry: permuting neurons within a hidden layer changes the weights but leaves the function realized by the weights unchanged. This has motivated the development of permutation-equivariant weight-space architectures (Navon et al., 2023; Zhou et al., 2023b;a; Lim et al., 2023; Kofinas et al., 2024), which represent the current state of the art for learning over MLP weights.

Symmetry-preserving architectures restrict the hypothesis space and may therefore impose fundamental limits on approximation power. Expressivity analysis is therefore critical in this setting: it provides principled guidance on when existing architectures suffice for a given task and when more expressive designs are required. Indeed, in other structured domains studied in Geometric Deep Learning

*Equal contribution [1]Technion – Israel Institute of Technology [2]NVIDIA Research. Correspondence to: Adir Dayan <adir.dayan@campus.technion.ac.il>, Yam Eitan <yam.eitan@campus.technion.ac.il>.

*Proceedings of the 43$^{rd}$ International Conference on Machine Learning*, Seoul, South Korea. PMLR 306, 2026. Copyright 2026 by the author(s).

(GDL) (Bronstein et al., 2021), including graphs, sets, and general data with symmetries (Zaheer et al., 2017; Morris et al., 2023; Maron et al., 2020; Dym & Maron, 2020; Ravanbakhsh, 2020; Keriven & Peyré, 2019; Azizian & Lelarge, 2020; Maron et al., 2019b), expressivity analyses have yielded both theoretical foundations and practical insights. Importantly, expressivity in weight-space learning is more nuanced than in many other GDL settings. Beyond approximating arbitrary symmetry-preserving maps on weight space, one may also ask whether networks can approximate *function-space* maps, i.e., maps whose outputs depend only on the function realized by the input weights rather than on a particular parameterization. Unfortunately, theoretical understanding in both settings remains limited, with only a few initial results (Navon et al., 2023; Lim et al., 2023; Kalogeropoulos et al., 2024). Figure 1 summarizes the community's current understanding (red arrows) of the approximation capabilities of equivariant weight-space networks. Characterizing expressivity in weight-space learning is therefore a central open question.

**In this paper**, we address this question by developing a unified expressivity theory for permutation-equivariant weight-space networks[1]. Throughout, we focus on weight-space networks operating on MLP weights, as this setting is the most developed both theoretically and empirically. We first clarify the relations between existing architectures, showing that most prominent permutation-equivariant networks (Navon et al., 2023; Zhou et al., 2023a; Lim et al., 2023; Kofinas et al., 2024) are equivalent in expressive power, with Neural Functional Transformers (NFTs) (Zhou et al., 2023b) as the sole exception. We further show that, under a general-position assumption on the input weights (i.e., unique bias terms per layer), which almost surely holds in practice, NFTs match the expressive power of the other architectures.

Building on this architectural equivalence, we study expressivity in weight-space learning more broadly, aiming to cover all natural settings in which the approximation power of weight-space networks should be examined. To this end, we organize our analysis around four fundamental approximation settings. First, we consider *function-space functionals*, maps that assign an output vector to an input network and depend only on its realized function. Second, we study *permutation-invariant functionals*, weight-space maps whose output is constant under hidden-neuron permutations but may still depend on a particular parameterization and not necessarily on the realized function. Third, we examine *function-space operators*, i.e., maps $\Psi$ from the space of continuous functions to itself. We aim to approximate such operators via equivariant weight-to-weight transformations, mapping weights that realize $f$ to weights that realize

a function close to $\Psi(f)$. Finally, we study *permutation-equivariant operators*, namely weight-to-weight maps that respect hidden-neuron symmetries but are not determined solely by the realized function. Figure 2 lists real-world examples for each of these four settings.

We analyze the expressive power of permutation-equivariant weight-space networks in each of these settings, showing that (i) they are universal approximators for *function-space functionals*; (ii) they are not universal for *permutation-invariant functionals* and *permutation-equivariant operators* in full generality, but universality is achieved under a general-position assumption on the input weights; (iii) *function-space operators* cannot be universally approximated when the input weights are restricted to a fixed architecture, but universality holds once the input architecture is allowed to be sufficiently large. Taken together, this yields a comprehensive expressivity characterization across all four settings. Figure 1 summarizes our main theoretical contributions (blue arrows) alongside previous results (red arrows).

Guided by our theoretical analysis, we propose a simple modification applicable to any existing weight-space network that addresses the expressivity limitations in the function-space operator setting. Instead of predicting a single output network, the model predicts multiple networks and ensembles them, effectively enabling outputs with higher capacity than the input model. Evaluated on INR editing tasks, a standard benchmark for function-space operators in the weight-space learning literature, this simple theory-driven modification yields consistent improvements across architectures, achieving up to a 34% improvement over prior SOTA and demonstrating the practical relevance of our framework.

**Contributions.** In summary, this paper makes the following contributions:

1. **Expressive equivalence of architectures.** We prove that all prominent permutation-equivariant weight-space networks for MLPs are equally expressive.

2. **Approximation framework.** We identify four natural approximation settings: function-space functionals, permutation-invariant functionals, function-space operators, and permutation-equivariant operators.

3. **Universality characterization.** For each setting, we characterize when universality is achievable: we prove universality under natural *general-position* assumptions, and identify regimes in which universality does not hold.

4. **Empirical gains.** Guided by our theory, we propose simple modifications to existing weight-space models that achieve up to a 34% improvement over prior SOTA on a standard model editing task.

---

[1]We exclude scale-equivariant networks such as Kalogeropoulos et al. (2024); Tran et al. (2024), as they involve different symmetry groups and are thus beyond our scope.

**Function-space functionals** $\mathcal{F} \to \mathbb{R}^n$
1. Model accuracy prediction
2. INR classification

**Permutation-invariant functionals** $\mathcal{V} \to \mathbb{R}^n$
3. Weight $\ell_2$-norm prediction
4. Loss landscape curvature prediction

**Function-space operators** $\mathcal{F} \to \mathcal{F}$
5. Image and 3D scene model editing
6. Domain adaptation

**Permutation-equivariant operators** $\mathcal{V} \to \mathcal{V}$
7. Pruning mask prediction
8. Gradient prediction for meta-optimization

*Figure 2.* Real-world examples of target functions for all approximation settings in weight-space learning. $\mathcal{V}$ and $\mathcal{F}$ denote weight-space and function-space, respectively. 1. (Lim et al., 2023; Zhou et al., 2023a; Kofinas et al., 2024) 2. (Navon et al., 2023; Zhou et al., 2023a; Lim et al., 2022) 4. (Gelberg et al., 2026) 5. (Zhou et al., 2023b; Lim et al., 2022; Kofinas et al., 2024) 6. (Navon et al., 2023) 7. (Zhou et al., 2023a) 8. (Kofinas et al., 2024; Gelberg et al., 2026; Zhou et al., 2024)

## 2. Previous work

Existing theoretical results on the expressivity of permutation-equivariant weight-space networks remain partial and largely focus on establishing specific capabilities rather than general approximation guarantees. In particular, Navon et al. (2023) showed that Deep Weight Space (DWS) networks can simulate a forward pass of the MLP defined by the input weights, and derived initial expressivity results in the function-space functional setting under additional assumptions on the target map. Kalogeropoulos et al. (2024) showed that ScaleGMNs can simulate both a forward pass and backpropagation with respect to the input weights. Similarly, Lim et al. (2023) established that Graph Meta-Networks (GMNs) can simulate a forward pass, and further proved that GMNs can express other weight-space models, including Neural Functional Networks (NFNs) (Zhou et al., 2023a) and StatNN (Unterthiner et al., 2020). While these results provide important insight into the capabilities of weight-space networks, they do not yield a comprehensive expressivity characterization or general universality guarantees. See Appendix A for an extended discussion.

## 3. Preliminaries

**Notation.** We begin by introducing basic notation used throughout the paper. An MLP architecture $A$ with $L$ layers is specified by a pair $(\boldsymbol{d}, \sigma)$, where $\boldsymbol{d} = (d_0, \dots, d_L) \in \mathbb{N}^{L+1}$ denotes the width of each layer and $\sigma$ is an activation function. Given MLP parameters $\boldsymbol{v} = (\boldsymbol{W}_1, \boldsymbol{b}_1, \dots, \boldsymbol{W}_L, \boldsymbol{b}_L)$, we denote by $f_{\boldsymbol{v}} : \mathbb{R}^{d_0} \to \mathbb{R}^{d_L}$ the function realized by the network with parameters $\boldsymbol{v}$, that is:

$$f_{\boldsymbol{v}}(\boldsymbol{x}) = \boldsymbol{W}_L\, \sigma\Big(\cdots \sigma(\boldsymbol{W}_1 \boldsymbol{x} + \boldsymbol{b}_1) \cdots \Big) + \boldsymbol{b}_L.$$

For any metric spaces $X, Y$, we denote the space of continuous functions $f : X \to Y$ by $\mathcal{C}(X, Y)$. For an integer $n \in \mathbb{N}$, we denote $[n] = \{1, \dots, n\}$.

**Weight space of a fixed architecture.** Throughout the paper, we consider weight-space networks that take as input the parameters of a fixed MLP architecture. We thus begin by formally defining the corresponding weight space in which these parameters reside.

**Definition 3.1** (weight space). Given an architecture $A = (\boldsymbol{d}, \sigma)$ with $L$ layers, for $\ell = 1, \dots, L$ the parameters of the $\ell$-th layer consist of a weight matrix and a bias vector

$$\boldsymbol{W}_\ell \in \mathbb{R}^{d_\ell \times d_{\ell-1}}, \qquad \boldsymbol{b}_\ell \in \mathbb{R}^{d_\ell}.$$

We denote by $\mathcal{W}_\ell \cong \mathbb{R}^{d_\ell \times d_{\ell-1}}$ and $\mathcal{B}_\ell \cong \mathbb{R}^{d_\ell}$ the corresponding weight and bias parameter spaces of layer $\ell$. The weight space associated with the architecture $A$ is defined as the direct sum

$$\mathcal{V}_A \coloneqq \Big(\bigoplus_{\ell=1}^{L} \mathcal{W}_\ell\Big) \oplus \Big(\bigoplus_{\ell=1}^{L} \mathcal{B}_\ell\Big) \cong \bigoplus_{\ell=1}^{L} \big(\mathbb{R}^{d_\ell \times d_{\ell-1}} \oplus \mathbb{R}^{d_\ell}\big).$$

We further associate to $\mathcal{V}_A$ the *realization map* $\mathcal{R} : \mathcal{V}_A \to \mathcal{C}(\mathbb{R}^{d_0}, \mathbb{R}^{d_L})$, defined by $\mathcal{R}(\boldsymbol{v}) \coloneqq f_{\boldsymbol{v}}$, which gives a continuous mapping from weight space to the space of continuous functions (see Proposition C.2 in the Appendix).

**Weight-space symmetries.** For any MLP, permuting the neurons within any hidden layer alters the raw parameterization of the network while leaving the underlying function it represents invariant. The changes to the weights induced by neuron permutations are formalized as a group representation on weight space, which we define below.

**Definition 3.2** (weight-space representation). For a given architecture $A = (\boldsymbol{d}, \sigma)$ with $L$ layers, define the corresponding neuron permutation group as the direct product of permutation groups of the hidden layers $1, \dots, L-1$:

$$G_A \coloneqq S_{d_1} \times \cdots \times S_{d_{L-1}}.$$

For $g = (\tau_1, \dots, \tau_{L-1}) \in G_A$, define a representation $\rho(g) : \mathcal{V}_A \to \mathcal{V}_A$ by

$$\rho(g)(\boldsymbol{W}_1, \boldsymbol{b}_1, \dots, \boldsymbol{W}_L, \boldsymbol{b}_L) \coloneqq (\boldsymbol{W}_1', \boldsymbol{b}_1', \dots, \boldsymbol{W}_L', \boldsymbol{b}_L'),$$

$$\begin{aligned}
\boldsymbol{W}_1' &= \boldsymbol{P}_{\tau_1}^\top \boldsymbol{W}_1, & \boldsymbol{b}_1' &= \boldsymbol{P}_{\tau_1}^\top \boldsymbol{b}_1, \\
\boldsymbol{W}_\ell' &= \boldsymbol{P}_{\tau_\ell}^\top \boldsymbol{W}_\ell \boldsymbol{P}_{\tau_{\ell-1}}, & \boldsymbol{b}_\ell' &= \boldsymbol{P}_{\tau_\ell}^\top \boldsymbol{b}_\ell \quad \ell = 2, \dots, L-1, \\
\boldsymbol{W}_L' &= \boldsymbol{W}_L \boldsymbol{P}_{\tau_{L-1}}, & \boldsymbol{b}_L' &= \boldsymbol{b}_L.
\end{aligned}$$

Here, $\boldsymbol{P}_{\tau_\ell}$ is the permutation matrix corresponding to $\tau_\ell$. By construction, $f_{\rho(g)\boldsymbol{v}} = f_{\boldsymbol{v}}$ for all $g \in G_A$ and $\boldsymbol{v} \in \mathcal{V}_A$.

As noted above, equivariant weight-space networks are designed to respect the symmetries induced by the representation of $G_A$. We recall the standard notions of invariance and equivariance below.

**Definition 3.3** (invariance and equivariance). Let $H$ be a group and let $(\mathcal{U}, \rho)$ be a representation of $H$. A map $\Psi : \mathcal{U} \to \mathbb{R}^n$ is said to be $H$-*invariant* if, for every $h \in H$ and every $\boldsymbol{u} \in \mathcal{U}$, it holds that $\Psi(\rho(h)\,\boldsymbol{u}) = \Psi(\boldsymbol{u})$. Similarly, a map $\Psi : \mathcal{U} \to \mathcal{U}$ is said to be $H$-*equivariant* if, for every $h \in H$ and every $\boldsymbol{u} \in \mathcal{U}$, it holds that $\Psi(\rho(h)\,\boldsymbol{u}) = \rho(h)\,\Psi(\boldsymbol{u})$.

Throughout, we slightly abuse terminology and use "permutation-invariant" and "permutation-equivariant" to refer to $G_A$-invariant and $G_A$-equivariant, respectively. Note that all prominent symmetry-preserving weight-space models have both an invariant and an equivariant version.

**Exclusion set and general position (GP).** In many settings, permutation-equivariant architectures are not universal across the entire input space, but they do achieve universality when one restricts attention to inputs that lie outside a small *exclusion set*, which is typically described as a union of lower-dimensional linear subspaces (Maron et al., 2020; Finkelshtein et al., 2025; Gelberg et al., 2026). We follow the same strategy here.

**Definition 3.4** (bias exclusion set). For a given architecture $A$, we define the weight space bias exclusion set by

$$\mathcal{E}_A := \left\{ \boldsymbol{v} \in \mathcal{V}_A \ \middle| \ \begin{array}{l} \exists \ell \in [L-1],\ \exists\, 1 \le i < j \le d_\ell \\ \text{such that } (\boldsymbol{b}_\ell)_i = (\boldsymbol{b}_\ell)_j \end{array} \right\}.$$

$\boldsymbol{v} \notin \mathcal{E}_A$ indicates that within every hidden layer, all neuron biases are pairwise distinct. This assumption is natural: $\mathcal{E}_A$ contains only degenerate parameter configurations and has Lebesgue measure zero; therefore, it is unlikely to arise under random initialization or stochastic training dynamics.

We note that the exclusion set defined above is merely one convenient choice, and that our theory extends naturally to several alternative exclusion sets. Alternative definitions, additional discussion, and an empirical validation of the GP assumption on standard pretrained networks are provided in Appendix G.

Whenever $\boldsymbol{v} \notin \mathcal{E}_A$, we say that $\boldsymbol{v}$ is in *general position (GP)*. When $A$ is clear from context, we write $\mathcal{V}_A$, $G_A$, and $\mathcal{E}_A$ as $\mathcal{V}$, $G$, and $\mathcal{E}$, respectively.

## 4. Approximation framework

When analyzing the expressivity of weight-space networks, several natural approximation settings arise, differing both in the type of target maps and the notion of approximation. We identify four fundamental settings: (1) *function-space functionals*, which map functions over a compact domain to output vectors; (2) *permutation-invariant functionals*, which map input weights to invariant output vectors and may not depend solely on the underlying function; (3) *function-space operators*, which map functions to functions; (4) *permutation-equivariant operators*, which map weights to weights while respecting permutation symmetries. Below, we define the corresponding notion of approximation via weight-space maps for each setting.

**Definition 4.1** (approximation via weight-space maps). Fix an MLP architecture $A = (\boldsymbol{d}, \sigma)$ and compact sets $X \subseteq \mathbb{R}^{d_0}$ and $K \subseteq \mathcal{V}_A$. Given a family $\mathcal{N}$ of weight-space maps and a target map $\Psi$, we say that $\mathcal{N}$ approximates $\Psi$ on $K$ if for every $\epsilon > 0$ there exists $\Phi \in \mathcal{N}$ such that one of the following holds, depending on the domain and codomain of $\Psi$:

1. *Function-space functionals*
   $(\Psi : \mathcal{C}(X, \mathbb{R}^{d_L}) \to \mathbb{R}^n, \quad \Phi : K \to \mathbb{R}^n)$:

$$\sup_{\boldsymbol{v} \in K} \left\| \Psi(f_{\boldsymbol{v}}) - \Phi(\boldsymbol{v}) \right\|_2 < \epsilon. \tag{1}$$

2. *Permutation-invariant functionals*
   $(\Psi : \mathcal{V}_A \to \mathbb{R}^n, \quad \Phi : K \to \mathbb{R}^n)$:

$$\sup_{\boldsymbol{v} \in K} \left\| \Psi(\boldsymbol{v}) - \Phi(\boldsymbol{v}) \right\|_2 < \epsilon. \tag{2}$$

3. *Function-space operators*
   $(\Psi : \mathcal{C}(X, \mathbb{R}^{d_L}) \to \mathcal{C}(X, \mathbb{R}^{d_L}), \quad \Phi : K \to \mathcal{V}_A)$:

$$\sup_{\boldsymbol{v} \in K} \left\| \Psi(f_{\boldsymbol{v}}) - f_{\Phi(\boldsymbol{v})} \right\|_\infty < \epsilon. \tag{3}$$

4. *Permutation-equivariant operators*
   $(\Psi : \mathcal{V}_A \to \mathcal{V}_A, \quad \Phi : K \to \mathcal{V}_A)$:

$$\sup_{\boldsymbol{v} \in K} \left\| \Psi(\boldsymbol{v}) - \Phi(\boldsymbol{v}) \right\|_2 < \epsilon. \tag{4}$$

## 5. Expressive equivalence of weight-space networks

We begin by comparing the expressive power of the prominent permutation-equivariant weight-space networks introduced in the literature, namely Deep Weight Space (DWS) networks (Navon et al., 2023), Neural Functional Networks, including both the neuron-permutation and hidden-neuron permutation variants (NP-NFN and HNP-NFN) (Zhou et al., 2023a), Graph Meta-Networks (GMNs) (Lim et al., 2023), Neural Graph GNNs (NG-GNNs) (Kofinas et al., 2024), and Neural Functional Transformers (NFTs) (Zhou et al., 2023b). Our goal is to characterize and compare the classes of functions these networks can approximate, using the approximation framework introduced in Section 4. For completeness, formal definitions of all architectures are given in Appendix D.2. Accordingly, we associate with each network-class the set of weight-space maps it can approximate.

**Definition 5.1.** Let $K \subseteq \mathcal{V}$ be compact, and let $\pi \in \Pi = \{\text{DWS}, \text{NP-NFN}, \text{HNP-NFN}, \text{GMN}, \text{NG-GNN}, \text{NFT}\}$ denotes a class of permutation-equivariant weight-space networks. For an output set $Y \in \{\mathbb{R}^n, \mathcal{V}\}$, define

$$\mathcal{N}^{\pi}(K; Y) := \left\{ \Psi \in \mathcal{C}(\mathcal{V}, Y) \mid \begin{array}{c} \Psi \text{ can be approximated} \\ \text{on } K \text{ by } \pi\text{-networks} \end{array} \right\},$$

We further define

$$\mathcal{N}^{\pi}_{\text{inv}}(K) := \mathcal{N}^{\pi}(K; \mathbb{R}^n), \qquad \mathcal{N}^{\pi}_{\text{equi}}(K) := \mathcal{N}_{\pi}(K; \mathcal{V}),$$

as the sets of invariant maps and equivariant operators, respectively, that $\pi$-networks can approximate on $K$.

Interestingly, we find that all the previously mentioned networks, except for NFTs, have exactly the same expressive power, despite having different architectures.

**Theorem 5.2.** *Let $K \subseteq \mathcal{V}$ be a compact set. Then, for any $\pi, \pi' \in \Pi \setminus \{NFT\}$,*

$$\mathcal{N}^{\pi}_{\text{inv}}(K) = \mathcal{N}^{\pi'}_{\text{inv}}(K), \qquad \mathcal{N}^{\pi}_{\text{equi}}(K) = \mathcal{N}^{\pi'}_{\text{equi}}(K).$$

The proof of Theorem 5.2, presented in Appendix D, proceeds by explicitly approximating the base layers of one network using those of another, thereby establishing mutual approximation.

We next turn to the remaining case of NFTs. Due to their non-standard attention mechanisms, the expressive power of NFTs is not equivalent to any of the architectures discussed above in full generality. However, we show that equivalence does hold for GP input weights (see Definition 3.4).

**Proposition 5.3.** *Let $\pi \in \Pi \setminus \{NFT\}$. There exists a compact set $K \subset \mathcal{V}$ such that*

$$\mathcal{N}^{\text{NFT}}_{\text{inv}}(K) \neq \mathcal{N}^{\pi}_{\text{inv}}(K), \qquad \mathcal{N}^{\text{NFT}}_{\text{equi}}(K) \neq \mathcal{N}^{\pi}_{\text{equi}}(K).$$

*However, for every compact set $K \subset \mathcal{V} \setminus \mathcal{E}$,*

$$\mathcal{N}^{\text{NFT}}_{\text{inv}}(K) = \mathcal{N}^{\pi}_{\text{inv}}(K), \qquad \mathcal{N}^{\text{NFT}}_{\text{equi}}(K) = \mathcal{N}^{\pi}_{\text{equi}}(K).$$

The proof of Proposition 5.3 is given in Appendix D.5. Taken together, Theorem 5.2 and Proposition 5.3 establish that all prominent permutation-equivariant weight-space networks are expressively equivalent, with NFTs matching this class under a GP assumption. Accordingly, throughout the remainder of the paper, we analyze the expressive power of a *generic* permutation-equivariant weight-space network, without committing to a specific architectural instantiation. In particular, all results apply to DWS, GMNs, NFNs, and NG-GNNs, and results stated under GP apply to NFTs as well.

# 6. Expressive power of permutation-invariant weight-space networks

In this section, we investigate the expressive power of *invariant* weight-space networks $\Phi : \mathcal{V} \to \mathbb{R}^n$. In studying these networks, two natural approximation settings arise: *function-space functionals* and *permutation-invariant functionals*. These settings, along with their formal notions of approximation, are detailed in items 1 and 2 of Definition 4.1. All proofs for this section are provided in Appendix E.

## 6.1. Approximating function-space functionals

We begin by establishing universality with respect to function-space functionals.

**Theorem 6.1.** *For any compact $K \subseteq \mathcal{V}$, every continuous function-space functional $\Psi : \mathcal{C}(X, \mathbb{R}^{d_L}) \to \mathbb{R}^n$ can be approximated on $K$ by invariant weight-space networks.*

*Proof sketch.* By Theorem 5.2 we can use DWS as a representative of permutation-invariant weight-space networks. The first step builds on Navon et al. (2023), which shows that DWS networks can approximate the forward pass of the MLP function realized by given input weights, evaluated at any arbitrary point. We use this to establish the following separation property: if $\boldsymbol{v}, \boldsymbol{v}' \in \mathcal{V}_A$ satisfy $\Phi(\boldsymbol{v}) = \Phi(\boldsymbol{v}')$ for every DWS network $\Phi$, then they realize the same function, i.e., $f_{\boldsymbol{v}} = f_{\boldsymbol{v}'}$. Next, we invoke the following separation-to-approximation result proved in Pacini et al. (2025b): Let $\mathcal{N}$ be a family of invariant networks on a space $\mathcal{U}$, constructed using a composition of equivariant affine layers and interleaving pointwise nonlinearities. Then any continuous function $F : \mathcal{U} \to \mathbb{R}^n$ satisfying

$$\left( \forall M \in \mathcal{N}, \ M(\boldsymbol{u}) = M(\boldsymbol{u}') \right) \ \Rightarrow \ F(\boldsymbol{u}) = F(\boldsymbol{u}') \quad (5)$$

can be approximated uniformly on compact subsets of $\mathcal{U}$ by functions in $\mathcal{N}$ [2]. Applying this result to our setting, consider the induced map $\Psi^* : K \to \mathbb{R}^n$ defined by $\Psi^*(\boldsymbol{v}) = \Psi(f_{\boldsymbol{v}})$. By construction, $\Psi^*$ is constant on all weight configurations that realize the same function and therefore constant on any pair of weights that are indistinguishable by DWS networks. Additionally, since the realization map is continuous (Proposition C.2), $\Psi^*$ is continuous as well. It follows that $\Psi^*$ can be approximated arbitrarily well by DWS networks, which completes the proof. $\square$

## 6.2. Approximating permutation-invariant functionals

We next examine the expressive power of invariant weight-space networks with respect to permutation-invariant functionals. Since neuron permutations preserve the realized function, every function-space functional is inherently

---

[2]This result is reminiscent of Stone–Weierstrass-type arguments, where separation implies uniform approximation.

permutation-invariant; however, the converse does not hold. While function-space functionals represent a fundamental setting, many practically significant weight-space quantities cannot be expressed in this form. For instance, the $\ell_2$-norm of the weights is a natural statistic, commonly used as a regularizer to improve generalization, yet it is not a function-space functional: the same realized function may admit multiple parameterizations with different $\ell_2$-norms.

Similarly, quantities related to the curvature of the loss landscape (e.g., the determinant of the loss function's Hessian matrix) are used in several applications (e.g., uncertainty estimation (Immer et al., 2021; Daxberger et al., 2021) and influence functions (Grosse et al., 2023; Koh & Liang, 2017)) and depend not only on the realized function, but also on the local geometry of the surrounding parameter space. Importantly, these quantities are permutation-invariant (Gelberg et al., 2026): neuron permutations preserve the loss value and its local geometry, and therefore leave curvature-based quantities unchanged. Motivated by these considerations, we evaluate weight-space networks within this broader class of permutation-invariant functionals. Notably, in contrast to our results for function-space functionals, we show that invariant weight-space networks are *not* universal for this broader class in full generality.

**Proposition 6.2.** *There exists a compact set $K \subset \mathcal{V}$ and a permutation-invariant map $\Psi : K \to \mathbb{R}$ that cannot be approximated on $K$ by invariant weight-space networks.*

*Proof sketch.* By Theorem 5.2, it suffices to consider NG-GNN as a representative permutation-invariant weight-space network. We construct two binary weight configurations $\boldsymbol{v}, \boldsymbol{v}' \in \mathcal{V}$ (see Figure 3 for an illustration) such that (i) the second-layer weight matrices $\boldsymbol{W}_2$ and $\boldsymbol{W}_2'$ have different ranks, and (ii) the neural graphs induced by $\boldsymbol{v}$ and $\boldsymbol{v}'$ are indistinguishable by the Weisfeiler–Leman (WL) test (Morris et al., 2023). First, since matrix rank is invariant under neuron permutations, $\boldsymbol{v}$ and $\boldsymbol{v}'$ lie in distinct $G$-orbits. As $G$ is finite, there exists a continuous permutation-invariant map $\Psi : \mathcal{V} \to \mathbb{R}$ separating these orbits, e.g., with $\Psi(\boldsymbol{v}) = 0$ and $\Psi(\boldsymbol{v}') = 1$. Second, NG-GNN applies message passing to the induced neural graphs, and thus cannot distinguish the two inputs as they are WL-indistinguishable (Morris et al., 2019). Hence it cannot approximate $\Psi$ on any compact set containing both $\boldsymbol{v}$ and $\boldsymbol{v}'$. $\square$

While Proposition 6.2 establishes a limitation of invariant weight-space networks in full generality, we show next that universality can be achieved under a GP assumption on input weights.

**Theorem 6.3.** *Let $K \subset \mathcal{V} \setminus \mathcal{E}$ be compact. Then any permutation-invariant functional $\Psi : \mathcal{V} \to \mathbb{R}^n$ can be approximated on $K$ by permutation-invariant weight-space networks.*

**Computational Graphs for Proposition 6.2**

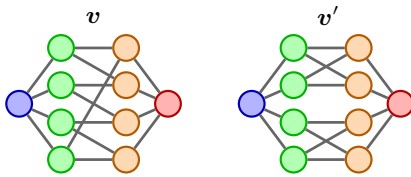

*Figure 3.* Computational graphs induced by weights $\boldsymbol{v}$ and $\boldsymbol{v}'$ used in the proof of Proposition 6.2. The weight matrices have binary entries, where 1 corresponds to an edge and 0 to the absence of an edge, and bias terms are equipped with features encoding their layer index. Both graphs admit identical 1-WL colorings (node colors) and are therefore indistinguishable by message-passing GNNs (NG-GNNs). Formal definitions of $\boldsymbol{v}$ and $\boldsymbol{v}'$ are given in Appendix E.1, Proposition E.8.

Analogous universality results for weight-space architectures operating on transformer parameters, rather than MLP weights, can be established through minor modifications of the proof of Theorem 6.3 (sketch below), extending the scope of our theory. Details are provided in Appendix H.

*Proof sketch.* Since $K \cap \mathcal{E} = \emptyset$, all bias vectors within each layer have pairwise distinct entries, which allows us to construct a continuous canonization map $\mathrm{canon} : K \to \mathcal{V}$ that maps each weight $\boldsymbol{v}$ to a canonical representative of its permutation orbit, and maps all orbit elements to the same output. The map $\mathrm{canon}$ is obtained by applying a permutation $g = (\tau_1, \ldots, \tau_{L-1})$, with $\tau_\ell \in S_{d_\ell}$, that orders neurons in each layer by sorting bias values; the distinct-bias assumption ensures that this ordering is unique and varies *continuously* with $\boldsymbol{v}$.

We then construct an approximation of the map $\boldsymbol{v} \mapsto \mathrm{Flat}(\mathrm{canon}(\boldsymbol{v}))$ using a DWS model, where $\mathrm{Flat}$ denotes a flattening of the weight space elements into a vector in $\mathbb{R}^M$, with $M := \sum_{\ell=1}^{L} d_\ell(1 + d_{\ell-1})$. To this end, note that for each layer $\ell$, the ranking map $\boldsymbol{b}_\ell \mapsto \mathrm{argsort}(\boldsymbol{b}_\ell)$ is permutation-equivariant with respect to neuron permutations in the $\ell$-th layer. Moreover, since $K \cap \mathcal{E} = \emptyset$, the induced ordering is locally constant and hence the ranking map is continuous on $K$. Because DWS layers subsume the DeepSets primitives (Zaheer et al., 2017), and DeepSets are universal for continuous permutation-equivariant maps (Segol & Lipman, 2019), a DWS network can approximate this ranking operation on $K$. Finally, combining the resulting ranks with pointwise MLP updates allows us to approximate $\mathrm{Flat}(\mathrm{canon}(\boldsymbol{v}))$. Appendix Lemma F.7 provides a formal construction for approximating the canonization map by DWS networks.

Permutation invariance then implies that $\Psi$ factors through the canonization map, i.e., there exists a continuous function $\tilde{f} : \mathbb{R}^M \to \mathbb{R}^n$ such that $\Psi(\boldsymbol{v}) = \tilde{f}(\mathrm{Flat}(\mathrm{canon}(\boldsymbol{v})))$ for all $\boldsymbol{v} \in K$. Since $\tilde{f}$ is continuous on a compact domain, it can be approximated by an MLP head composed with the DWS model mentioned above, yielding an approximation

of $\Psi$ on $K$ and completing the proof. □

**Discussion.** Notably, the construction in the proof above relies only on a restricted subset of the DWS operations presented in Navon et al. (2023). This suggests that universality can already be achieved using substantially fewer layer types than in the full DWS architecture, indicating that the network can be simplified without sacrificing expressive power. Full details are provided in Appendix F.1.

Additionally, Proposition 6.2 highlights a potential limitation in low-precision regimes. When weights or biases are heavily quantized (e.g., low-bit or binary networks), the probability of encountering degeneracies increases, making it more likely for inputs to fall inside the exclusion set $\mathcal{E}_A$. This phenomenon is empirically validated in Appendix G.1. In such cases, full universality for invariant weight functionals may fail, which suggests that applications involving extreme quantization may benefit from the development of more expressive architectural variants.

## 7. Expressive power of permutation-equivariant weight-space networks

In this section, we analyze the expressive power of *equivariant* weight-space networks $\Phi : \mathcal{V} \to \mathcal{V}$. In studying these networks, two natural approximation settings arise: *function-space operators* and *permutation-equivariant operators*. These settings, along with their formal notions of approximation, are detailed in items 3 and 4 of Definition 4.1.

### 7.1. Approximating function-space operators

In contrast to the invariant case, where function-space functionals form a subclass of permutation-invariant functionals, the equivariant setting exhibits a different phenomenon: some function-space operators cannot be approximated by any permutation-equivariant weight-to-weight map, even under GP assumption on the inputs. Intuitively, many natural operators take a function as input and increase its geometric complexity in the output, producing functions that require a richer representation class. For example, when representing a natural image or a 3D scene as a function using implicit neural representations (e.g., INRs (Sitzmann et al., 2020) or NeRFs (Mildenhall et al., 2021)), a natural transformation is a *zoom-out* operator: the original scene is preserved at a smaller scale, while new regions of the scene that are absent in the input are introduced. Another example arises in function-level domain adaptation (Navon et al., 2023), where one may wish to map a function corresponding to a global minimum of one loss to a global minimum of a different loss that incorporates additional data points not present during training. In both cases, the output function may exhibit greater complexity than the input. For a detailed discussion see Appendix F.2

Current equivariant weight-space networks are inherently limited in this regard, since they are constrained to output weights of the *same* architecture as their input. Because the representational capacity of MLPs with a fixed architecture is bounded (e.g., ReLU MLPs of fixed size has a bounded number of linear regions (Montúfar et al., 2014)), it is unsurprising that these networks cannot, in general, approximate transformations that increase geometric complexity.

**Proposition 7.1** (Informal). *For any fixed ReLU architecture $A = (\boldsymbol{d}, \mathrm{ReLU})$, there exists a family of natural continuous function-space operators $\mathcal{N} = \{\Psi : \mathcal{C}(X, \mathbb{R}^{d_L}) \to \mathcal{C}(X, \mathbb{R}^{d_L})\}$ that* cannot *be approximated by permutation-equivariant weight-space networks defined over $\mathcal{V}_A$.*

The proposition is formally stated and proved in Appendix F.2. We note that while the construction of $\mathcal{N}$ relies on the use of ReLU activations, we believe it could be adapted to other commonly used nonlinearities by generalizing the notion of linearity regions. Encouragingly, this expressivity limitation can be overcome by allowing the underlying architecture to be sufficiently large to accommodate the complexity of the target function-space operator.

**Theorem 7.2** (Informal). *Let $\Psi : \mathcal{C}(X, \mathbb{R}^m) \to \mathcal{C}(X, \mathbb{R}^m)$ be a continuous function-space operator where $X \subseteq \mathbb{R}^n$ is a compact set, and let $K \subseteq \mathcal{C}(X, \mathbb{R}^m)$ be a compact function set with respect to the supremum norm. Then, for any sufficiently large architecture $A$ and any compact set $K' \subset \mathcal{V}_A \setminus \mathcal{E}_A$ whose realized functions approximate those in $K$ to sufficient accuracy, the map $\Psi$ can be approximated on $K'$ by permutation-equivariant weight-space networks.*

The theorem is illustrated in Figure 4, and is formally stated and proved in Appendix F.2. Taken together, these results indicate that increasing architectural capacity of the input weights can substantially enhance expressivity, potentially unlocking new capabilities for weight-space networks. This may have practical significance for weight-space learning, as many prior model-editing studies (Navon et al., 2023; Zhou et al., 2023a; Kofinas et al., 2024) consider relatively small MLPs, often with only two hidden layers and modest hidden dimensions; see Section 8 for empirical validation. More broadly, an interesting direction for future work is to explore possible connections between the above findings and the well-known benefits of overparameterization in deep learning, where larger models are often easier to optimize (Du et al., 2019; Allen-Zhu et al., 2019) and can exhibit improved generalization (Belkin et al., 2019; Kaplan et al., 2020).

*Proof sketch.* The proof proceeds in three steps. First, we construct a continuous lifting of the function-space operator $\Psi$ to weight-space. That is, a continuous map $\Psi' : K' \to \mathcal{V}$ such that $f_{\Psi'(\boldsymbol{v})} \approx \Psi(f_{\boldsymbol{v}})$, for all $\boldsymbol{v} \in K'$.

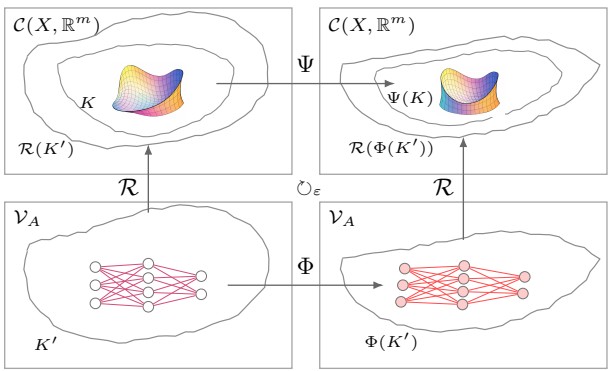

*Figure 4.* Diagram illustrating Theorem 7.2. A function-space operator $\Psi : K \subset \mathcal{C}(X, \mathbb{R}^m) \to \mathcal{C}(X, \mathbb{R}^m)$ is approximated by a permutation-equivariant weight-space operator $\Phi : K' \subset \mathcal{V}_A \setminus \mathcal{E}_A \to \mathcal{V}_A$. The set $K'$ serves as an approximation of $K$ under the realization map $\mathcal{R}$. The diagram is approximately commutative, showing that $\Phi$ approximates $\Psi$ via $\mathcal{R}$.

Since $K \subset \mathcal{C}(X, \mathbb{R}^m)$ is compact, so is $\Psi(K)$, and thus $\Psi(K)$ admits a finite open cover by $\epsilon$-balls centered at reference functions $f_{\boldsymbol{v}_1}, \ldots, f_{\boldsymbol{v}_M}$, provided the architecture $A$ is chosen sufficiently large. By construction, the functions realized by weights in $K'$ approximate those in $K$, implying that $\Psi(f_{\boldsymbol{v}})$ lies in this cover for all $\boldsymbol{v} \in K'$. Using a continuous partition of unity subordinate to this cover, we express each $\Psi(f_{\boldsymbol{v}})$ as a continuous convex combination of the reference functions. This combination can in turn be implemented by an architecture containing the corresponding MLP weights $\boldsymbol{v}_1, \ldots, \boldsymbol{v}_M$ as parallel sub-networks along with a final layer encoding the combination coefficients. This yields a continuous map $\boldsymbol{v} \mapsto \Psi'(\boldsymbol{v})$. Second, as $\Psi'$ is not permutation-equivariant, we augment it using a continuous canonization map that selects a unique representative from each permutation orbit, obtaining a continuous permutation-equivariant operator $\tilde{\Psi} : K' \to \mathcal{V}$ such that $f_{\tilde{\Psi}(\boldsymbol{v})} \approx \Psi(f_{\boldsymbol{v}})$. Finally, since $K' \subset \mathcal{V} \setminus \mathcal{E}$, the universality result established in the next subsection guarantees that permutation-equivariant weight-space networks can approximate $\tilde{\Psi}$ arbitrarily well on $K'$, completing the proof. $\square$

## 7.2. Approximating permutation-equivariant operators

As in the invariant setting, many practically relevant weight-space transformations cannot be expressed as function-space operators. For example, pruning methods typically depend on parameter-level quantities such as weight magnitude (e.g., threshold-based pruning (Han et al., 2015)) or local properties of the loss landscape (e.g., (LeCun et al., 1989; Hassibi et al., 1993)). Although such transformations are permutation-equivariant, they are not determined solely by the function realized by the weights. Similar considerations apply to tasks such as adapted gradient prediction (Zhou et al., 2024; Kofinas et al., 2024) and winning-ticket mask prediction (Zhou et al., 2023a), which may likewise depend on weight-space geometry rather than only on the realized

function. Motivated by these considerations, we also study the expressive power of equivariant weight-space networks in relation to permutation-equivariant operators.

The analysis of the expressive power of weight-space networks with respect to permutation-equivariant operators closely parallels the invariant case. We begin by showing that equivariant weight-space networks are *not* universal for this class across the entire weight space.

**Proposition 7.3.** *There exists a compact set $K \subset \mathcal{V}$ and a permutation-equivariant operator $\Psi : K \to \mathcal{V}$ that cannot be approximated on $K$ by permutation-equivariant[3] weight-space networks.*

Proposition 7.3 follows directly from Proposition 6.2, since any permutation-invariant weight-space map can be converted into a permutation-equivariant one via broadcasting (see Appendix F.2 for a complete proof). As in the invariant case, while Proposition 7.3 shows that equivariant weight-space networks are not universal in full generality, we show next that universality is achieved for GP inputs.

**Theorem 7.4.** *Let $K \subset \mathcal{V} \setminus \mathcal{E}$ be compact. Then any permutation-equivariant operator $\Psi : \mathcal{V} \to \mathcal{V}$ can be approximated on $K$ by permutation-equivariant weight-space networks.*

As before, Analogous universality results for transformer-weight-space architectures can be established through minor modifications of the proof of Theorem 7.4 (sketch below). Details are provided in Appendix H.

*Proof sketch.* Let $\boldsymbol{v} = (\boldsymbol{W}_1, \boldsymbol{b}_1, \ldots, \boldsymbol{W}_L, \boldsymbol{b}_L) \in K$ and write $\Psi(\boldsymbol{v}) = (\boldsymbol{W}'_1, \boldsymbol{b}'_1, \ldots, \boldsymbol{W}'_L, \boldsymbol{b}'_L)$ for the target output. We first construct a DWS model that approximates an intermediate map $\text{cañon} : \mathcal{V} \to \mathcal{V}^{M+1}$, where $M := \sum_{\ell=1}^{L} d_\ell(1 + d_{\ell-1})$, which augments each parameter entry with a canonical, permutation-invariant summary. Specifically, using the canonization map $\text{canon}$ and the flattening map $\text{Flat}$ defined in the proof of 6.3, we define $\text{cañon}(\boldsymbol{v})$ by broadcasting $\text{Flat}(\text{canon}(\boldsymbol{v}))$ to every weight and bias entry along an additional feature dimension:

$$(\boldsymbol{W}^*_\ell)_{i,j,:} = \big[(\boldsymbol{W}_\ell)_{i,j},\ \text{Flat}(\text{canon}(\boldsymbol{v}))\big] \quad (6)$$

$$(\boldsymbol{b}^*_\ell)_{i,:} = \big[(\boldsymbol{b}_\ell)_i,\ \text{Flat}(\text{canon}(\boldsymbol{v}))\big]. \quad (7)$$

We then show that any equivariant operator $\Psi$ factors through $\text{cañon}$, i.e., there exists a continuous function $\tilde{f} : \mathbb{R}^{M+1} \to \mathbb{R}$ such that for all layers $\ell$ and indices $i, j$,

$$(\boldsymbol{W}'_\ell)_{i,j} = \tilde{f}\big((\boldsymbol{W}^*_\ell)_{i,j,:}\big), \qquad (\boldsymbol{b}'_\ell)_i = \tilde{f}\big((\boldsymbol{b}^*_\ell)_{i,:}\big). \quad (8)$$

This factorization is obtained by first computing the full output $\Psi(\text{canon}(\boldsymbol{v}))$ and then using the local input entry

---

[3]In fact, the same argument shows that the statement extends to scale- and permutation-equivariant networks as well.

to select the corresponding component from it. Since $\tilde{f}$ is continuous on a compact domain, it can be approximated by an MLP applied pointwise across the feature dimension. By Corollary D.15, this pointwise update can be realized by a DWS network, yielding the desired approximation of $\Psi$ on $K$ and completing the proof. $\qquad\square$

## 8. Empirical Gains from Increased Output Capacity

As discussed in Section 7.1, permutation-equivariant weight-space networks are fundamentally limited in approximating function-space operators when constrained to output networks with the same architecture as the input (Proposition 7.1). Our theory further shows that this limitation can be overcome by increasing the output architecture size (Theorem 7.2). Motivated by this insight, we propose Output Capacity Expansion (OCE), a simple modification applicable to any weight-space network: using a final feature dimension of $k > 1$ and interpreting the output as an ensemble of $k$ MLPs, thereby increasing the effective output capacity. The final prediction is obtained by averaging the outputs of these MLPs (See Appendix I.1 for more details). We empirically validate[4] OCE, obtaining substantial improvements over existing weight-space methods with only minor architectural changes and achieving SOTA performance on an established weight-space benchmark.

*Table 1.* Performance on the MNIST INR dilation benchmark (3 seeds). The best two results are highlighted in bold. $k$ denotes the output feature dimension, i.e., the number of models in the output ensemble.

| Method | Reference | MSE in $10^{-2}$ ($\downarrow$) |
|---|---|---|
| NFT | (Zhou et al., 2023b) | $5.10 \pm 0.04$ |
| NP-NFN | (Kofinas et al., 2024) | $2.55 \pm 0.00$ |
| NG-GNN-64 | (Kofinas et al., 2024) | $2.06 \pm 0.01$ |
| ScaleGMN-B | (Kalogeropoulos et al., 2024) | $1.89 \pm 0.00$ |
| NG-T-64 | (Kofinas et al., 2024) | $1.75 \pm 0.01$ |
| ScaleGMN + GradMetaNet++ | (Gelberg et al., 2026) | $1.60 \pm 0.01$ |
| DWS ($k$=1) | (Gelberg et al., 2026) | $2.29 \pm 0.01$ |
| GMN ($k$=1) | (Gelberg et al., 2026) | $1.96 \pm 0.02$ |
| DWS + OCE ($k$=8) | This paper | $\mathbf{1.36 \pm 0.03}$ |
| GMN + OCE ($k$=8) | This paper | $\mathbf{1.06 \pm 0.13}$ |

**Experimental details.** We evaluate OCE on the MNIST INR *dilation* benchmark used in prior work (Zhou et al., 2023a; Kalogeropoulos et al., 2024; Kofinas et al., 2024; Gelberg et al., 2026). This benchmark consists of input–output pairs of MNIST images, where the target image is a dilated version of the input. Importantly, this task corresponds to a function-space operator, since the target depends only on the function realized by the input weights rather than on a specific parameterization. We evaluate OCE over two representative permutation-equivariant weight-space ar-

chitectures: DWS (Navon et al., 2023) and GMN (Lim et al., 2023). Full experimental details are provided in Appendix I.

**Results and discussion.** Our results show that OCE consistently improves the performance of existing weight-space architectures. As shown in Table 1, both DWS+OCE and GMN+OCE achieve strong results, even compared to methods that leverage additional signals such as gradients or probes. In particular, GMN+OCE achieves a $34\%$ improvement over the previous SOTA. Furthermore, Table 2 in Appendix I shows that, for both architectures, increasing the number of ensemble models, and thus the effective output capacity, consistently improves performance, yielding MSE reductions of up to $41\%$ for DWS and $46\%$ for GMN, despite using no additional parameters overall.

These results suggest that a key limitation of prior weight-space approaches on function-space operator tasks stem from the limited capacity of their output representations, highlighting Theorem 7.2 as useful architectural guidance.

## 9. Conclusion

This work develops a comprehensive theoretical foundation for permutation-equivariant weight-space networks. We first show that all prominent weight-space networks effectively fall into a single expressivity class, thus unifying a diverse body of prior work. We then identify four natural and practical settings for weight-space learning: function-space functionals, permutation-invariant functionals, function-space operators, and permutation-equivariant operators, and analyze expressivity in each. Our theoretical results clarify both the capabilities and inherent limitations of weight-space networks, and precisely identify natural conditions, such as GP assumptions and sufficiently large input architectures, under which these networks achieve universality. Motivated by our theory, we propose OCE, an architectural augmentation procedure compatible with all existing weight space models. OCE achieves consistent improvements over base weight space networks achieving SOTA performance on the standard INR dilation benchmark (Zhou et al., 2023a) and demonstrating the practical relevance of our theory. Together, these findings provide principled guidance and theoretical guarantees for the design and analysis of weight-space networks.

**Limitations and future work.** Our theoretical results focus on expressive power, leaving questions of optimization and generalization to future work. Another promising direction is the design of weight-space architectures that map smaller input networks to larger output networks. Our theory suggests that increasing output capacity can help mitigate expressivity limitations in moderate-size settings. While OCE provides a simple first step in this direction, richer and more principled approaches remain to be explored.

---

[4]Code available at `https://github.com/dayanadir/capacity_increase_inr_editing_experiment`.

## Acknowledgements

HM is supported by the Israel Science Foundation through a personal grant (ISF 264/23) and an equipment grant (ISF 532/23), and by the Career Advancement Chairs in Artificial Intelligence – Schmidt Futures. YE is supported by the Zeff PhD fellowship.

## Impact statement

This paper is primarily theoretical, aiming to advance the mathematical foundations of machine learning, in particular weight-space learning. As such, while there may be potential societal consequences for this line of work, we do not believe it raises any direct or immediate societal impact considerations.

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

# Contents

## A. Extended previous work

**Weight-space networks.** A growing body of work studies neural architectures that take the weights of other neural networks as input (Eilertsen et al., 2020; Unterthiner et al., 2020; Schürholt et al., 2021; Dupont et al., 2022; De Luigi et al., 2023), where the dominant design principle in this literature is *equivariance to hidden-neuron permutations*. Several architectural families that respect this symmetry have been proposed. Early permutation-equivariant weight-space networks include Deep Weight Space (DWS) networks (Navon et al., 2023) and Neural Functional Networks (NFNs) (Zhou et al., 2023a), which are constructed by characterizing the space of affine maps equivariant to neuron permutations and composing them with pointwise nonlinearities. Neural Functional Transformers (NFTs) (Zhou et al., 2023b) extend this paradigm by replacing linear equivariant layers with a structured attention mechanism. A parallel line of work encodes neural network parameters as graphs, and applies message passing over them. These include Graph Meta-Networks (GMNs) (Lim et al., 2023) and Neural Graph GNNs (NG-GNNs) (Kofinas et al., 2024). Beyond permutation equivariance, additional works develop weight-space networks that incorporate other symmetry structures inherent to architectures using specific activation functions, such as scale or sign symmetries (Kalogeropoulos et al., 2024; Vo et al., 2024; Tran et al., 2024). Since these networks are defined with respect to different symmetry groups and target different function spaces, we view them as complementary and leave a unified analysis to future work. Other works in the weight-space literature explore related directions, including learning over low-rank adaptations (LoRA) (Putterman et al., 2024), weight-space data augmentation (Shamsian et al., 2024), Kolmogorov–Arnold–based architectures (Elbaz et al., 2025), parameter generation (Wang et al., 2025), and model-zoo construction and analysis (Schürholt et al., 2025; Honegger et al., 2023b;a).

**Expressive power of equivariant networks.** Theoretical analysis of expressive power under symmetry constraints has a long history across data modalities. Perhaps the most thoroughly studied case is graph-structured data, where permutation-equivariant architectures are known to have inherent expressivity limitations (Morris et al., 2019; Xu et al., 2018). This has motivated a large body of work aimed at enhancing GNN expressivity, including higher-order methods (Morris et al., 2019; Maron et al., 2019a), subgraph-based approaches (Zhang & Li, 2021; Cotta et al., 2021; Bevilacqua et al., 2021; Bar-Shalom et al., 2024; Frasca et al., 2022), topological methods (Rieck et al., 2019; Bodnar et al., 2021; Eitan et al., 2025b), positional and structural encodings (Dwivedi et al., 2023; Lim et al., 2022; Eitan et al., 2025a; Bouritsas et al., 2022; Southern et al., 2025), and more. Closely related phenomena arise in $(SO(3))$-equivariant point-cloud networks, where widely used architectures are not universal, motivating higher-order designs to improve expressive power (Dym & Maron, 2020; Hordan et al., 2024a;b). In contrast, for *sets*, classical permutation-invariant architectures such as DeepSets are universal (Qi et al., 2017; Segol & Lipman, 2019). For more structured inputs—such as *sets of symmetric elements* (Maron et al., 2020), gradient bags (Gelberg et al., 2026), and graph foundation models (Finkelshtein et al., 2025)—equivariant architectures are not universal in full generality; however, universality can be recovered under mild assumptions on the input space. Recent work has refined the theory of equivariant approximation by developing general tools that connect separation properties, invariance constraints, and universality, and by characterizing how architectural choices affect expressive power (Pacini et al., 2025b;a).

**Expressive power of weight-space networks.** Despite rapid architectural progress, the theoretical understanding of expressivity in weight-space learning remains limited. Existing results are largely task-driven, showing that specific architectures can realize particular operations on network parameters—for example, approximating forward or backward passes (Navon et al., 2023; Lim et al., 2023; Kalogeropoulos et al., 2024), or subsuming other weight-space models (Lim et al., 2023). While these works establish important capabilities, they do not provide a general characterization of expressive power. In contrast, our work develops a global view of expressivity in weight-space learning by establishing equivalence among permutation-equivariant architectures and proving universality results across several natural approximation settings in weight space. Figure 1 summarizes the current understanding of expressivity in this setting (red arrows).

## B. Extended preliminaries

**Notation.** We begin by introducing basic notation used throughout the paper. An MLP architecture $A$ with $L$ layers is specified by a pair $(\boldsymbol{d}, \sigma)$, where $\boldsymbol{d} = (d_0, \ldots, d_L) \in \mathbb{N}^{L+1}$ denotes the width of each layer (here $d_0$ and $d_L$ are the input and output dimensions respectively) and $\sigma$ is an activation function. Given MLP parameters $\boldsymbol{v} = (\boldsymbol{W}_1, \boldsymbol{b}_1, \ldots, \boldsymbol{W}_L, \boldsymbol{b}_L)$, we let $f_{\boldsymbol{v}} : \mathbb{R}^{d_0} \to \mathbb{R}^{d_L}$ denote the function computed by the network with parameters $\boldsymbol{v}$ and often refer to $f_{\boldsymbol{v}}$ as the function realized by $\boldsymbol{v}$.

$$f_{\boldsymbol{v}}(\boldsymbol{x}) = \boldsymbol{W}_L \, \sigma\Big( \boldsymbol{W}_{L-1} \, \sigma\big( \cdots \sigma(\boldsymbol{W}_1 \boldsymbol{x} + \boldsymbol{b}_1) \cdots \big) + \boldsymbol{b}_{L-1} \Big) + \boldsymbol{b}_L. \tag{9}$$

Given a compact set $X \subseteq \mathbb{R}^n$ and an arbitrary (not necessarily compact) set $Y \subseteq \mathbb{R}^m$, we denote by $\mathcal{C}(X, Y)$ the space of continuous functions $f : X \to Y$, equipped with the uniform norm $\| \cdot \|_\infty$. When $Y = \mathbb{R}$, we write $\mathcal{C}(X)$ for brevity. For an integer $n \in \mathbb{N}$, we denote $[n] = \{1, \ldots, n\}$.

**Weight space of a fixed architecture** Throughout the paper, we consider models whose inputs are the parameters of a fixed MLP architecture. As a first step, we formally define the corresponding input space, namely the weight-space associated with a given architecture.

**Definition B.1** (weight-space). Given an architecture $A = (\boldsymbol{d}, \sigma)$ with $L$ layers, for $\ell = 1, \ldots, L$ the parameters of the $\ell$-th layer consist of a weight matrix and a bias vector

$$\boldsymbol{W}_\ell \in \mathbb{R}^{d_\ell \times d_{\ell-1}}, \qquad \boldsymbol{b}_\ell \in \mathbb{R}^{d_\ell}. \tag{10}$$

We denote by $\mathcal{W}_\ell \cong \mathbb{R}^{d_\ell \times d_{\ell-1}}$ and $\mathcal{B}_\ell \cong \mathbb{R}^{d_\ell}$ the corresponding weight and bias parameter spaces of layer $\ell$.

The weight-space associated with the architecture $A$ is defined as the direct sum

$$\mathcal{V}_A := \left(\bigoplus_{\ell=1}^L \mathcal{W}_\ell\right) \oplus \left(\bigoplus_{\ell=1}^L \mathcal{B}_\ell\right) \cong \bigoplus_{\ell=1}^L \left(\mathbb{R}^{d_\ell \times d_{\ell-1}} \oplus \mathbb{R}^{d_\ell}\right). \tag{11}$$

For a given integer $c \in \mathbb{N}$, we define the weight-space with feature dimension $c$ by

$$\mathcal{V}_A^c := \left(\bigoplus_{\ell=1}^L \mathcal{W}_\ell^c\right) \oplus \left(\bigoplus_{\ell=1}^L \mathcal{B}_\ell^c\right) \cong \bigoplus_{\ell=1}^L \left(\mathbb{R}^{d_\ell \times d_{\ell-1} \times c} \oplus \mathbb{R}^{d_\ell \times c}\right). \tag{12}$$

Elements of $\mathcal{V}_A^c$ are tuples $\boldsymbol{v} = (\boldsymbol{W}_1, \boldsymbol{b}_1, \ldots, \boldsymbol{W}_L, \boldsymbol{b}_L)$, where each weight tensor $\boldsymbol{W}_\ell$ has three indices and each bias tensor $\boldsymbol{b}_\ell$ has two. For a fixed feature index $r \in \{1, \ldots, c\}$, we use the notation

$$\boldsymbol{W}_{\ell,:,:,r} \in \mathbb{R}^{d_\ell \times d_{\ell-1}} \quad \text{and} \quad (\boldsymbol{b}_\ell)_{:,r} \in \mathbb{R}^{d_\ell} \tag{13}$$

to denote the corresponding slices of the weight and bias tensors. Slices across the feature dimension are denoted similarly. For each $r \in \{1, \ldots, c\}$ and $\boldsymbol{v} \in \mathcal{V}_A^c$, we define

$$\boldsymbol{v}_r := \left((\boldsymbol{W}_1)_{:,:,r}, (\boldsymbol{b}_1)_{:,r}, \ldots, (\boldsymbol{W}_L)_{:,:,r}, (\boldsymbol{b}_L)_{:,r}\right). \tag{14}$$

The final axis is referred to as the *feature dimension*.

When the architecture is clear from context, we slightly abuse notation and write $\mathcal{V}$ instead of $\mathcal{V}_A$.

Note that the realization map $\mathcal{R} : \mathcal{V}_A \to \mathcal{C}(X, \mathbb{R}^{d_L})$ that sends weights $\boldsymbol{v}$ to their realized function $f_{\boldsymbol{v}}$ is continuous (see Proposition C.2).

**Weight-space symmetries.** Most weight-space networks (e.g., (Navon et al., 2023; Kofinas et al., 2024; Lim et al., 2023)) are designed to account for permutation symmetries of MLP parameters. Specifically, permuting the neurons within any hidden layer alters the raw parameterization of the network while leaving invariant the underlying function it represents, as well as local geometric properties of the loss landscape. The changes to the weights caused by neuron permutations are represented as a natural group action on the weight-space, which we formalize below.

**Definition B.2** (Weight-space symmetry group). For a given architecture $A = (\boldsymbol{d}, \sigma)$ with $L$ layers, define the corresponding neuron permutation group by

$$G_A := S_{d_1} \times \cdots \times S_{d_{L-1}}. \tag{15}$$

$G_A$ is the direct product of permutation groups of the hidden layers $1, \ldots, L-1$. For $g = (\tau_1, \ldots, \tau_{L-1}) \in G_A$, define an action $\rho(g) : \mathcal{V}_A^c \to \mathcal{V}_A^c$ by

$$\rho(g)(\boldsymbol{W}_1, \boldsymbol{b}_1, \ldots, \boldsymbol{W}_L, \boldsymbol{b}_L) := (\boldsymbol{W}_1', \boldsymbol{b}_1', \ldots, \boldsymbol{W}_L', \boldsymbol{b}_L'), \tag{16}$$

where

$$(\boldsymbol{W}'_1)_{:,:,i} = \boldsymbol{P}_{\tau_1}^\top (\boldsymbol{W}_1)_{:,:,i}, \qquad\qquad (\boldsymbol{b}'_1)_{:,i} = \boldsymbol{P}_{\tau_1}^\top (\boldsymbol{b}_1)_{:,i},$$
$$(\boldsymbol{W}'_l)_{:,:,i} = \boldsymbol{P}_{\tau_l}^\top (\boldsymbol{W}_l)_{:,:,i} \boldsymbol{P}_{\tau_{l-1}}, \qquad\qquad (\boldsymbol{b}'_l)_{:,i} = \boldsymbol{P}_{\tau_l}^\top (\boldsymbol{b}_l)_{:,i}, \quad l = 2, \dots, L-1,$$
$$(\boldsymbol{W}'_L)_{:,:,i} = (\boldsymbol{W}_L)_{:,:,i} \boldsymbol{P}_{\tau_{L-1}}, \qquad\qquad (\boldsymbol{b}'_L)_{:,i} = (\boldsymbol{b}_L)_{:,i}.$$

Here, $\boldsymbol{P}_{\tau_\ell}$ is the permutation matrix corresponding to $\tau_\ell$, and its action is applied independently across the feature dimension.

Similar to before, when the architecture is clear from context, we slightly abuse notation and write $G$ instead of $G_A$.

As noted above, weight-space networks are designed to respect the symmetries induced by the action of a group. We recall the standard notions of invariance and equivariance below.

**Definition B.3** (Invariance and equivariance). Let $G$ be a group and let $(\mathcal{U}, \rho)$ be a representation of $G$. A map $\Psi : \mathcal{U} \to \mathbb{R}^n$ is said to be *G-invariant* if, for every $g \in G$ and every $\boldsymbol{u} \in \mathcal{U}$,

$$\Psi(\rho(g)\,\boldsymbol{u}) = \Psi(\boldsymbol{u}). \tag{17}$$

If $(\mathcal{U}', \rho')$ is another representation of $G$, a map $\Psi : \mathcal{U} \to \mathcal{U}'$ is said to be *G-equivariant* if, for every $g \in G$ and every $\boldsymbol{u} \in \mathcal{U}$,

$$\Psi(\rho(g)\,\boldsymbol{u}) = \rho'(g)\,\Psi(\boldsymbol{u}). \tag{18}$$

We next recall the definition of canonization maps, which will be central to our analysis.

**Definition B.4** (Canonization map). Let $G$ be a group and let $(\mathcal{U}, \rho)$, $(\mathcal{U}', \rho')$ be representations of $G$. A map $\Psi : K \subseteq \mathcal{U} \to \mathcal{U}'$ is said to be a canonization map, if for every $\boldsymbol{u} \in K$

$$\mathrm{canon}(\boldsymbol{v}) = \mathrm{canon}(\rho(g)\boldsymbol{v}) \quad \forall g \in G. \tag{19}$$

Finally, since many of our results involve compact subsets of weight-space that approximate compact subsets of function-space, we formalize this notion below.

**Definition B.5** ($\epsilon$-approximation of sets). Let $A = (\boldsymbol{d}, \sigma)$ be an MLP architecture with corresponding weight-space $\mathcal{V} = \mathcal{V}_A$, let $X \subseteq \mathbb{R}^{d_0}$ be compact, and let $K \subseteq \mathcal{C}(X, \mathbb{R}^{d_L})$ be a compact set of target functions. We say that a compact set $K' \subseteq \mathcal{V}$ is an $\epsilon$-approximation of $K$ if

$$d_H\big(\{f_{\boldsymbol{v}} \mid \boldsymbol{v} \in K'\},\, K\big) < \epsilon, \tag{20}$$

where $d_H$ denotes the Hausdorff distance induced by the uniform norm $\|\cdot\|_\infty$. Put simply, this means that every function in $K$ can be uniformly approximated up to $\epsilon$ by a function realized by parameters in $K'$, and conversely, every function realized by parameters in $K'$ is $\epsilon$-close to some function in $K$.

## C. Topological properties of the realization map

In this section, we establish fundamental topological properties of the realization map, which plays a central role in connecting weight-space networks to function-space functionals. We prove two key results: first, that the realization map is continuous when the function space is equipped with the supremum norm topology; and second, that on compact sets of weights, the supremum topology and quotient topology on the function space coincide. These results are essential for the universality proofs in subsequent sections, as they allow us to relate continuity properties of functionals on function space to continuity properties of their pullbacks on weight space.

Let $A = (\boldsymbol{d}, \sigma)$ be an architecture where $\boldsymbol{d} = (d_0, \dots, d_L)$ and $\mathcal{V}$ its associated weight space. Let $X \subset \mathbb{R}^{d_0}$ be a compact input domain.

**Definition C.1** (Realization map). The *realization map* $\mathcal{R} : \mathcal{V} \to \mathcal{C}(X, \mathbb{R}^{d_L})$ is defined by

$$\mathcal{R}(v) := f_v,$$

where $f_v : X \to \mathbb{R}^{d_L}$ denotes the function realized by the MLP with weights $v$.

We denote

$$\mathcal{F} := \mathcal{R}(\mathcal{V})$$

and endow $\mathcal{F}$ with two topologies:

- the *supremum topology* $\tau_\infty$ induced by the ambient sup norm on $\mathcal{C}(X, \mathbb{R}^{d_L})$;

- the *quotient topology* $\tau_q$ induced by the functional equivalence relation $\sim_{\text{func}}$ (see Definition E.1), transported to $\mathcal{F}$ via the bijection $\mathcal{R}^* : \mathcal{V}/\sim_{\text{func}} \to \mathcal{F}$.

### C.1. Continuity of the realization map

**Proposition C.2** (Continuity of the realization map). *The realization map $\mathcal{R} : \mathcal{V} \to \mathcal{C}(X, \mathbb{R}^{d_L})$ is continuous, where $\mathcal{C}(X, \mathbb{R}^{d_L})$ is equipped with the supremum norm topology.*

*Proof.* For each $v \in \mathcal{V}$, write $v = (\boldsymbol{W}_1, \boldsymbol{b}_1, \ldots, \boldsymbol{W}_L, \boldsymbol{b}_L)$, where $\boldsymbol{W}_\ell \in \mathcal{W}_\ell$ and $\boldsymbol{b}_\ell \in \mathcal{B}_\ell$ denote, respectively, the weights and biases of layer $\ell$. For $x \in X$, the network computes

$$h^{(0)}(x) = x, \qquad h^{(\ell)}(x) = \sigma\big(\boldsymbol{W}_\ell h^{(\ell-1)}(x) + \boldsymbol{b}_\ell\big), \quad \ell = 1, \ldots, L, \tag{21}$$

and $f_v(x)$ is obtained by applying a final affine map to $h^{(L)}(x)$.

Define the function $F : \mathcal{V} \times X \to \mathbb{R}^{d_L}$ by

$$F(v, x) := f_v(x).$$

Each layer computation is obtained from $(v, x)$ by a finite composition of affine maps and the activation $\sigma$, which is continuous by assumption. Therefore $F$ is continuous as a function of both $v$ and $x$.

To show that $\mathcal{R}$ is continuous, it suffices to show that if $v_k \to v$ in $\mathcal{V}$ (in Euclidean norm), then

$$\|f_{v_k} - f_v\|_\infty \to 0.$$

Fix $\epsilon > 0$. Since $F$ is continuous on the product space $\mathcal{V} \times X$ and $X$ is compact, the map $(v, x) \mapsto F(v, x)$ is uniformly continuous on any compact subset of $\mathcal{V} \times X$. Let $B \subset \mathcal{V}$ be a closed Euclidean ball containing $v$ and all $v_k$ for $k$ large enough. Then $B \times X$ is compact, so $F$ is uniformly continuous on $B \times X$.

Hence there exists $\delta > 0$ such that for all $(v', x'), (v'', x'') \in B \times X$,

$$\|(v', x') - (v'', x'')\| < \delta \quad \Rightarrow \quad \|F(v', x') - F(v'', x'')\| < \epsilon.$$

Since $v_k \to v$, there exists $N$ such that for all $k \geq N$, we have $\|v_k - v\| < \delta$. Then for all $x \in X$ and $k \geq N$,

$$|f_{v_k}(x) - f_v(x)| = |F(v_k, x) - F(v, x)| < \epsilon,$$

where the inequality follows from uniform continuity since $\|(v_k, x) - (v, x)\| = \|v_k - v\| < \delta$. Taking the supremum over $x \in X$, we obtain

$$\|f_{v_k} - f_v\|_\infty < \epsilon$$

for all $k \geq N$, which completes the proof. $\square$

### C.2. Equivalence of supremum and quotient topologies on compact sets

Having established the continuity of the realization map, we now investigate the relationship between two natural topologies on the function space $\mathcal{F}$: the supremum topology inherited from the ambient space $\mathcal{C}(X, \mathbb{R}^{d_L})$, and the quotient topology induced by functional equivalence. While these topologies may differ in general, we show that they coincide when restricted to the image of compact weight sets. This equivalence is crucial for establishing that continuous functionals on function space correspond to continuous maps on weight space.

By Proposition C.2, the realization map $\mathcal{R}$ is continuous into $(\mathcal{F}, \tau_\infty)$. Hence the induced map $\mathcal{R}^* : (\mathcal{V}/\sim_{\text{func}}, \tau_q) \to (\mathcal{F}, \tau_\infty)$ is continuous, and therefore

$$\tau_\infty \subseteq \tau_q.$$

In general, we cannot conclude equality of topologies on all of $\mathcal{F}$, since the map $\mathcal{R}$ need not be a quotient map globally.

**Topological equivalence on compact weight sets.**

The following theorem gives a precise condition under which the two topologies coincide.

**Proposition C.3** (Topological Equivalence on Compact Weight Sets). *Let $K \subset \mathcal{V}$ be compact, and define*

$$\mathcal{F}_K := \mathcal{R}(K) \subset \mathcal{C}(X, \mathbb{R}^{d_L}).$$

*Let $\tau_\infty^K$ denote the subspace topology on $\mathcal{F}_K$ inherited from $(\mathcal{C}(X, \mathbb{R}^{d_L}), \|\cdot\|_\infty)$, and let $\tau_q^K$ denote the quotient topology on $\mathcal{F}_K$ induced by the restricted map $\mathcal{R}_K := \mathcal{R}|_K$. Then*

$$\tau_\infty^K = \tau_q^K.$$

*Proof.* The restricted map $\mathcal{R}_K : K \to \mathcal{F}_K$ is continuous. Since $K$ is compact and $(\mathcal{F}_K, \tau_\infty^K)$ is a subspace of a Hausdorff space, the map $\mathcal{R}_K$ is a continuous surjection from a compact space to a Hausdorff space. Every such map is closed and hence a quotient map.

By definition of quotient topology, a set $U \subset \mathcal{F}_K$ is open in $\tau_q^K$ if and only if $\mathcal{R}_K^{-1}(U)$ is open in $K$. Since $\mathcal{R}_K$ is a quotient map into $(\mathcal{F}_K, \tau_\infty^K)$, the same condition characterizes openness in $\tau_\infty^K$. Thus the two topologies coincide:

$$\tau_\infty^K = \tau_q^K.$$

$\square$

**Corollary C.4.** *Let $K \subset \mathcal{V}$ be compact. A function $\Psi : \mathcal{F}_K \to \mathbb{R}$ is continuous with respect to the supremum norm if and only if $\Psi \circ \mathcal{R}_K : K \to \mathbb{R}$ is continuous.*

*Proof.* Since $\mathcal{R}_K : (K, \text{usual topology}) \to (\mathcal{F}_K, \tau_\infty^K)$ is a quotient map, the statement follows from the universal property of quotient topologies. $\square$

# D. Expressive power equivalence of weight-space networks

In this section, we establish that several recently proposed permutation-equivariant weight-space networks have identical expressive power when restricted to MLP weight-space architectures. This result unifies a diverse landscape of architectural designs, showing that despite their different structural forms—ranging from graph-based message passing to transformer-style attention mechanisms—these models can approximate each other to arbitrary precision on any compact set of weights. Our main contribution is to prove that for any MLP architecture $A$, the following model classes can mutually approximate one another:

- DWSNets$_{\text{eq}}^A$ (Navon et al., 2023): Deep networks composed of hidden-neuron permutation equivariant affine layers interleaved with pointwise nonlinearities (Definition D.7).
- HNP-NFN$_{\text{eq}}^A$ (Zhou et al., 2023a): Neural Functional Networks with hidden-neuron permutation equivariance, mathematically equivalent to DWSNets$_{\text{eq}}^A$ (Remark D.8).
- NP-NFN+PE$_{\text{eq}}^A$ (Zhou et al., 2023a): Neural Functional Networks of neuron permutation (including first and last layers) equivariant affine layers applied after a positional encoding that appends one-hot identifiers for the neurons in the first and the last layers (Definition D.9).
- GMN$_{\text{eq}}^A$ (Lim et al., 2023): Message-passing neural networks on the MLP parameter graph with node, edge, and global feature updates (Definition D.10).
- NG$_{\text{eq}}^A$ (Kofinas et al., 2024): A simplified variant of GMN with biases stored as node features and no global features (Definition D.11).
- NFT$_{\text{eq}}^A$ (Zhou et al., 2023b): A transformer-based permutation-equivariant architecture obtained by composing multiple types of equivariant attention updates (Definition D.12).

To establish expressive equivalence among these architectures, we first formalize the notion of expressive equivalence. This requires defining what it means for one model class to be able to express another, and then showing that this relationship is bidirectional. We begin by defining the notion of expressive containment, which captures the ability of one architecture to approximate functions from another.

**Definition D.1** (Expressive containment). Let $X \subseteq \mathbb{R}^n$ and $Y \subseteq \mathbb{R}^m$ be two sets, let $K \subseteq X$ be a compact set, and let $\mathcal{F}_1, \mathcal{F}_2 \subseteq \mathcal{C}(X, Y)$ be two classes of continuous functions. We say that $\mathcal{F}_2$ *can express* $\mathcal{F}_1$ with respect to $K$ if for every $\epsilon > 0$, and every $f_1 \in \mathcal{F}_1$, there exists $f_2 \in \mathcal{F}_2$ such that

$$\sup_{x \in K} \|f_1(x) - f_2(x)\| < \epsilon.$$

We denote this by $\mathcal{F}_1 \preceq_K \mathcal{F}_2$.

*Remark* D.2 (Transitivity of $\preceq_K$). If $\mathcal{F}_1 \preceq_K \mathcal{F}_2$ and $\mathcal{F}_2 \preceq_K \mathcal{F}_3$, then $\mathcal{F}_1 \preceq_K \mathcal{F}_3$. Indeed, let $f_1 \in \mathcal{F}_1$ and $\epsilon > 0$. By assumption, there exists $f_2 \in \mathcal{F}_2$ with $\sup_{x \in K} \|f_1(x) - f_2(x)\| < \epsilon/2$, and there exists $f_3 \in \mathcal{F}_3$ with $\sup_{x \in K} \|f_2(x) - f_3(x)\| < \epsilon/2$. By the triangle inequality, $\sup_{x \in K} \|f_1(x) - f_3(x)\| \leq \sup_{x \in K} \|f_1(x) - f_2(x)\| + \sup_{x \in K} \|f_2(x) - f_3(x)\| < \epsilon$.

**Definition D.3** (Expressive equivalence). Let $X \subseteq \mathbb{R}^n$ and $Y \subseteq \mathbb{R}^m$ be two sets, let $K \subseteq X$ be a compact set, and let $\mathcal{F}_1, \mathcal{F}_2 \subseteq \mathcal{C}(X, Y)$ be two classes of continuous functions. We say that $\mathcal{F}_1$ and $\mathcal{F}_2$ *have equivalent expressive power* with respect to $K$ if $\mathcal{F}_1 \preceq_K \mathcal{F}_2$ and $\mathcal{F}_2 \preceq_K \mathcal{F}_1$ (Definition D.1). We denote this by $\mathcal{F}_1 \simeq_K \mathcal{F}_2$.

*Remark* D.4 ($\simeq_K$ is an equivalence relation). For any compact set $K$, $\simeq_K$ is reflexive, symmetric, and transitive on $\mathcal{C}(X, Y)$. Reflexivity and symmetry are immediate from the definition. Transitivity follows from the transitivity of the $\preceq_K$ relation in Remark D.2.

*Remark* D.5 (Equivalence of definitions). The definition of expressive equivalence given above (Definition D.3) works with explicit network sets, such as $\mathrm{DWSNets}^A_{\mathrm{eq}}$ or $\mathrm{GMN}^A_{\mathrm{eq}}$, and depends on a specific compact set $K$. In contrast, the main paper (Definition 5.1) defines expressive power in terms of the set of continuous functions that can be approximated on compact sets, denoted $\mathcal{N}^\pi(K; Y)$ for a network architecture $\pi$ and compact set $K$. These two definitions are equivalent: when $\pi$ is an explicit network set (e.g., $\pi = \mathrm{DWSNets}^A_{\mathrm{eq}}$), the set $\mathcal{N}^\pi(K; Y)$ is precisely the closure of $\pi$ under the topology of uniform convergence on the compact set $K$. Consequently, for any compact set $K$, two network sets $\pi$ and $\pi'$ have equivalent expressive power in the sense of Definition D.3 (i.e., $\pi \simeq_K \pi'$) if and only if $\mathcal{N}^\pi(K; Y) = \mathcal{N}^{\pi'}(K; Y)$, establishing the equivalence of the two definitions.

## D.1. Main equivalence result

Having established the formal framework for expressive equivalence, we now state our main result. This theorem establishes that five prominent permutation-equivariant weight-space networks—DWSNets, HNP-NFN, NP-NFN+PE, GMN, and NG-GNN—have identical expressive power. The result holds for weight-space networks with arbitrary input and output feature dimensions, and applies uniformly across all compact sets of weights. Let $c_{\mathrm{in}}, c_{\mathrm{out}} \in \mathbb{N}$ denote the input and output feature dimensions, respectively.

**Theorem D.6** (Expressive equivalence of weight-space architectures). *For any MLP architecture $A = (\boldsymbol{d}, \sigma)$ and any compact set $K \subseteq \mathcal{V}^{c_{\mathrm{in}}}_A$,*

$$\mathrm{DWSNets}^A_{\mathrm{eq}} \simeq_K \mathrm{HNP\text{-}NFN}^A_{\mathrm{eq}} \simeq_K \mathrm{NP\text{-}NFN+PE}^A_{\mathrm{eq}} \simeq_K \mathrm{GMN}^A_{\mathrm{eq}} \simeq_K \mathrm{NG}^A_{\mathrm{eq}} \tag{22}$$

*(Definition D.3), where the model classes are defined in (D.7), (D.8), (D.9), (D.10), and (D.11), respectively.*

The proof of Theorem D.6 proceeds in several stages. We first provide formal definitions of each model class, establishing the precise mathematical structure of each architecture. We then develop a proof strategy based on cyclic equivalence arguments, showing that each architecture can approximate the next in a closed cycle. This approach allows us to establish mutual expressivity without requiring direct pairwise comparisons between all architectures.

## D.2. Network definitions

In this subsection, we provide formal definitions for each of the weight-space architectures considered in this section. We begin with notation for one-hot encodings, which are used extensively across several architectures to represent discrete features of the MLP parameters.

**Notation (one-hot encoding).** Several architectures use one-hot encodings to represent discrete attributes of weights and biases in the MLP (e.g., layer index, neuron type, edge direction). For any positive integer $n$ and index $k \in [n]$, we denote by $e_k^{(n)} \in \mathbb{R}^n$ the $k$-th standard basis vector, i.e., the one-hot encoding of index $k$ in $\mathbb{R}^n$:

$$\left(e_k^{(n)}\right)_j \;=\; \mathbf{1}[j = k] \;=\; \begin{cases} 1 & \text{if } j = k, \\ 0 & \text{otherwise.} \end{cases}$$

**DWSNets and HNP–NFN.**

**Definition D.7** (DWSNets$_{\mathrm{eq}}^A$ (Navon et al., 2023)). A DWS equivariant network $\Phi : \mathcal{V}_A^{c_{\mathrm{in}}} \to \mathcal{V}_A^{c_{\mathrm{out}}}$ is a deep architecture of the form

$$\text{DWSNets}_{\mathrm{eq}}^A \;:=\; \left\{ L_n \circ \sigma \circ L_{n-1} \circ \cdots \circ \sigma \circ L_1 \;\middle|\; \begin{array}{l} L_i \in \mathrm{Aff}_G\left(\mathcal{V}_A^{c_{i-1}}, \mathcal{V}_A^{c_i}\right), \quad i = 1, \ldots, n, \\ n \in \mathbb{N}, \quad c_i \in \mathbb{N}, \\ c_0 = c_{\mathrm{in}}, \quad c_n = c_{\mathrm{out}} \end{array} \right\}. \tag{23}$$

*Remark* D.8 (Equivalence of HNP–NFN and DWSNets). (Zhou et al., 2023a) introduced *HNP–NFNs* (Hidden Neuron Permutation Neural Functional Networks), which are mathematically equivalent to DWSNets. In both cases, the admissible linear layers are precisely the $G$-equivariant linear maps $\mathrm{Hom}_G(\mathcal{V}_A^{c_{\mathrm{in}}}, \mathcal{V}_A^{c_{\mathrm{out}}})$. Accordingly, we treat DWSNets$_{\mathrm{eq}}^A$ and HNP-NFN$_{\mathrm{eq}}^A$ as *synonymous* model classes throughout this appendix.

**NP–NFN with positional encoding.**

**Definition D.9** (NP-NFN+PE$_{\mathrm{eq}}^A$ (Zhou et al., 2023a)). Let

$$S := S_{d_0} \times G \times S_{d_L}$$

denote the full NP group.

**Neuron type set.** Define the neuron type set

$$\mathcal{T}_{\mathrm{PE}} := \{\mathtt{in}_1, \ldots, \mathtt{in}_{d_0}\} \cup \{\mathtt{out}_1, \ldots, \mathtt{out}_{d_L}\} \cup \{\mathtt{hidden}\}, \qquad |\mathcal{T}_{\mathrm{PE}}| = d_0 + d_L + 1.$$

Each neuron type $t \in \mathcal{T}_{\mathrm{PE}}$ is encoded as a one-hot vector $e_t \in \mathbb{R}^{|\mathcal{T}_{\mathrm{PE}}|}$.

**Positional encoding.** Define the *positional encoding* $\mathrm{PE} : \mathcal{V}_A^c \to \mathcal{V}_A^{c+2|\mathcal{T}_{\mathrm{PE}}|}$ by appending neuron type identifiers as additional channels. For $v = \{(W_\ell, b_\ell)\}_{\ell=1}^L \in \mathcal{V}_A^c$, we define $\mathrm{PE}(v) := \{(W_\ell^*, b_\ell^*)\}_{\ell=1}^L$, where each weight entry receives two type encodings—one for the *source* neuron and one for the *target* neuron:

$$(W_\ell^*)_{i,j,:} \;:=\; \begin{cases} [(W_1)_{i,j,:}, \; e_{\mathtt{in}_j}, \; e_{\mathtt{hidden}}] & \text{if } \ell = 1, \\ [(W_\ell)_{i,j,:}, \; e_{\mathtt{hidden}}, \; e_{\mathtt{hidden}}] & \text{if } 1 < \ell < L, \\ [(W_L)_{i,j,:}, \; e_{\mathtt{hidden}}, \; e_{\mathtt{out}_i}] & \text{if } \ell = L, \end{cases} \tag{24}$$

and each bias entry receives a zero-padded type encoding to match the feature dimension of weights:

$$(b_\ell^*)_{i,:} \;:=\; \begin{cases} [(b_\ell)_{i,:}, \; \mathbf{0}_{|\mathcal{T}_{\mathrm{PE}}|}, \; e_{\mathtt{hidden}}] & \text{if } 1 \le \ell < L, \\ [(b_L)_{i,:}, \; \mathbf{0}_{|\mathcal{T}_{\mathrm{PE}}|}, \; e_{\mathtt{out}_i}] & \text{if } \ell = L. \end{cases} \tag{25}$$

Here, $e_{\mathtt{in}_j}$ and $e_{\mathtt{out}_i}$ encode the unique identity of input neuron $j$ and output neuron $i$, respectively, while $e_{\mathtt{hidden}}$ is shared by all hidden neurons.

**Model class.** We define the NP–NFN+PE model class as

$$\text{NP-NFN+PE}_{\mathrm{eq}}^A \;:=\; \left\{ L_n \circ \sigma \circ L_{n-1} \circ \cdots \circ \sigma \circ L_1 \circ \mathrm{PE} \;\middle|\; \begin{array}{l} L_i \in \mathrm{Aff}_S\left(\mathcal{V}_A^{c_{i-1}}, \mathcal{V}_A^{c_i}\right), \quad i = 1, \ldots, n, \\ n \in \mathbb{N}, \quad c_i \in \mathbb{N}, \\ c_0 = c_{\mathrm{in}} + 2|\mathcal{T}_{\mathrm{PE}}| = c_{\mathrm{in}} + 2(d_0 + d_L + 1), \quad c_n = c_{\mathrm{out}} \end{array} \right\}. \tag{26}$$

**Message-Passing on MLP Parameter Graphs.** Both (Lim et al., 2023) (Graph MetaNetworks; GMN) and (Kofinas et al., 2024) (Neural Graphs; NG) propose applying message-passing neural networks to MLPs by viewing the MLP as an undirected graph. We first define the GMN framework on MLP graphs (Definition D.10), then describe the Neural Graphs variant which differs in its bias representation and does not allow global features (Definition D.11).

**Definition D.10** (GMN$_{\text{eq}}^A$ (Lim et al., 2023))**.** Fix an $L$-layer MLP with widths $(d_0, \ldots, d_L)$ and $c_{\text{in}}$ input channels, and let

$$\{(\boldsymbol{W}_\ell, \boldsymbol{b}_\ell)\}_{\ell=1}^L \in \mathcal{V}_A^{c_{\text{in}}}, \qquad \boldsymbol{W}_\ell \in \mathbb{R}^{d_\ell \times d_{\ell-1} \times c_{\text{in}}}, \quad \boldsymbol{b}_\ell \in \mathbb{R}^{d_\ell \times c_{\text{in}}}.$$

**Graph construction.** Define $\mathsf{G}_{\text{MLP}} = (N, E)$ as the directed MLP parameter graph with:

- Node set $N = \bigcup_{\ell=0}^L \{\nu_i^{(\ell)} : i \in [d_\ell]\} \cup \bigcup_{\ell=1}^L \{\beta^{(\ell)}\}$, comprising one *neuron node* $\nu_i^{(\ell)}$ per neuron and one *bias node* $\beta^{(\ell)}$ per layer.
- Edge set $E = E_{\text{w}} \cup E_{\text{b}}$, where

$$E_{\text{w}} := \{(\nu_j^{(\ell-1)}, \nu_i^{(\ell)}), (\nu_i^{(\ell)}, \nu_j^{(\ell-1)}) : \ell \in [L], i \in [d_\ell], j \in [d_{\ell-1}]\},$$
$$E_{\text{b}} := \{(\beta^{(\ell)}, \nu_i^{(\ell)}), (\nu_i^{(\ell)}, \beta^{(\ell)}) : \ell \in [L], i \in [d_\ell]\}.$$

**Edge feature initialization.** Define a map $p : E \to \mathbb{R}^{c_{\text{in}}}$ extracting the MLP parameters associated with each edge:

$$p(e) := \begin{cases} (\boldsymbol{W}_\ell)_{i,j,:} \in \mathbb{R}^{c_{\text{in}}}, & e \in \{(\nu_j^{(\ell-1)}, \nu_i^{(\ell)}), (\nu_i^{(\ell)}, \nu_j^{(\ell-1)})\}, \\ (\boldsymbol{b}_\ell)_{i,:} \in \mathbb{R}^{c_{\text{in}}}, & e \in \{(\beta^{(\ell)}, \nu_i^{(\ell)}), (\nu_i^{(\ell)}, \beta^{(\ell)})\}. \end{cases}$$

We define three discrete edge attributes, each represented as a one-hot vector:

- *Layer index* layer $: E \to \mathbb{R}^L$, where $\text{layer}(e) := \boldsymbol{e}_\ell^{(L)}$ is the one-hot encoding of $\ell \in [L]$;
- *Direction* dir $: E \to \mathbb{R}^2$, where $\text{dir}(e) := \boldsymbol{e}_1^{(2)}$ for forward edges and $\text{dir}(e) := \boldsymbol{e}_2^{(2)}$ for backward edges;
- *Parameter type* ptype $: E \to \mathbb{R}^2$, where $\text{ptype}(e) := \boldsymbol{e}_1^{(2)}$ for weight edges and $\text{ptype}(e) := \boldsymbol{e}_2^{(2)}$ for bias edges.

Concretely, for weight edges $e \in E_{\text{w}}$:

$$\text{layer}(e) := \boldsymbol{e}_\ell^{(L)}, \quad \text{ptype}(e) := \boldsymbol{e}_1^{(2)}, \quad \text{dir}(e) := \begin{cases} \boldsymbol{e}_1^{(2)}, & e = (\nu_j^{(\ell-1)}, \nu_i^{(\ell)}), \\ \boldsymbol{e}_2^{(2)}, & e = (\nu_i^{(\ell)}, \nu_j^{(\ell-1)}), \end{cases}$$

and for bias edges $e \in E_{\text{b}}$:

$$\text{layer}(e) := \boldsymbol{e}_\ell^{(L)}, \quad \text{ptype}(e) := \boldsymbol{e}_2^{(2)}, \quad \text{dir}(e) := \begin{cases} \boldsymbol{e}_1^{(2)}, & e = (\beta^{(\ell)}, \nu_i^{(\ell)}), \\ \boldsymbol{e}_2^{(2)}, & e = (\nu_i^{(\ell)}, \beta^{(\ell)}). \end{cases}$$

The initial edge feature vector $\boldsymbol{e}_e \in \mathbb{R}^{d_e}$ with $d_e := c_{\text{in}} + L + 4$ is the concatenation:

$$\boldsymbol{e}_e := [p(e), \text{layer}(e), \text{dir}(e), \text{ptype}(e)] \in \mathbb{R}^{c_{\text{in}} + L + 2 + 2}.$$

**Node feature initialization.** We define two discrete node features, each represented as a one-hot vector:

- *Layer index* nlayer $: N \to \mathbb{R}^{L+1}$, where $\text{nlayer}(v) := \boldsymbol{e}_\ell^{(L+1)}$ is the one-hot encoding of the layer index $\ell \in \{0, \ldots, L\}$:

$$\text{nlayer}(\nu_i^{(\ell)}) := \boldsymbol{e}_{\ell+1}^{(L+1)}, \qquad \text{nlayer}(\beta^{(\ell)}) := \boldsymbol{e}_{\ell+1}^{(L+1)};$$

- *Node type* ntype $: N \to \mathbb{R}^{|\mathcal{T}_{\text{GMN}}|}$, where $|\mathcal{T}_{\text{GMN}}| = d_0 + d_L + L + 1$ and

$$\mathcal{T}_{\text{GMN}} := \{\text{in}_1, \ldots, \text{in}_{d_0}\} \cup \{\text{out}_1, \ldots, \text{out}_{d_L}\} \cup \{\text{bias}_1, \ldots, \text{bias}_L\} \cup \{\text{hidden}\}.$$

Each node type $t \in \mathcal{T}_{\text{GMN}}$ is encoded as a one-hot vector $\boldsymbol{e}_t^{(|\mathcal{T}_{\text{GMN}}|)} \in \mathbb{R}^{|\mathcal{T}_{\text{GMN}}|}$:

$$\text{ntype}(v) := \begin{cases} \boldsymbol{e}_{\text{in}_i}^{(|\mathcal{T}_{\text{GMN}}|)}, & v = \nu_i^{(0)}, \\ \boldsymbol{e}_{\text{out}_i}^{(|\mathcal{T}_{\text{GMN}}|)}, & v = \nu_i^{(L)}, \\ \boldsymbol{e}_{\text{hidden}}^{(|\mathcal{T}_{\text{GMN}}|)}, & v = \nu_i^{(\ell)} \text{ for } 1 \le \ell \le L-1, \\ \boldsymbol{e}_{\text{bias}_\ell}^{(|\mathcal{T}_{\text{GMN}}|)}, & v = \beta^{(\ell)}. \end{cases}$$

The initial node feature vector $\boldsymbol{h}_v \in \mathbb{R}^{d_h}$ with $d_h = (L+1) + (d_0 + d_L + L + 1) = 2L + d_0 + d_L + 2$ is the concatenation:

$$\boldsymbol{h}_v := \begin{bmatrix} \mathrm{nlayer}(v), \ \mathrm{ntype}(v) \end{bmatrix} \in \mathbb{R}^{2L+d_0+d_L+2}.$$

**Global feature initialization.** The global feature vector is initialized to zero:

$$\boldsymbol{u} := \boldsymbol{0}_{d_u} \in \mathbb{R}^{d_u},$$

where $d_u \in \mathbb{N}$ is a hyperparameter specifying the global feature dimension.

**MPNN layer update.** A *GMN layer* is a tuple of MLPs $\mathcal{L} = (\phi_m, \phi_h, \phi_e, \phi_u)$ that updates the state $(\boldsymbol{h}, \boldsymbol{e}, \boldsymbol{u}) \mapsto (\boldsymbol{h}', \boldsymbol{e}', \boldsymbol{u}')$ via:

$$\boldsymbol{m}_{j \to i} = \phi_m\big(\boldsymbol{h}_i, \boldsymbol{h}_j, \boldsymbol{e}_{ij}, \boldsymbol{u}\big), \tag{27}$$

$$\boldsymbol{h}'_i = \phi_h\Big(\boldsymbol{h}_i, \sum_{j \in \mathcal{N}(i)} \boldsymbol{m}_{j \to i}, \ \boldsymbol{u}\Big), \tag{28}$$

$$\boldsymbol{e}'_{ij} = \phi_e\big(\boldsymbol{h}_i, \boldsymbol{h}_j, \boldsymbol{e}_{ij}, \boldsymbol{u}\big), \tag{29}$$

$$\boldsymbol{u}' = \phi_u\Big(\sum_{v \in \mathcal{V}} \boldsymbol{h}_v, \sum_{e \in \mathcal{E}} \boldsymbol{e}_e, \ \boldsymbol{u}\Big), \tag{30}$$

where $\mathcal{N}(i)$ denotes the neighbors of node $i$ in the undirected graph (messages flow in both directions along edges), and $\boldsymbol{u} \in \mathbb{R}^{d_u}$ is a global feature vector.

**Model class.** We define the GMN-on-MLP model class as

$$\mathrm{GMN}^A_{\mathrm{eq}} := \left\{ L_n \circ L_{n-1} \circ \cdots \circ L_1 \ \middle| \ \begin{array}{l} L_i = (\phi_{m,i}, \phi_{h,i}, \phi_{e,i}, \phi_{u,i}) \text{ is a GMN layer} \\ n \in \mathbb{N}, \quad i = 1, \ldots, n \end{array} \right\}, \tag{31}$$

**Definition D.11** ($\mathrm{NG}^A_{\mathrm{eq}}$ (Kofinas et al., 2024)). Neural Graphs are a simplified variant of GMN (Definition D.10) with biases stored as node features and no global features.

**Graph construction.** Define $\mathsf{G}_{\mathrm{NG}} = (N, E)$ with:

- Node set $N = \bigcup_{\ell=0}^{L} \{\nu_i^{(\ell)} : i \in [d_\ell]\}$ (neuron nodes only, no bias nodes);
- Edge set $E = E_{\mathrm{w}}$ (weight edges only, as defined in Definition D.10).

**Edge feature initialization.** Using the edge attribute functions $p$, layer, and dir from Definition D.10, the initial edge feature vector $\boldsymbol{e}_e \in \mathbb{R}^{d_e}$ with $d_e := c_{\mathrm{in}} + L + 2$ is:

$$\boldsymbol{e}_e := \begin{bmatrix} p(e), \ \mathrm{layer}(e), \ \mathrm{dir}(e) \end{bmatrix} \in \mathbb{R}^{c_{\mathrm{in}}+L+2}.$$

Note that ptype is omitted since all edges are weight edges.

**Node feature initialization.** Node features include biases as additional channels (for $\ell > 0$) and node types via one-hot encoding. We adapt the node type function from Definition D.10 to exclude bias nodes:

$$\mathcal{T}_{\mathrm{NG}} := \{\mathtt{in}_1, \ldots, \mathtt{in}_{d_0}\} \cup \{\mathtt{out}_1, \ldots, \mathtt{out}_{d_L}\} \cup \{\mathtt{hidden}\}, \qquad |\mathcal{T}_{\mathrm{NG}}| = d_0 + d_L + 1.$$

The node type function $\mathrm{ntype} : N \to \mathbb{R}^{|\mathcal{T}_{\mathrm{NG}}|}$ is defined as:

$$\mathrm{ntype}(v) := \begin{cases} \boldsymbol{e}_{\mathtt{in}_i}^{(|\mathcal{T}_{\mathrm{NG}}|)}, & v = \nu_i^{(0)}, \\ \boldsymbol{e}_{\mathtt{out}_i}^{(|\mathcal{T}_{\mathrm{NG}}|)}, & v = \nu_i^{(L)}, \\ \boldsymbol{e}_{\mathtt{hidden}}^{(|\mathcal{T}_{\mathrm{NG}}|)}, & v = \nu_i^{(\ell)} \text{ for } 1 \le \ell \le L-1. \end{cases}$$

Using nlayer from Definition D.10, the initial node feature vector $\boldsymbol{h}_{\nu_i^{(\ell)}} \in \mathbb{R}^{d_h}$ with $d_h := (L+1) + (d_0 + d_L + 1) + c_{\text{in}}$ is:

$$\boldsymbol{h}_{\nu_i^{(\ell)}} := \begin{cases} \left[\text{nlayer}(\nu_i^{(0)}),\ \text{ntype}(\nu_i^{(0)}),\ \boldsymbol{0}_{c_{\text{in}}}\right] & \text{if } \ell = 0, \\ \left[\text{nlayer}(\nu_i^{(\ell)}),\ \text{ntype}(\nu_i^{(\ell)}),\ (\boldsymbol{b}_\ell)_{i,:}\right] & \text{if } 0 < \ell \le L. \end{cases}$$

**MPNN layer update (no global features).** An *NG layer* is a tuple of MLPs $\mathcal{L} = (\phi_m, \phi_h, \phi_e)$ that updates the state $(\boldsymbol{h}, \boldsymbol{e}) \mapsto (\boldsymbol{h}', \boldsymbol{e}')$ via:

$$\boldsymbol{m}_{j \to i} = \phi_m\big(\boldsymbol{h}_i, \boldsymbol{h}_j, \boldsymbol{e}_{ij}\big), \tag{32}$$

$$\boldsymbol{h}'_i = \phi_h\Big(\boldsymbol{h}_i, \sum_{j \in \mathcal{N}(i)} \boldsymbol{m}_{j \to i}\Big), \tag{33}$$

$$\boldsymbol{e}'_{ij} = \phi_e(\boldsymbol{h}_i, \boldsymbol{h}_j, \boldsymbol{e}_{ij}), \tag{34}$$

where $\mathcal{N}(i)$ denotes the neighbors of node $i$.

**Model class.** We define the Neural Graphs model class as

$$\text{NG}_{\text{eq}}^A := \left\{ L_n \circ L_{n-1} \circ \cdots \circ L_1 \ \middle|\ \begin{array}{l} L_i = (\phi_{m,i}, \phi_{h,i}, \phi_{e,i}) \text{ is an NG layer} \\ n \in \mathbb{N}, \qquad i = 1, \ldots, n \end{array} \right\}, \tag{35}$$

**NFTs.**

**Definition D.12** (NFTs (Zhou et al., 2023b))**.**

**Attention primitive.** NFTs are built from dot-product attention. Given a query $q \in \mathbb{R}^d$ and key–value pairs $\{(k_p, v_p)\}_{p=1}^N$ with $k_p \in \mathbb{R}^d$ and $v_p \in \mathbb{R}^{d_v}$, define

$$\text{ATTN}\big(q, \{(k_p, v_p)\}_{p=1}^N\big) := \sum_{p=1}^N \alpha_p(q)\, v_p, \qquad \alpha_p(q) := \frac{\exp(\langle q, k_p \rangle)}{\sum_{p'=1}^N \exp(\langle q, k_{p'} \rangle)}. \tag{36}$$

When the query $q$ is array-valued, we apply ATTN elementwise in the obvious way.

**Pointwise operators on weight features.** We extend standard pointwise Transformer components to weight-space elements by applying them independently to each feature vector. Concretely, for $\boldsymbol{v} = (\boldsymbol{W}_1, \boldsymbol{b}_1, \ldots, \boldsymbol{W}_L, \boldsymbol{b}_L)$ we define $\text{LN}(\boldsymbol{v}) = (\boldsymbol{W}'_1, \boldsymbol{b}'_1, \ldots, \boldsymbol{W}'_L, \boldsymbol{b}'_L)$ and $\text{MLP}(\boldsymbol{v}) = (\boldsymbol{W}''_1, \boldsymbol{b}''_1, \ldots, \boldsymbol{W}''_L, \boldsymbol{b}''_L)$ by

$$(\boldsymbol{W}'_\ell)_{i,j,:} = \text{LN}\big((\boldsymbol{W}_\ell)_{i,j,:}\big), \qquad (\boldsymbol{b}'_\ell)_{i,:} = \text{LN}\big((\boldsymbol{b}_\ell)_{i,:}\big), \tag{37}$$

and

$$(\boldsymbol{W}''_\ell)_{i,j,:} = \text{MLP}\big((\boldsymbol{W}_\ell)_{i,j,:}\big), \qquad (\boldsymbol{b}''_\ell)_{i,:} = \text{MLP}\big((\boldsymbol{b}_\ell)_{i,:}\big), \tag{38}$$

where $\text{LN}: \mathbb{R}^c \to \mathbb{R}^c$ denotes LayerNorm and $\text{MLP}: \mathbb{R}^c \to \mathbb{R}^c$ is a feed-forward network (e.g., $c \to c_{\text{ff}} \to c$ with a nonlinearity).

**Layer position encodings.** To distinguish different base-network layers, NFTs add a learned embedding to each weight layer. Let $\phi_\ell^b, \phi_\ell^w \in \mathbb{R}^c$ be trainable *layer encodings* for $\ell = 1, \ldots, L$. Define $\text{LAYERENC}: \mathcal{V}_A^c \to \mathcal{V}_A^c$ by $\text{LAYERENC}(\boldsymbol{v}) = (\boldsymbol{W}'_1, \boldsymbol{b}'_1, \ldots, \boldsymbol{W}'_L, \boldsymbol{b}'_L)$

$$(\boldsymbol{W}_\ell)_{i,j,:} + \phi_\ell^w, \qquad (\boldsymbol{W}_\ell)_{i,j,:} + \phi_\ell^b \tag{39}$$

**Linear projections.** Let $\theta_Q, \theta_K, \theta_V \in \mathbb{R}^{c \times c}$ be trainable matrices. Given $\boldsymbol{v} \in \mathcal{V}_A^c$, define for each layer $\ell$ and index pair $(i, j)$

$$(\boldsymbol{Q}_\ell^w)_{i,j,:} := \theta_Q(\boldsymbol{W}_\ell)_{i,j,:}, \qquad (\boldsymbol{K}_\ell^w)_{i,j,:} := \theta_K(\boldsymbol{W}_\ell)_{i,j,:}, \qquad (\boldsymbol{V}_\ell^w)_{i,j,:} := \theta_V(\boldsymbol{W}_\ell)_{i,j,:}, \tag{40}$$

$$(\boldsymbol{Q}_\ell^b)_{i,:} := \theta_Q(\boldsymbol{b}_\ell)_{i,:}, \qquad (\boldsymbol{K}_\ell^b)_{i,:} := \theta_K(\boldsymbol{b}_\ell)_{i,:}, \qquad (\boldsymbol{V}_\ell^b)_{i,:} := \theta_V(\boldsymbol{b}_\ell)_{i,:}, \tag{41}$$

We will also use the shorthand $(\boldsymbol{K}, \boldsymbol{V})_{i,j,k}^{(\ell)} := ((\boldsymbol{K}_\ell)_{i,j,k}, (\boldsymbol{V}_\ell)_{i,j,k})$.

**Self Attention.** NFT self-attention aggregates information in several ways. The full attention map $\mathrm{SA}: \mathcal{V}_A^c \to \mathcal{V}_A^c$ it uses is given by $\mathrm{SA}(\boldsymbol{v}) = (\boldsymbol{W}_1', \boldsymbol{b}_1', \ldots, \boldsymbol{W}_L', \boldsymbol{b}_L')$

$$(\boldsymbol{W}_\ell')_{i,j,:} = \mathrm{ATTN}((\boldsymbol{Q}_\ell^w)_{i,:,:}, \mathrm{KV1}) + \mathrm{ATTN}((\boldsymbol{Q}_\ell^w)_{:,j,:}, \mathrm{KV2}) + \mathrm{ATTN}((\boldsymbol{Q}_\ell^w)_{i,j,:}, \mathrm{KV3}) \tag{42}$$

$$(\boldsymbol{b}_\ell')_{i,:} = \mathrm{ATTN}((\boldsymbol{Q}_\ell^b)_{i,:}, \mathrm{KV2}) + \mathrm{ATTN}((\boldsymbol{Q}_\ell^b)_{i,:}, \mathrm{KV3}) \tag{43}$$

where

$$\mathrm{KV1} = \left\{ (K, V)_{:,q,:}^{(\ell-1)} \right\}_{q=1}^{d_{\ell-2}} \cup \left\{ (\boldsymbol{K}^w, \boldsymbol{V}^w)_{p,:}^{(\ell)} \right\}_{p=1}^{d_\ell} \cup \left\{ (\boldsymbol{K}^b, \boldsymbol{V}^b)^{(\ell-1)} \right\}, \tag{44}$$

$$\mathrm{KV2} = \left\{ (\boldsymbol{K}^w, \boldsymbol{V}^w)_{:,q,:}^{(\ell)} \right\}_{q=1}^{d_{\ell-1}} \cup \left\{ (\boldsymbol{K}^w, \boldsymbol{V}^w)_{p,:,:}^{(\ell+1)} \right\}_{p=1}^{d_{\ell+1}} \cup \left\{ (\boldsymbol{K}^b, \boldsymbol{V}^b)^{(\ell)} \right\}, \tag{45}$$

$$\mathrm{KV3} = \left\{ (\boldsymbol{K}^w, \boldsymbol{V}^w)_{p,q,:}^{(s)} : \forall s, p, q \right\} \cup \left\{ (\boldsymbol{K}^b, \boldsymbol{V}^b)_{p,:}^{(s)} : \forall s, p \right\}, \tag{46}$$

We note that each such attention summand can use its own separate key,query and value projections $\theta_Q, \theta_K, \theta_V$.

**NFT block.** An NFT block is a Transformer-style residual update operating on weight-space features:

$$\boldsymbol{z} := \boldsymbol{v} + \mathrm{SA}(\mathrm{LN}(\boldsymbol{v})), \tag{47}$$
$$\mathrm{BLOCK}(\boldsymbol{v}) := \boldsymbol{z} + \mathrm{MLP}(\mathrm{LN}(\boldsymbol{z})). \tag{48}$$

We note that while the full architecture proposed in (Zhou et al., 2023b) may use multi-headed attention, we keep our formulation in single-head form for simplicity.

**Invariant pooling via cross-attention.** To obtain a permutation-invariant representation, NFTs pool the final weight-space features into a fixed-size vector using cross-attention $\mathrm{CA}: \mathcal{V}_A^c \to \mathbb{R}^c$ between all weight-space entries and a single learnable token, followed by a final MLP.

**Model class.** We define the NFTE model class as

$$\mathrm{NFT}_{\mathrm{eq}}^A := \left\{ \mathrm{MLP} \circ \mathrm{CA} \circ \mathrm{BLOCK}_n \circ \cdots \circ \mathrm{BLOCK}_1 \circ \mathrm{LAYERENC} \circ \mathrm{MLP} \mid n \in \mathbb{N} \right\}. \tag{49}$$

### D.3. Proof strategy and supporting lemmas

Before we prove Theorem D.6, we state and prove a few key lemmas that will be used in the proof.

**Layer-wise approximation implies network approximation.** A key technical insight is that to show one architecture can express another, it suffices to approximate each layer of the target architecture using the source architecture. The following lemma formalizes why layerwise approximations compose to yield approximations of full networks, which is essential for our cyclic argument.

**Lemma D.13** (Layer-wise approximation implies network approximation). *Let $X \subset \mathbb{R}^{d_0}$ be a compact domain, and let $\mathcal{F}_1, \ldots, \mathcal{F}_L$ be families of continuous functions where $\mathcal{F}_i$ consists of functions from $\mathbb{R}^{d_{i-1}} \to \mathbb{R}^{d_i}$ for some $d_1, \ldots, d_L$. Let $\mathcal{F}$ be the family of functions $\{ f_L \circ \cdots \circ f_1 : X \to \mathbb{R}^{d_L} \mid f_i \in \mathcal{F}_i \}$ that are compositions of functions $f_i \in \mathcal{F}_i$.*

*Suppose that for each $i = 1, \ldots, L$, there exists a family $\tilde{\mathcal{F}}_i$ such that for every $f_i \in \mathcal{F}_i$ and every $\epsilon > 0$, there exists $\tilde{f}_i \in \tilde{\mathcal{F}}_i$ that uniformly approximates $f_i$ on its domain:*

$$\sup_{\boldsymbol{x} \in \mathrm{dom}(f_i)} \|\tilde{f}_i(\boldsymbol{x}) - f_i(\boldsymbol{x})\| < \epsilon, \tag{50}$$

*where $\mathrm{dom}(f_1) = X$ and $\mathrm{dom}(f_i) = \mathbb{R}^{d_{i-1}}$ for $i \geq 2$.*

*Then for every $f \in \mathcal{F}$ and every $\epsilon > 0$, there exists $\tilde{f} \in \tilde{\mathcal{F}}_L \circ \cdots \circ \tilde{\mathcal{F}}_1$ (the family of compositions of functions from $\tilde{\mathcal{F}}_i$) such that $\tilde{f}$ uniformly approximates $f$ on $X$.*

*Proof.* This is a restatement of Lim et al. (2022, Lemma 6) in our notation. The key observation is that if each layer $f_i$ in a composition can be uniformly approximated on the image of the preceding layers (which is compact by continuity), then the full composition can be uniformly approximated. The proof proceeds by constructing approximations layer by layer, ensuring that the approximation error at each stage remains bounded on the compact image of the previous layers. $\square$

**Realization of MLP applied pointwise to feature vectors.** Another key technical property that we will use repeatedly in the proofs below is that networks whose affine layers act pointwise on feature channel vectors can realize any MLP applied pointwise to feature channel vectors. This property is essential for simulating architectures like GMN and NG, which use MLPs in their update functions. The ability to realize arbitrary pointwise MLPs allows us to approximate complex update mechanisms using simpler pointwise affine operations combined with nonlinearities. We state this as a general lemma, and then provide specific instantiations for DWS networks and NP–NFN networks.

**Lemma D.14** (Networks with pointwise affine layers can realize pointwise MLP applications on feature channels)**.** *Let $\mathcal{V}^k$ be a weight space with feature dimension $k$, and let $M : \mathbb{R}^k \to \mathbb{R}^{k'}$ be an MLP. Consider a network architecture that can realize the pointwise application of any affine map $A : \mathbb{R}^k \to \mathbb{R}^{k'}$ (i.e., $A(\boldsymbol{x}) = \boldsymbol{x}\boldsymbol{A} + \boldsymbol{u}$) to feature channel vectors:*

$$(\boldsymbol{W}'_\ell)_{i,j,:} = (\boldsymbol{W}_\ell)_{i,j,:}\boldsymbol{A} + \boldsymbol{u}, \qquad (\boldsymbol{b}'_\ell)_{i,:} = (\boldsymbol{b}_\ell)_{i,:}\boldsymbol{A} + \boldsymbol{u}, \tag{51}$$

*for all layers $\ell$ and all valid indices $i, j$. Then such networks can realize the pointwise application of $M$ to feature channel vectors.*

*Proof.* An MLP $M : \mathbb{R}^k \to \mathbb{R}^{k'}$ is a composition of affine maps interleaved with pointwise nonlinearities. Specifically, $M$ can be written as

$$M = A_n \circ \sigma \circ A_{n-1} \circ \cdots \circ \sigma \circ A_1, \tag{52}$$

where each $A_i : \mathbb{R}^{d_{i-1}} \to \mathbb{R}^{d_i}$ is an affine map (i.e., $A_i(\boldsymbol{x}) = \boldsymbol{x}\boldsymbol{A}_i + \boldsymbol{u}_i$ for some matrix $\boldsymbol{A}_i$ and vector $\boldsymbol{u}_i$), $\sigma$ is a pointwise nonlinearity (e.g., ReLU), and $d_0 = k$, $d_n = k'$.

By assumption, the network can realize each affine map $A_i$ applied pointwise to feature channel vectors. By composing these pointwise affine layers corresponding to each $A_i$ in the MLP decomposition, and interleaving them with the same pointwise nonlinearities $\sigma$ used in the MLP, we obtain a network that realizes $M$ applied pointwise to each feature channel vector, as required. $\square$

**Corollary D.15** (DWS networks can realize pointwise MLP applications on feature channels)**.** *DWS networks can realize the pointwise application of any MLP to feature channel vectors.*

*Proof.* For any affine map $A : \mathbb{R}^k \to \mathbb{R}^{k'}$ with $A(\boldsymbol{x}) = \boldsymbol{x}\boldsymbol{A} + \boldsymbol{u}$, the DWS layer $L_{\boldsymbol{A},\boldsymbol{u}} : \mathcal{V}^k \to \mathcal{V}^{k'}$ defined by

$$(\boldsymbol{W}'_\ell)_{i,j,:} = (\boldsymbol{W}_\ell)_{i,j,:}\boldsymbol{A} + \boldsymbol{u}, \qquad (\boldsymbol{b}'_\ell)_{i,:} = (\boldsymbol{b}_\ell)_{i,:}\boldsymbol{A} + \boldsymbol{u}, \tag{53}$$

acts identically on every feature channel vector, is affine, and is equivariant with respect to hidden neuron permutations (cf. (Navon et al., 2023)). Hence, $L_{\boldsymbol{A},\boldsymbol{u}}$ can be realized as a DWS layer. The result follows from Lemma D.14. $\square$

**Corollary D.16** (NP–NFN networks can realize pointwise MLP applications on feature channels)**.** *NP–NFN networks (with positional encoding) can realize the pointwise application of any MLP to feature channel vectors.*

*Proof.* For any affine map $A : \mathbb{R}^k \to \mathbb{R}^{k'}$ with $A(\boldsymbol{x}) = \boldsymbol{x}\boldsymbol{A} + \boldsymbol{u}$, the NP–NFN affine layer $L_{\boldsymbol{A},\boldsymbol{u}} : \mathcal{V}^k \to \mathcal{V}^{k'}$ defined by

$$(\boldsymbol{W}'_\ell)_{i,j,:} = (\boldsymbol{W}_\ell)_{i,j,:}\boldsymbol{A} + \boldsymbol{u}, \qquad (\boldsymbol{b}'_\ell)_{i,:} = (\boldsymbol{b}_\ell)_{i,:}\boldsymbol{A} + \boldsymbol{u}, \tag{54}$$

acts identically on every feature channel vector and is $S$-equivariant (where $S$ is the full neuron-permutation group), since it does not depend on neuron indices. Hence, $L_{\boldsymbol{A},\boldsymbol{u}}$ can be realized as an NP–NFN affine layer. The result follows from Lemma D.14. $\square$

## D.4. Proof of main theorem

*Proof of Theorem D.6.* Let $K \subset \mathcal{V}^{c_{\text{in}}}$ be an arbitrary compact set. We establish the theorem via the following cycle of expressive containments (Definition D.1):

$$\text{DWSNets}_{\text{eq}}^{A} \preceq_{K} \text{NP-NFN+PE}_{\text{eq}}^{A} \qquad \text{(Prop. D.17)} \qquad (55)$$

$$\text{NP-NFN+PE}_{\text{eq}}^{A} \preceq_{K} \text{GMN}_{\text{eq}}^{A} \qquad \text{(Prop. D.18)} \qquad (56)$$

$$\text{GMN}_{\text{eq}}^{A} \preceq_{K} \text{NG}_{\text{eq}}^{A} \qquad \text{(Prop. D.19)} \qquad (57)$$

$$\text{NG}_{\text{eq}}^{A} \preceq_{K} \text{DWSNets}_{\text{eq}}^{A} \qquad \text{(Prop. D.20)} \qquad (58)$$

Since $\preceq_{K}$ is transitive (remark D.2), closing the cycle implies that all four model classes have equivalent expressive power with respect to $K$ (Definition D.3), i.e., $\text{DWSNets}_{\text{eq}}^{A} \simeq_{K} \text{NP-NFN+PE}_{\text{eq}}^{A} \simeq_{K} \text{GMN}_{\text{eq}}^{A} \simeq_{K} \text{NG}_{\text{eq}}^{A}$. The equivalence $\text{DWSNets}_{\text{eq}}^{A} \simeq_{K} \text{HNP-NFN}_{\text{eq}}^{A}$ follows from Remark D.8. Since $K$ was arbitrary, the result holds for all compact sets $K \subseteq \mathcal{V}^{c_{\text{in}}}$. $\qquad \square$

### D.4.1. DWSNETS AND NP–NFN+PE

We begin the cycle by showing that DWS networks can be approximated by NP–NFN+PE networks. The key insight is that NP–NFN+PE networks, which use positional encodings to identify neurons in the first and last layers, can simulate the hidden-neuron permutation equivariant operations of DWS networks. This establishes the first link in our cycle of expressive containments.

**Proposition D.17.** *Let $K \subset \mathcal{V}^{c_{\text{in}}}$ be a compact set. Then $\text{DWSNets}_{\text{eq}}^{A} \preceq_{K} \text{NP-NFN+PE}_{\text{eq}}^{A}$.*

*Proof.* By Lemma D.13 that shows that layer-wise approximation implies network approximation, it suffices to show that any DWS affine layer $L \in \text{Aff}_{G}(\mathcal{V}^{c_{\text{in}}}, \mathcal{V}^{c_{\text{out}}})$ can be approximated by an NP–NFN network applied after the positional encoding PE.

Recall that the weight space with $c_{\text{in}}$ feature channels is:

$$\mathcal{V}^{c_{\text{in}}} := \left( \bigoplus_{\ell=1}^{L} \mathcal{W}_{\ell}^{c_{\text{in}}} \right) \oplus \left( \bigoplus_{\ell=1}^{L} \mathcal{B}_{\ell}^{c_{\text{in}}} \right) \cong \bigoplus_{\ell=1}^{L} \left( \mathbb{R}^{d_{\ell} \times d_{\ell-1} \times c_{\text{in}}} \oplus \mathbb{R}^{d_{\ell} \times c_{\text{in}}} \right). \qquad (59)$$

By the DWS basis blocks characterization (Navon et al., 2023), any DWS affine layer decomposes as a linear combination of basis maps of four types: $\mathcal{W}_{\ell}^{c_{\text{in}}} \to \mathcal{W}_{\ell'}^{c_{\text{out}}}$, $\mathcal{B}_{\ell}^{c_{\text{in}}} \to \mathcal{B}_{\ell'}^{c_{\text{out}}}$, $\mathcal{W}_{\ell}^{c_{\text{in}}} \to \mathcal{B}_{\ell'}^{c_{\text{out}}}$, and $\mathcal{B}_{\ell}^{c_{\text{in}}} \to \mathcal{W}_{\ell'}^{c_{\text{out}}}$ (for $0 \leq \ell, \ell' \leq L$), plus a bias term.

For *interior layers* where $0 < \ell, \ell' < L$, the hidden-neuron permutation group $G = S_{d_1} \times \cdots \times S_{d_{L-1}}$ coincides with the restrictions of the full neuron-permutation group $S := S_{d_0} \times G \times S_{d_L}$ to these layers. Hence DWS affine layers acting only on interior components are automatically $S$-equivariant, and thus are NP–NFN affine layers.

The non-trivial cases are *boundary basis blocks* involving layers $\ell \in \{0, L\}$ or $\ell' \in \{0, L\}$, where $G$-equivariance differs from $S$-equivariance. For these, we exploit the positional encoding features appended by PE (Definition D.9).

We demonstrate the construction for a representative boundary case: the bias-to-bias map $\mathcal{B}_{L}^{c_{\text{in}}} \to \mathcal{B}_{L}^{c_{\text{out}}}$. The remaining boundary cases follow by analogous arguments.

**Representative boundary case: $G$-equivariant bias-to-bias map at the output layer.** By the characterization of (Navon et al., 2023) (Table 6), the most general $G$-equivariant linear map $T_{\boldsymbol{A}} : \mathcal{B}_{L}^{c_{\text{in}}} \to \mathcal{B}_{L}^{c_{\text{out}}}$ is given by:

$$(T_{\boldsymbol{A}}(\boldsymbol{b}_{L}))_{j,k} = \sum_{i=1}^{d_L} \sum_{m=1}^{c_{\text{in}}} \boldsymbol{A}_{i,m,j,k} \cdot (\boldsymbol{b}_{L})_{i,m}, \qquad (60)$$

for an arbitrary tensor $\boldsymbol{A} \in \mathbb{R}^{d_L \times c_{\text{in}} \times d_L \times c_{\text{out}}}$. The tensor $\boldsymbol{A}$ mixes both neuron indices and feature channels.

The positional encoding PE (Definition D.9) augments the output-layer bias with unique identifiers and zero padding:

$$\text{PE}(\boldsymbol{b}_L) = \boldsymbol{b}_L^{(0)}, \qquad (\boldsymbol{b}_L^{(0)})_{i,:} \coloneqq \big[(\boldsymbol{b}_L)_{i,:}, \ \boldsymbol{0}_{|\mathcal{T}_{\text{PE}}|}, \ \boldsymbol{e}_{\text{out}_i}\big] \in \mathbb{R}^{c_{\text{in}}+2|\mathcal{T}_{\text{PE}}|}, \tag{61}$$

where $\boldsymbol{e}_{\text{out}_i} \in \mathbb{R}^{|\mathcal{T}_{\text{PE}}|}$ is the one-hot encoding of output neuron $i$, and the zero padding ensures matching feature dimensions with weights.

Throughout this construction, we will define the maps only on $\mathcal{B}_L$ and the weights at all layers, biases at layers $\ell < L$ are set to zero by the NP–NFN layers, as we are only approximating the $\mathcal{B}_L \to \mathcal{B}_L$ block.

Define the layer $L_1 : \mathcal{V}^{c_{\text{in}}+2|\mathcal{T}_{\text{PE}}|}|_{\mathcal{B}_L} \to \mathcal{V}^{c_{\text{in}}+d_L \cdot c_{\text{out}}+|\mathcal{T}_{\text{PE}}|}|_{\mathcal{B}_L}$ by:

$$(\boldsymbol{b}_L^{(1)})_{i,:} \coloneqq \big[(\boldsymbol{b}_L)_{i,:}, \ \text{vec}(\boldsymbol{A}_{i,:,:,:}), \ \boldsymbol{e}_{\text{out}_i}\big] \in \mathbb{R}^{c_{\text{in}}+d_L \cdot c_{\text{out}}+|\mathcal{T}_{\text{PE}}|} \tag{62}$$

where $\boldsymbol{A}_{i,:,:,:} \in \mathbb{R}^{c_{\text{in}} \times d_L \times c_{\text{out}}}$ is the slice of $\boldsymbol{A}$ corresponding to input neuron $i$, and $\text{vec}(\cdot)$ denotes flattening to a vector of dimension $c_{\text{in}} \cdot d_L \cdot c_{\text{out}}$. Note that the zero padding from the positional encoding is dropped as it serves no purpose in this construction.

We verify $S$-equivariance of each component:

(i) The term $(\boldsymbol{b}_L)_{i,:}$ is the identity map, which is $S$-equivariant.
(ii) The slice $\boldsymbol{A}_{i,:,:,:}$ is extracted using the one-hot $\boldsymbol{e}_{\text{out}_i}$. Formally, $\text{vec}(\boldsymbol{A}_{i,:,:,:}) = (\boldsymbol{e}_{\text{out}_i})^\top \boldsymbol{A}^{\text{reshape}}$ where $\boldsymbol{A}^{\text{reshape}} \in \mathbb{R}^{d_L \times (c_{\text{in}} \cdot d_L \cdot c_{\text{out}})}$ is a reshaped view of $\boldsymbol{A}$. Since $\boldsymbol{A}$ is a constant (affine bias), this is an $S$-equivariant affine map.
(iii) The one-hot $\boldsymbol{e}_{\text{out}_i}$ is preserved from the positional encoding, an $S$-equivariant operation.

Since $L_1$ is a concatenation of $S$-equivariant linear maps with affine bias terms, we have $L_1 \in \text{Aff}_S(\mathcal{V}^{c_{\text{in}}+2|\mathcal{T}_{\text{PE}}|}|_{\mathcal{B}_L}, \mathcal{V}^{c_{\text{in}}+d_L \cdot c_{\text{out}}+|\mathcal{T}_{\text{PE}}|}|_{\mathcal{B}_L})$, i.e., $L_1$ is an NP–NFN affine layer.

Define the continuous function $\psi : \mathbb{R}^{c_{\text{in}}+d_L \cdot c_{\text{out}}+|\mathcal{T}_{\text{PE}}|} \to \mathbb{R}^{d_L \cdot c_{\text{out}}+|\mathcal{T}_{\text{PE}}|}$ by:

$$\psi\big((\boldsymbol{b}_L)_{i,:}, \text{vec}(\boldsymbol{A}_{i,:,:,:}), \boldsymbol{e}_{\text{out}_i}\big) = \Big[\boldsymbol{e}_{\text{out}_i}, \ \text{vec}\Big(\sum_{m=1}^{c_{\text{in}}} (\boldsymbol{b}_L)_{i,m} \cdot \boldsymbol{A}_{i,m,:,:}\Big)\Big], \tag{63}$$

Let $K^{(1)} \coloneqq L_1(\text{PE}(K|_{\mathcal{B}_L})) \subset \mathbb{R}^{d_L \times (c_{\text{in}}+d_L \cdot c_{\text{out}}+|\mathcal{T}_{\text{PE}}|)}$ be the compact set of intermediate representations. Since $K$ is compact, PE and $L_1$ are continuous, and the image of a compact set under a continuous map is compact, $K^{(1)}$ is compact.

By the universal approximation theorem for MLPs (Cybenko, 1989; Hornik, 1991), for any $\epsilon > 0$, there exists an MLP $\Psi : \mathbb{R}^{c_{\text{in}}+d_L \cdot c_{\text{out}}+|\mathcal{T}_{\text{PE}}|} \to \mathbb{R}^{d_L \cdot c_{\text{out}}+|\mathcal{T}_{\text{PE}}|}$ such that:

$$\sup_{\boldsymbol{x} \in K_{\text{flat}}^{(1)}} \|\Psi(\boldsymbol{x}) - \psi(\boldsymbol{x})\| < \epsilon/4, \tag{64}$$

where $K_{\text{flat}}^{(1)} \subset \mathbb{R}^{c_{\text{in}}+d_L \cdot c_{\text{out}}+|\mathcal{T}_{\text{PE}}|}$ is the compact set obtained by collecting all row vectors of matrices in $K^{(1)}$.

By Corollary D.16, NP–NFN networks can realize the pointwise application of the MLP $\Psi$ to feature channel vectors. Define $M_2 : \mathcal{V}^{c_{\text{in}}+d_L \cdot c_{\text{out}}+|\mathcal{T}_{\text{PE}}|}|_{\mathcal{B}_L} \to \mathcal{V}^{d_L \cdot c_{\text{out}}+|\mathcal{T}_{\text{PE}}|}|_{\mathcal{B}_L}$ by applying $\Psi$ pointwise:

$$(\boldsymbol{b}_L^{(2)})_{i,:} \coloneqq \Psi\big((\boldsymbol{b}_L^{(1)})_{i,:}\big) \approx \Big[\boldsymbol{e}_{\text{out}_i}, \ \text{vec}\Big(\sum_{m=1}^{c_{\text{in}}} (\boldsymbol{b}_L)_{i,m} \cdot \boldsymbol{A}_{i,m,:,:}\Big)\Big] \in \mathbb{R}^{d_L \cdot c_{\text{out}}+|\mathcal{T}_{\text{PE}}|}. \tag{65}$$

$M_2$ is $S$-equivariant because it applies the same MLP $\Psi$ to each neuron's feature vector independently, and can be realized by NP–NFN networks via Corollary D.16.

Define the layer $L_3 : \mathcal{V}^{d_L \cdot c_{\text{out}}+|\mathcal{T}_{\text{PE}}|}|_{\mathcal{B}_L} \to \mathcal{V}^{d_L \cdot c_{\text{out}}+|\mathcal{T}_{\text{PE}}|}|_{\mathcal{B}_L}$ by:

$$(\boldsymbol{b}_L^{(3)})_{j,:} \coloneqq \Big[\boldsymbol{e}_{\text{out}_j}, \ \sum_{i=1}^{d_L} (\boldsymbol{b}_L^{(2)})_{i,|\mathcal{T}_{\text{PE}}|+1:}\Big] \in \mathbb{R}^{d_L \cdot c_{\text{out}}+|\mathcal{T}_{\text{PE}}|}. \tag{66}$$

This layer consists of two operations, the first $|\mathcal{T}_{\mathrm{PE}}|$ channels preserve $\boldsymbol{e}_{\mathrm{out}_j}$ and the last $d_L \cdot c_{\mathrm{out}}$ channels compute the sum over all last layer neurons and broadcast to each last layer neuron. This is an $S$-equivariant operation.

Both operations are $S$-equivariant linear maps, so $L_3 \in \mathrm{Hom}_S(\mathcal{V}^{d_L \cdot c_{\mathrm{out}}+|\mathcal{T}_{\mathrm{PE}}|}|_{\mathcal{B}_L}, \mathcal{V}^{d_L \cdot c_{\mathrm{out}}+|\mathcal{T}_{\mathrm{PE}}|}|_{\mathcal{B}_L})$, i.e., $L_3$ is an NP–NFN linear layer.

The intermediate representation satisfies:

$$(\boldsymbol{b}_L^{(3)})_{j,:} \approx \left[\boldsymbol{e}_{\mathrm{out}_j}, \ \mathrm{vec}\Big(\sum_{i=1}^{d_L}\sum_{m=1}^{c_{\mathrm{in}}}(\boldsymbol{b}_L)_{i,m}\cdot\boldsymbol{A}_{i,m,:,:}\Big)\right] = \left[\boldsymbol{e}_{\mathrm{out}_j}, \ \mathrm{vec}(T_{\boldsymbol{A}}(\boldsymbol{b}_L))\right] \in \mathbb{R}^{d_L\cdot c_{\mathrm{out}}+|\mathcal{T}_{\mathrm{PE}}|}. \tag{67}$$

Define the continuous function $\phi : \mathbb{R}^{d_L\cdot c_{\mathrm{out}}+|\mathcal{T}_{\mathrm{PE}}|} \to \mathbb{R}^{c_{\mathrm{out}}}$ by:

$$\phi\big(\boldsymbol{e}_{\mathrm{out}_j}, \mathrm{vec}(T_{\boldsymbol{A}}(\boldsymbol{b}_L))\big) = (T_{\boldsymbol{A}}(\boldsymbol{b}_L))_{j,:} \in \mathbb{R}^{c_{\mathrm{out}}}, \tag{68}$$

which extracts the $j$-th row of the output matrix using the one-hot identifier $\boldsymbol{e}_{\mathrm{out}_j}$.

Let $K^{(3)} := L_3(M_2(L_1(\mathrm{PE}(K|_{\mathcal{B}_L}))))$ be the compact set of intermediate representations at this stage. Again, $K^{(3)}$ is compact as the image of the compact set $K$ under continuous maps.

By the universal approximation theorem for MLPs (Cybenko, 1989; Hornik, 1991), for any $\epsilon > 0$, there exists an MLP $\rho : \mathbb{R}^{d_L\cdot c_{\mathrm{out}}+|\mathcal{T}_{\mathrm{PE}}|} \to \mathbb{R}^{c_{\mathrm{out}}}$ such that:

$$\sup_{\boldsymbol{x}\in K_{\mathrm{flat}}^{(3)}} \|\rho(\boldsymbol{x}) - \phi(\boldsymbol{x})\| < \epsilon/4, \tag{69}$$

where $K_{\mathrm{flat}}^{(3)}$ is the compact set of row vectors from $K^{(3)}$.

By Corollary D.16, NP–NFN networks can realize the pointwise application of the MLP $\rho$ to feature channel vectors. Define $M_4 : \mathcal{V}^{d_L\cdot c_{\mathrm{out}}+|\mathcal{T}_{\mathrm{PE}}|}|_{\mathcal{B}_L} \to \mathcal{V}^{c_{\mathrm{out}}}|_{\mathcal{B}_L}$ by:

$$(\boldsymbol{b}_L')_{j,:} := \rho\big((\boldsymbol{b}_L^{(3)})_{j,:}\big) \approx (T_{\boldsymbol{A}}(\boldsymbol{b}_L))_{j,:} \in \mathbb{R}^{c_{\mathrm{out}}}. \tag{70}$$

As in step (b), $M_4$ is $S$-equivariant because it applies the same MLP $\rho$ pointwise, and can be realized by NP–NFN networks via Corollary D.16.

The uniform approximation of the composition $M_4 \circ L_3 \circ M_2 \circ L_1$ on $K|_{\mathcal{B}_L}$ follows from Lemma D.13. Concretely, since $L_1$, $L_3$ are exact NP–NFN layers and $M_2$, $M_4$ approximate continuous functions on the compact sets $K_{\mathrm{flat}}^{(1)}$ and $K_{\mathrm{flat}}^{(3)}$ respectively, we obtain:

$$\sup_{\boldsymbol{v}\in K} \|M_4 \circ L_3 \circ M_2 \circ L_1(\mathrm{PE}(\boldsymbol{v})) - T_{\boldsymbol{A}}(\boldsymbol{v}|_{\mathcal{B}_L})\| < \epsilon. \tag{71}$$

The remaining boundary basis blocks are handled analogously. $\qquad\square$

### D.4.2. NP–NFN+PE AND GMN

Next, we show that NP–NFN+PE networks can be approximated by GMN networks. This step connects the functional network perspective to the graph-based message passing framework. The proof leverages the fact that GMN's graph structure can encode the positional information used by NP–NFN+PE, and that GMN's message passing operations can simulate the affine equivariant layers of NP–NFN+PE.

**Proposition D.18.** *Let $K \subset \mathcal{V}^{c_{\mathrm{in}}}$ be a compact set. Then* $\mathrm{NP\text{-}NFN+PE}_{\mathrm{eq}}^A \preceq_K \mathrm{GMN}_{\mathrm{eq}}^A$.

*Proof.* By Lim et al. (2023, Proposition 10), GMN on MLP parameter graphs can express any NP–NFN layer. It therefore suffices to show that the positional encoding PE can be realized by a single GMN layer. Then we can use lemma D.13 to conclude that $\mathrm{NP\text{-}NFN+PE}_{\mathrm{eq}}^A \preceq_K \mathrm{GMN}_{\mathrm{eq}}^A$.

Recall from Definition D.9 that $\mathrm{PE} : \mathcal{V}^c \to \mathcal{V}^{c+2|\mathcal{T}_{\mathrm{PE}}|}$ appends neuron type identifiers to each weight and bias entry, where $\mathcal{T}_{\mathrm{PE}} = \{\mathtt{in}_1, \ldots, \mathtt{in}_{d_0}\} \cup \{\mathtt{out}_1, \ldots, \mathtt{out}_{d_L}\} \cup \{\mathtt{hidden}\}$. We will show this encoding can be realized by a single GMN layer using the ntype feature of GMN nodes.

By Definition D.10, the GMN node features are initialized as:

$$\boldsymbol{h}_v := \big[\mathrm{nlayer}(v),\ \mathrm{ntype}(v)\big] \in \mathbb{R}^{2L+d_0+d_L+2},$$

where $\mathrm{ntype}(v)$ encodes unique identities for input nodes ($\boldsymbol{e}_{\mathrm{in}_j}$ for $\nu_j^{(0)}$), output nodes ($\boldsymbol{e}_{\mathrm{out}_i}$ for $\nu_i^{(L)}$), hidden nodes (shared $\boldsymbol{e}_{\mathrm{hidden}}$), and bias nodes ($\boldsymbol{e}_{\mathrm{bias}_\ell}$).

The GMN edge features are initialized as:

$$\boldsymbol{e}_e := \big[p(e),\ \mathrm{layer}(e),\ \mathrm{dir}(e),\ \mathrm{ptype}(e)\big] \in \mathbb{R}^{c_{\mathrm{in}}+L+4},$$

where $p(e) \in \mathbb{R}^{c_{\mathrm{in}}}$ is the parameter value (weight or bias), $\mathrm{layer}(e)$ is the one-hot layer encoding, $\mathrm{dir}(e)$ indicates forward/backward direction, and $\mathrm{ptype}(e)$ distinguishes weight edges from bias edges.

The GMN edge update equation 29 allows propagating endpoint node features to edges:

$$\boldsymbol{e}'_{uv} = \phi_e\big(\boldsymbol{h}_u, \boldsymbol{h}_v, \boldsymbol{e}_{uv}, \boldsymbol{u}\big).$$

Define a GMN layer $L_{\mathrm{PE}} = (\phi_m, \phi_h, \phi_e, \phi_u)$ with $\phi_m, \phi_h, \phi_u$ as identities and

$$\phi_e : (\boldsymbol{h}_u, \boldsymbol{h}_v, \boldsymbol{e}_{uv}, \boldsymbol{u}) \mapsto \big[p(e),\ \mathrm{extract}_{\mathrm{src}}(\boldsymbol{h}_u, \boldsymbol{h}_v, \boldsymbol{e}_{uv}),\ \mathrm{extract}_{\mathrm{tgt}}(\boldsymbol{h}_u, \boldsymbol{h}_v, \boldsymbol{e}_{uv}),\ \mathrm{layer}(e),\ \mathrm{dir}(e),\ \mathrm{ptype}(e)\big],$$

$$\mathrm{extract}_{\mathrm{src}}(\boldsymbol{h}_u, \boldsymbol{h}_v, \boldsymbol{e}_{uv}) := \mathrm{ptype}(e)_1 \cdot \big(\mathrm{dir}(e)_1 \cdot \Pi_{\mathrm{GMN}\to\mathrm{PE}}(\mathrm{ntype}(u)) + \mathrm{dir}(e)_2 \cdot \Pi_{\mathrm{GMN}\to\mathrm{PE}}(\mathrm{ntype}(v))\big),$$
$$\mathrm{extract}_{\mathrm{tgt}}(\boldsymbol{h}_u, \boldsymbol{h}_v, \boldsymbol{e}_{uv}) := \mathrm{dir}_1 \cdot \Pi_{\mathrm{GMN}\to\mathrm{PE}}(\mathrm{ntype}(v)) + \mathrm{dir}_2 \cdot \Pi_{\mathrm{GMN}\to\mathrm{PE}}(\mathrm{ntype}(u)),$$

where $\mathrm{dir}(e)_1 = 1$ for forward edges and $\mathrm{dir}(e)_2 = 1$ for backward edges, and $\mathrm{ptype}(e)_1 = 1$ for weight edges (outputting the source neuron type) and $\mathrm{ptype}(e)_1 = 0$ for bias edges (outputting zeros).

The key step is the projection map $\Pi_{\mathrm{GMN}\to\mathrm{PE}} : \mathbb{R}^{|\mathcal{T}_{\mathrm{GMN}}|} \to \mathbb{R}^{|\mathcal{T}_{\mathrm{PE}}|}$ that maps GMN node types to PE neuron types:

$$\Pi_{\mathrm{GMN}\to\mathrm{PE}}(\mathrm{ntype}(v)) := \begin{cases} \boldsymbol{e}_{\mathrm{in}_j}^{(|\mathcal{T}_{\mathrm{PE}}|)} & \text{if } \mathrm{ntype}(v) = \boldsymbol{e}_{\mathrm{in}_j}^{(|\mathcal{T}_{\mathrm{GMN}}|)}, \\ \boldsymbol{e}_{\mathrm{out}_i}^{(|\mathcal{T}_{\mathrm{PE}}|)} & \text{if } \mathrm{ntype}(v) = \boldsymbol{e}_{\mathrm{out}_i}^{(|\mathcal{T}_{\mathrm{GMN}}|)}, \\ \boldsymbol{e}_{\mathrm{hidden}}^{(|\mathcal{T}_{\mathrm{PE}}|)} & \text{if } \mathrm{ntype}(v) = \boldsymbol{e}_{\mathrm{hidden}}^{(|\mathcal{T}_{\mathrm{GMN}}|)}, \\ \boldsymbol{0}_{|\mathcal{T}_{\mathrm{PE}}|} & \text{if } \mathrm{ntype}(v) = \boldsymbol{e}_{\mathrm{bias}_\ell}^{(|\mathcal{T}_{\mathrm{GMN}}|)} \text{ for any } \ell. \end{cases} \tag{72}$$

Since GMN's $\mathcal{T}_{\mathrm{GMN}}$ contains all types in PE's $\mathcal{T}_{\mathrm{PE}}$ plus additional bias node types, this projection preserves the relevant neuron type information while discarding the bias node types (which are not used in PE). The projection $\Pi_{\mathrm{GMN}\to\mathrm{PE}}$ can be implemented as a linear map (matrix multiplication) and is therefore realizable by GMN's MLP components.

Since $\Pi_{\mathrm{GMN}\to\mathrm{PE}}$ is a linear map and $\mathrm{extract}_{\mathrm{src}}, \mathrm{extract}_{\mathrm{tgt}}$ are coordinatewise polynomial functions of $(\boldsymbol{h}_u, \boldsymbol{h}_v, \boldsymbol{e}_{uv})$ (formed by linear projection followed by additions and products of coordinates with one-hot indicators), it follows that the map

$$(\boldsymbol{h}_u, \boldsymbol{h}_v, \boldsymbol{e}_{uv}, \boldsymbol{u}) \longmapsto \phi_e(\boldsymbol{h}_u, \boldsymbol{h}_v, \boldsymbol{e}_{uv}, \boldsymbol{u})$$

is a polynomial map, hence continuous.

Recall $K \subset \mathcal{V}^{c_{\mathrm{in}}}$ is compact, and let

$$\Gamma : \mathcal{V}^{c_{\mathrm{in}}} \to \mathbb{R}^{d_h} \times \mathbb{R}^{d_h} \times \mathbb{R}^{d_e} \times \mathbb{R}^{d_u}$$

denote the fixed continuous initialization map that sends $\boldsymbol{v} \in \mathcal{V}^{c_{\mathrm{in}}}$ to the corresponding tuple $(\boldsymbol{h}_u, \boldsymbol{h}_v, \boldsymbol{e}_{uv}, \boldsymbol{u})$ for an edge $(u, v)$. Then $\Gamma(K)$ is compact.

By the universal approximation theorem, for every $\varepsilon > 0$ there exists an MLP $\widehat{\phi}_e$ such that

$$\sup_{z \in \Gamma(K)} \big\|\widehat{\phi}_e(z) - \phi_e(z)\big\| < \varepsilon.$$

In particular, on inputs arising from $\boldsymbol{v} \in K$, the edge-update MLP $\widehat{\phi}_e$ uniformly approximates the polynomial rule $\phi_e$.

After applying $L_{\mathrm{PE}}$, both forward and backward edge features become:

$$\boldsymbol{e}'_e \approx \begin{cases} \left[ \mathrm{PE}(\boldsymbol{W}_\ell)_{i,j,:}, \ \mathrm{layer}(e), \ \mathrm{dir}(e), \ \mathrm{ptype}(e) \right] & e = (\nu_i^{(\ell-1)}, \nu_j^{(\ell)}), \\ \left[ \mathrm{PE}(\boldsymbol{b}_\ell)_{j,:}, \ \mathrm{layer}(e), \ \mathrm{dir}(e), \ \mathrm{ptype}(e) \right] & e = (\beta_i^{(\ell)}, \nu_j^{(\ell)}), \end{cases}$$

Thus, the MLP parameters augmented with the projected positional encodings (mapped from $\mathcal{T}_{\mathrm{GMN}}$ to $\mathcal{T}_{\mathrm{PE}}$ space) are now stored in the GMN edge features (on both forward and backward edges), which is precisely the input format for NP–NFN layers operating on $\mathcal{V}^{c_{\mathrm{in}}+2|\mathcal{T}_{\mathrm{PE}}|}$. Therefore we can continue the simulation same as Lim et al. (2023, Proposition 10) and finish the proof. □

### D.4.3. GMN AND NG-GNN

NG-GNN is a simplified variant of GMN that stores biases as node features rather than using separate bias nodes and does not allow global features. However, NG-GNN's simpler structure is sufficient to simulate GMN's more complex update mechanisms.

**Proposition D.19.** *Let $K \subset \mathcal{V}^{c_{\mathrm{in}}}$ be a compact set. Then $\mathrm{GMN}^A_{\mathrm{eq}} \preceq_K \mathrm{NG}^A_{\mathrm{eq}}$.*

*Proof.* To establish that $\mathrm{GMN}^A_{\mathrm{eq}} \preceq_K \mathrm{NG}^A_{\mathrm{eq}}$, we must show that any GMN layer can be simulated by a composition of NG layers.

**Setup and notation.** We are given a single GMN layer $L^{\mathrm{GMN}} = (\phi_m^{\mathrm{GMN}}, \phi_h^{\mathrm{GMN}}, \phi_e^{\mathrm{GMN}}, \phi_u^{\mathrm{GMN}})$ that updates the state $(\boldsymbol{h}^{\mathrm{GMN}}, \boldsymbol{e}^{\mathrm{GMN}}, \boldsymbol{u}^{\mathrm{GMN}}) \mapsto (\boldsymbol{h}'^{\mathrm{GMN}}, \boldsymbol{e}'^{\mathrm{GMN}}, \boldsymbol{u}'^{\mathrm{GMN}})$ according to equations equation 27–equation 30. Here:

- $\boldsymbol{h}_v^{\mathrm{GMN}} \in \mathbb{R}^{d_h}$ denotes the GMN node feature for node $v$ (either a neuron node $\nu_i^{(\ell)}$ or a bias node $\beta^{(\ell)}$),
- $\boldsymbol{e}_e^{\mathrm{GMN}} \in \mathbb{R}^{d_e}$ denotes the GMN edge feature for edge $e$,
- $\boldsymbol{u}^{\mathrm{GMN}} \in \mathbb{R}^{d_u}$ denotes the GMN global feature vector.

Our goal is to construct a composition of NG layers $L_n \circ \cdots \circ L_1$ that simulates this GMN layer. Each NG layer $L_i = (\phi_{m,i}, \phi_{h,i}, \phi_{e,i})$ updates the state $(\boldsymbol{h}^{(i)}, \boldsymbol{e}^{(i)}) \mapsto (\boldsymbol{h}^{(i+1)}, \boldsymbol{e}^{(i+1)})$ according to equations equation 32–equation 34, where:

- $\boldsymbol{h}_{\nu_j^{(\ell)}}^{(i)} \in \mathbb{R}^{d_h^{(i)}}$ denotes the NG node feature for neuron node $\nu_j^{(\ell)}$ at step $i$,
- $\boldsymbol{e}_e^{(i)} \in \mathbb{R}^{d_e^{(i)}}$ denotes the NG edge feature for edge $e$ at step $i$.

Note that NG has no global features and no explicit bias nodes; we will simulate these by storing the corresponding GMN features within NG node and edge features.

**Key differences and simulation strategy.** The key differences between GMN and NG are:

- GMN maintains a global feature vector $\boldsymbol{u}^{\mathrm{GMN}} \in \mathbb{R}^{d_u}$, while NG has no global state. We simulate this by maintaining a copy of $\boldsymbol{u}^{\mathrm{GMN}}$ in each NG node feature.
- GMN uses explicit bias nodes $\beta^{(\ell)}$ per layer with bias parameters stored on edges, while NG stores bias parameters directly in neuron node features. We simulate bias nodes by storing their features within the corresponding neuron node features.
- GMN edge features include a parameter type indicator $\mathrm{ptype}(e)$, while NG edges are all of the same type (weights only). We add this indicator to NG edge features during initialization.

Our proof strategy is to simulate a single GMN layer using a constant-depth composition of NG layers. We proceed in three main phases:

**Phase 1: Feature initialization**: Embed GMN node, edge, and global features into NG node and edge features.
**Phase 2: Edge and node updates**: Simulate GMN message computation, edge updates, and node updates using NG layers.
**Phase 3: Global feature update**: Compute and propagate the GMN global feature update using NG message passing.

**Phase 1: Feature initialization.** We begin by encoding the GMN state into NG features. The initial NG node and edge

features are:

$$\boldsymbol{h}_{\nu_i^{(\ell)}}^{(0)} := \left[\mathrm{nlayer}(\nu_i^{(\ell)}),\ \mathrm{ntype}(\nu_i^{(\ell)}),\ (\boldsymbol{b}_\ell)_{i,:}\right] \in \mathbb{R}^{d_h}, \quad \text{for } 0 \le \ell \le L,\ i \in [d_\ell], \tag{73}$$

$$\boldsymbol{e}_e^{(0)} := \left[p(e),\ \mathrm{layer}(e),\ \mathrm{dir}(e)\right] \in \mathbb{R}^{d_e}, \quad \text{for } e \in E_{\mathrm{w}}, \tag{74}$$

where $(\boldsymbol{b}_0)_{i,:} := \boldsymbol{0}_{c_{\mathrm{in}}}$ for the input layer. We assume the initial GMN global feature is $\boldsymbol{u}^{\mathrm{GMN}} = \boldsymbol{0}_{d_u}$.

**Step 1.1: Augmenting features with GMN placeholders.** The first NG layer $L_1$ augments the node features to include placeholders for GMN bias node features and the global feature. This layer prepares the feature space to accommodate GMN's additional structure (bias nodes and global features) that NG does not natively support. For each neuron node $\nu_i^{(\ell)}$ with $\ell \ge 1$, we append:

- A placeholder for the bias node feature $\boldsymbol{h}_{\beta^{(\ell)}}^{\mathrm{GMN}}$,
- A placeholder for bias edge features (forward and backward),
- The parameter type indicator $\mathrm{ptype}$,
- The global feature $\boldsymbol{u}^{\mathrm{GMN}}$ (initialized to zero).

For the input layer ($\ell = 0$), we append zero vectors of appropriate dimensions.

Formally, we define the first NG layer $L_1 = (\phi_{m,1}, \phi_{h,1}, \phi_{e,1})$ as follows. The message function is identically zero: $\phi_{m,1}(\boldsymbol{h}_u, \boldsymbol{h}_v, \boldsymbol{e}_e) := \boldsymbol{0}$ (no message passing needed at this stage). The node update function $\phi_{h,1}$ augments features:

$$\boldsymbol{h}_{\nu_i^{(\ell)}}^{(1)} := \phi_{h,1}\Big(\boldsymbol{h}_{\nu_i^{(\ell)}}^{(0)},\ \sum_{j \in \mathcal{N}(i)} \boldsymbol{0}\Big) = \begin{cases} \left[\boldsymbol{h}_{\nu_i^{(\ell)}}^{(0)},\ \boldsymbol{e}_\ell^{(L)},\ \boldsymbol{e}_2^{(2)},\ \boldsymbol{0}_{d_u}\right] & \text{if } 1 \le \ell \le L, \\ \left[\boldsymbol{h}_{\nu_i^{(0)}}^{(0)},\ \boldsymbol{0}_L,\ \boldsymbol{0}_2,\ \boldsymbol{0}_{d_u}\right] & \text{if } \ell = 0, \end{cases} \tag{75}$$

where $\boldsymbol{e}_\ell^{(L)}$ is the one-hot layer encoding for layer $\ell$ (effectively identifying which bias node type would be associated with this layer), and $\boldsymbol{e}_2^{(2)}$ indicates bias parameter type. Note that since NG nodes don't natively support bias node types from $\mathcal{T}_{\mathrm{GMN}}$, we use layer identification instead to distinguish bias features from different layers, which provides sufficient information for GMN simulation. The edge update function $\phi_{e,1}$ adds the parameter type indicator to distinguish weight edges from bias edges:

$$\boldsymbol{e}_e^{(1)} := \phi_{e,1}(\boldsymbol{h}_u^{(0)}, \boldsymbol{h}_v^{(0)}, \boldsymbol{e}_e^{(0)}) = \left[(\boldsymbol{W}_\ell)_{i,j,:},\ \mathrm{layer}(e),\ \mathrm{dir}(e),\ \boldsymbol{e}_1^{(2)}\right], \tag{76}$$

where $\boldsymbol{e}_1^{(2)}$ indicates weight parameter type. Note that all operations are defined for both forward and backward edges; the $\mathrm{dir}(e)$ feature allows the message and edge update functions to distinguish between directions.

**Step 1.2: Reorganizing features for GMN simulation.** The second NG layer $L_2$ reorganizes features to prepare for GMN simulation. This layer duplicates and reorders components so that each neuron node $\nu_j^{(\ell)}$ stores all necessary GMN information in a structured format:

- Its own GMN node feature $\boldsymbol{h}_{\nu_j^{(\ell)}}^{\mathrm{GMN}}$,
- The GMN bias node feature $\boldsymbol{h}_{\beta^{(\ell)}}^{\mathrm{GMN}}$ (to be computed),
- The GMN bias edge features $\boldsymbol{e}_{(\beta^{(\ell)}, \nu_j^{(\ell)})}^{\mathrm{GMN}}$ and $\boldsymbol{e}_{(\nu_j^{(\ell)}, \beta^{(\ell)})}^{\mathrm{GMN}}$ (to be computed),
- The global feature $\boldsymbol{u}^{\mathrm{GMN}}$,
- The layer width $d_{\ell-1}$ (used later for message aggregation).

Formally:

$$\boldsymbol{h}_{\nu_j^{(\ell)}}^{(2)} := \left[\boldsymbol{h}_{\nu_j^{(\ell)}}^{\mathrm{GMN}},\ \boldsymbol{h}_{\beta^{(\ell)}}^{\mathrm{GMN}},\ \boldsymbol{e}_{(\beta^{(\ell)}, \nu_j^{(\ell)})}^{\mathrm{GMN}},\ \boldsymbol{e}_{(\nu_j^{(\ell)}, \beta^{(\ell)})}^{\mathrm{GMN}},\ \boldsymbol{u}^{\mathrm{GMN}},\ d_{\ell-1}\right], \tag{77}$$

$$\boldsymbol{e}_e^{(2)} := \left[\boldsymbol{e}_e^{\mathrm{GMN}},\ \boldsymbol{u}^{\mathrm{GMN}}\right], \tag{78}$$

where $d_{\ell-1}$ is extracted from the layer encoding in the node features. Note that at this stage, the bias node and edge features are placeholders that will be computed in subsequent steps.

**Phase 2: Edge and node updates.** We now simulate the GMN edge and node update functions using NG layers. This phase computes the updated GMN features by applying the GMN update functions pointwise on the stored features.

**Step 2.1: Weight edge updates.** The third NG layer $L_3$ computes the GMN weight edge update. This layer applies the GMN edge update function $\phi_e^{\mathrm{GMN}}$ to each weight edge. For each weight edge $e = (\nu_i^{(\ell-1)}, \nu_j^{(\ell)})$, the GMN edge update function takes as input the incident node features, the edge feature, and the global feature. Since all these are stored in $\boldsymbol{h}^{(2)}$ and $\boldsymbol{e}^{(2)}$, we can apply $\phi_e^{\mathrm{GMN}}$ pointwise:

$$
\begin{aligned}
\boldsymbol{e}_e^{(3)} &:= \phi_{e,3}(\boldsymbol{h}_{\nu_i^{(\ell-1)}}^{(2)}, \boldsymbol{h}_{\nu_j^{(\ell)}}^{(2)}, \boldsymbol{e}_e^{(2)}) \\
&= \big[\boldsymbol{e}_e^{\mathrm{GMN}}, \ \phi_e^{\mathrm{GMN}}(\boldsymbol{h}_{\nu_i^{(\ell-1)}}^{\mathrm{GMN}}, \boldsymbol{h}_{\nu_j^{(\ell)}}^{\mathrm{GMN}}, \boldsymbol{e}_e^{\mathrm{GMN}}, \boldsymbol{u}^{\mathrm{GMN}}), \ \boldsymbol{u}^{\mathrm{GMN}}\big] \\
&= \big[\boldsymbol{e}_e^{\mathrm{GMN}}, \ \boldsymbol{e}'^{\mathrm{GMN}}_e, \ \boldsymbol{u}^{\mathrm{GMN}}\big],
\end{aligned}
\tag{79}
$$

where $\boldsymbol{e}'^{\mathrm{GMN}}_e$ denotes the updated GMN edge feature. This layer preserves the original edge feature and appends the updated feature, allowing us to maintain both for subsequent computations.

**Step 2.2: Bias edge updates.** The fourth NG layer $L_4$ computes the GMN bias edge updates. The GMN bias edges connect bias nodes $\beta^{(\ell)}$ to neuron nodes $\nu_j^{(\ell)}$. Since NG has no explicit bias nodes, we simulate the bias edge update by storing the bias edge features in the neuron node features. This layer computes the updated bias edge features by applying the GMN edge update function to the stored bias node and neuron node features. We set the message function to zero: $\phi_{m,4}(\boldsymbol{h}_u, \boldsymbol{h}_v, \boldsymbol{e}_e) := \boldsymbol{0}$ (no message passing needed). The node update function $\phi_{h,4}$ computes the bias edge updates from the stored information:

$$
\begin{aligned}
\boldsymbol{h}_{\nu_j^{(\ell)}}^{(3)} &:= \phi_{h,4}\Big(\boldsymbol{h}_{\nu_j^{(\ell)}}^{(2)}, \sum_{i \in \mathcal{N}(j)} \boldsymbol{0}\Big) \\
&= \big[\boldsymbol{h}_{\nu_j^{(\ell)}}^{\mathrm{GMN}}, \ \boldsymbol{h}_{\beta^{(\ell)}}^{\mathrm{GMN}}, \ \boldsymbol{e}_{(\beta^{(\ell)},\nu_j^{(\ell)})}^{\mathrm{GMN}}, \ \boldsymbol{e}'^{\mathrm{GMN}}_{(\beta^{(\ell)},\nu_j^{(\ell)})}, \\
&\qquad \boldsymbol{e}_{(\nu_j^{(\ell)},\beta^{(\ell)})}^{\mathrm{GMN}}, \ \boldsymbol{e}'^{\mathrm{GMN}}_{(\nu_j^{(\ell)},\beta^{(\ell)})}, \ \boldsymbol{u}^{\mathrm{GMN}}, \ d_{\ell-1}\big],
\end{aligned}
\tag{80}
$$

where $\boldsymbol{e}'^{\mathrm{GMN}}_{(\beta^{(\ell)},\nu_j^{(\ell)})}$ and $\boldsymbol{e}'^{\mathrm{GMN}}_{(\nu_j^{(\ell)},\beta^{(\ell)})}$ are computed by applying the GMN edge update function to the bias edges, using the stored bias node feature $\boldsymbol{h}_{\beta^{(\ell)}}^{\mathrm{GMN}}$ and neuron node feature $\boldsymbol{h}_{\nu_j^{(\ell)}}^{\mathrm{GMN}}$.

**Step 2.3: Message computation and aggregation.** The fifth NG layer $L_5$ computes GMN messages and aggregates them for neuron node updates. This layer simulates the GMN message computation and aggregation process. For each edge $e = (\nu_i^{(\ell-1)}, \nu_j^{(\ell)})$, the message function $\phi_{m,5}$ extracts the appropriate GMN messages. For forward edges:

$$
\boldsymbol{m}_{(\nu_i^{(\ell-1)},\nu_j^{(\ell)})}^{(3)} := \phi_{m,5}(\boldsymbol{h}_{\nu_i^{(\ell-1)}}^{(3)}, \boldsymbol{h}_{\nu_j^{(\ell)}}^{(3)}, \boldsymbol{e}_e^{(3)}) = \big[\boldsymbol{m}_{(\nu_i^{(\ell-1)},\nu_j^{(\ell)})}^{\mathrm{GMN}}, \ \boldsymbol{m}_{(\beta^{(\ell)},\nu_j^{(\ell)})}^{\mathrm{GMN}}, \ \boldsymbol{0}\big],
\tag{81}
$$

where $\boldsymbol{m}_{(\nu_i^{(\ell-1)},\nu_j^{(\ell)})}^{\mathrm{GMN}}$ is the GMN message from neuron $\nu_i^{(\ell-1)}$ to $\nu_j^{(\ell)}$, and $\boldsymbol{m}_{(\beta^{(\ell)},\nu_j^{(\ell)})}^{\mathrm{GMN}}$ is the GMN message from bias node $\beta^{(\ell)}$ to neuron $\nu_j^{(\ell)}$. For backward edges:

$$
\boldsymbol{m}_{(\nu_j^{(\ell)},\nu_i^{(\ell-1)})}^{(3)} := \phi_{m,5}(\boldsymbol{h}_{\nu_j^{(\ell)}}^{(3)}, \boldsymbol{h}_{\nu_i^{(\ell-1)}}^{(3)}, \boldsymbol{e}_e^{(3)}) = \big[\boldsymbol{m}_{(\nu_j^{(\ell)},\nu_i^{(\ell-1)})}^{\mathrm{GMN}}, \ \boldsymbol{0}, \ \boldsymbol{m}_{(\nu_j^{(\ell)},\beta^{(\ell)})}^{\mathrm{GMN}}\big],
\tag{82}
$$

where $\boldsymbol{m}_{(\nu_j^{(\ell)},\beta^{(\ell)})}^{\mathrm{GMN}}$ is the GMN message from neuron $\nu_j^{(\ell)}$ to bias node $\beta^{(\ell)}$. The message function can distinguish between forward and backward edges using the $\mathrm{dir}(e)$ feature.

The node update function $\phi_{h,5}$ aggregates messages and applies the GMN node update. For each neuron node $\nu_j^{(\ell)}$, we aggregate messages from all neighbors:

$$
\begin{aligned}
\boldsymbol{s}_{\nu_j^{(\ell)}} &:= \sum_{i=1}^{d_{\ell-1}} \boldsymbol{m}_{(\nu_i^{(\ell-1)},\nu_j^{(\ell)})}^{(3)} + \sum_{k=1}^{d_{\ell+1}} \boldsymbol{m}_{(\nu_k^{(\ell+1)},\nu_j^{(\ell)})}^{(3)} \\
&= \Big[\sum_{i=1}^{d_{\ell-1}} \boldsymbol{m}_{(\nu_i^{(\ell-1)},\nu_j^{(\ell)})}^{\mathrm{GMN}} + \sum_{k=1}^{d_{\ell+1}} \boldsymbol{m}_{(\nu_k^{(\ell+1)},\nu_j^{(\ell)})}^{\mathrm{GMN}}, \ d_{\ell-1} \cdot \boldsymbol{m}_{(\beta^{(\ell)},\nu_j^{(\ell)})}^{\mathrm{GMN}}, \ \sum_{k=1}^{d_{\ell+1}} \boldsymbol{m}_{(\nu_j^{(\ell)},\beta^{(\ell+1)})}^{\mathrm{GMN}}\Big].
\end{aligned}
\tag{83}
$$

Note that the bias message $m^{\text{GMN}}_{(\beta^{(\ell)},\nu_j^{(\ell)})}$ appears with coefficient $d_{\ell-1}$ because it is stored in each forward message. We use the stored value $d_{\ell-1}$ in $h^{(3)}_{\nu_j^{(\ell)}}$ to cancel this coefficient, extracting the true bias message. The node update then applies the GMN node update function:

$$
\begin{aligned}
h^{(4)}_{\nu_j^{(\ell)}} &:= \phi_{h,5}\Big(h^{(3)}_{\nu_j^{(\ell)}},\ s_{\nu_j^{(\ell)}}\Big) \\
&= \big[h^{\text{GMN}}_{\nu_j^{(\ell)}},\ h'^{\text{GMN}}_{\nu_j^{(\ell)}},\ h^{\text{GMN}}_{\beta^{(\ell)}},\ e^{\text{GMN}}_{(\beta^{(\ell)},\nu_j^{(\ell)})},\ e'^{\text{GMN}}_{(\beta^{(\ell)},\nu_j^{(\ell)})}, \\
&\qquad e^{\text{GMN}}_{(\nu_j^{(\ell)},\beta^{(\ell)})},\ e'^{\text{GMN}}_{(\nu_j^{(\ell)},\beta^{(\ell)})},\ u^{\text{GMN}},\ d_{\ell-1}\big],
\end{aligned}
\tag{84}
$$

where $h'^{\text{GMN}}_{\nu_j^{(\ell)}}$ is the updated GMN neuron node feature, computed by applying $\phi_h^{\text{GMN}}$ to the aggregated messages, the original node feature, and the global feature. The edge features remain unchanged: $e^{(4)}_e := e^{(3)}_e$.

**Step 2.4: Bias node updates.** The sixth NG layer $L_6$ updates the GMN bias node features. This layer aggregates messages from all neurons in a layer to update the corresponding bias node feature. We compute messages from all neurons in layer $\ell$ to the bias node $\beta^{(\ell)}$:

$$
m^{(4)}_{(\nu_i^{(\ell-1)},\nu_j^{(\ell)})} := \phi_{m,6}(h^{(4)}_{\nu_i^{(\ell-1)}}, h^{(4)}_{\nu_j^{(\ell)}}, e^{(4)}_e) = \sum_{k=1}^{d_\ell} m^{\text{GMN}}_{(\nu_k^{(\ell)},\beta^{(\ell)})},
\tag{85}
$$

and for backward edges: $m^{(4)}_{(\nu_j^{(\ell)},\nu_i^{(\ell-1)})} := 0$. The node update aggregates these messages and applies the GMN bias node update:

$$
\begin{aligned}
h^{(5)}_{\nu_j^{(\ell)}} &:= \phi_{h,6}\Big(h^{(4)}_{\nu_j^{(\ell)}},\ \sum_{i=1}^{d_{\ell-1}} m^{(4)}_{(\nu_i^{(\ell-1)},\nu_j^{(\ell)})} + \sum_{k=1}^{d_{\ell+1}} m^{(4)}_{(\nu_k^{(\ell+1)},\nu_j^{(\ell)})}\Big) \\
&= \big[h^{\text{GMN}}_{\nu_j^{(\ell)}},\ h'^{\text{GMN}}_{\nu_j^{(\ell)}},\ h^{\text{GMN}}_{\beta^{(\ell)}},\ h'^{\text{GMN}}_{\beta^{(\ell)}}, \\
&\qquad e^{\text{GMN}}_{(\beta^{(\ell)},\nu_j^{(\ell)})},\ e'^{\text{GMN}}_{(\beta^{(\ell)},\nu_j^{(\ell)})},\ e^{\text{GMN}}_{(\nu_j^{(\ell)},\beta^{(\ell)})},\ e'^{\text{GMN}}_{(\nu_j^{(\ell)},\beta^{(\ell)})},\ u^{\text{GMN}},\ d_{\ell-1}\big],
\end{aligned}
\tag{86}
$$

where $h'^{\text{GMN}}_{\beta^{(\ell)}}$ is the updated GMN bias node feature. The edge features remain unchanged: $e^{(5)}_e := e^{(4)}_e = [e^{\text{GMN}}_e, e'^{\text{GMN}}_e, u^{\text{GMN}}]$.

**Phase 3: Global feature update.** The final phase simulates the GMN global feature update $\phi_u^{\text{GMN}}$, which aggregates all node and edge features. The GMN global update function requires computing:

$$
u'^{\text{GMN}} = \phi_u^{\text{GMN}}\Big(\sum_{v \in N} h'^{\text{GMN}}_v,\ \sum_{e \in E} e'^{\text{GMN}}_e,\ u^{\text{GMN}}\Big),
\tag{87}
$$

where $N$ and $E$ are the GMN node and edge sets, respectively. Our goal is to compute the two sums $\sum_{v \in N} h'^{\text{GMN}}_v$ and $\sum_{e \in E} e'^{\text{GMN}}_e$ using NG message passing.

**Step 3.1: Feature consolidation.** The seventh NG layer $L_7$ consolidates the GMN features to prepare for aggregation. This layer combines neuron and bias node features within each layer, and pairs forward and backward bias edge features:

$$
h^{(6)}_{\nu_j^{(\ell)}} := \big[h'^{\text{GMN}}_{\nu_j^{(\ell)}} + h'^{\text{GMN}}_{\beta^{(\ell)}},\ e'^{\text{GMN}}_{(\beta^{(\ell)},\nu_j^{(\ell)})} + e'^{\text{GMN}}_{(\nu_j^{(\ell)},\beta^{(\ell)})},\ u^{\text{GMN}}\big],
\tag{88}
$$

$$
e^{(6)}_e := \big[e'^{\text{GMN}}_e,\ u^{\text{GMN}}\big].
\tag{89}
$$

This consolidation ensures that each neuron node carries both its own feature and the corresponding bias node feature, simplifying subsequent aggregation steps.

**Step 3.2: Forward propagation of sums.** We now propagate layer-wise sums forward through the network. For each layer $\ell \in [L]$, we define an NG layer $L_\ell$ that performs two operations:

- **Node aggregation**: Sums all node features from layer $\ell - 1$ and adds this sum to each node in layer $\ell$. This propagates the cumulative sum of all previous layers forward.
- **Forward edge aggregation**: Sums all forward edge features in layer $\ell$ and adds this sum to each incident node in layer $\ell$. This captures forward edge contributions as we move through the network.

After applying $L$ such layers sequentially ($L_1 \circ L_2 \circ \cdots \circ L_L$), each node in the last layer (layer $L$) contains:

$$h_{\nu_j^{(L)}}^{(L+6)} = \Big[ \sum_{\ell=0}^{L-1} \sum_{i=1}^{d_\ell} (h_{\nu_i^{(\ell)}}'^{\mathrm{GMN}} + h_{\beta(\ell)}'^{\mathrm{GMN}}) + \text{(forward edge contributions)}, \ u^{\mathrm{GMN}} \Big], \tag{90}$$

where the forward edge contributions include sums over all forward weight and bias edges. *Crucially*, nodes within the last layer do not communicate with each other during forward propagation, so the sum of last-layer nodes themselves is not yet included.

**Step 3.3: Backward propagation to complete aggregation.** We propagate backward to complete the aggregation. The backward propagation serves two essential purposes:

- **Backward edge aggregation**: Sum backward edges (which were not included in the forward pass).
- **Last-layer node aggregation**: Aggregate the last-layer node features, since they don't communicate with each other during forward propagation.

For each layer $\ell \in [L]$, we define an NG layer $M_\ell$ that:

- **Backward edge summation**: Sums all backward edge features from layer $\ell$ and adds this sum to each node in layer $\ell - 1$.
- **Last-layer aggregation** (for $\ell = L$): Sums all last-layer node features and propagates this sum backward to layer $L - 1$.

After applying $L$ backward layers sequentially ($M_L \circ M_{L-1} \circ \cdots \circ M_1$), each first-layer node contains the complete sums:

$$h_{\nu_j^{(0)}}^{(2L+6)} = \Big[ \sum_{\ell=0}^{L} \sum_{i=1}^{d_\ell} (h_{\nu_i^{(\ell)}}'^{\mathrm{GMN}} + h_{\beta(\ell)}'^{\mathrm{GMN}}), \ \sum_{\ell=1}^{L} \sum_{e \in E_\ell} e_e'^{\mathrm{GMN}}, \ u^{\mathrm{GMN}} \Big], \tag{91}$$

where $E_\ell$ denotes all edges (both forward and backward) in layer $\ell$. The complete sum now includes:

- All node features from layers $0$ through $L$ (including the last-layer nodes that were aggregated during backward propagation),
- All edge features from both forward and backward directions.

**Step 3.4: Global feature update and broadcast.** Finally, we apply the GMN global update function $\phi_u^{\mathrm{GMN}}$ to compute the updated global feature. Each first-layer node now has access to the complete sums, so we can apply the update function pointwise:

$$u'^{\mathrm{GMN}} = \phi_u^{\mathrm{GMN}} \Big( \sum_{\ell=0}^{L} \sum_{i=1}^{d_\ell} (h_{\nu_i^{(\ell)}}'^{\mathrm{GMN}} + h_{\beta(\ell)}'^{\mathrm{GMN}}), \ \sum_{\ell=1}^{L} \sum_{e \in E_\ell} e_e'^{\mathrm{GMN}}, \ u^{\mathrm{GMN}} \Big), \tag{92}$$

where $E_\ell$ denotes all edges in layer $\ell$. The updated global feature $u'^{\mathrm{GMN}}$ is then broadcast to all nodes using additional NG layers that copy it forward from layer to layer, ensuring every node has access to the updated global state.

**Conclusion.** We have shown that a single GMN layer can be simulated by a constant-depth composition of NG layers (specifically, $O(L)$ NG layers). Since the GMN functions $\phi_m^{\mathrm{GMN}}$, $\phi_h^{\mathrm{GMN}}$, $\phi_e^{\mathrm{GMN}}$, and $\phi_u^{\mathrm{GMN}}$ are MLPs, and all operations in our construction are either pointwise MLP applications or message aggregations (which NG supports), the simulation is exact on compact sets. By lemma D.13 which shows that layer-wise approximation implies network approximation, we conclude that $\mathrm{GMN}_{\mathrm{eq}}^A \preceq_K \mathrm{NG}_{\mathrm{eq}}^A$.

$\square$

### D.4.4. NG-GNN AND DWSNETS

Finally, we close the cycle by showing that NG-GNN networks can be approximated by DWS networks. This completes the cycle and establishes mutual expressivity. The proof shows that DWS networks can simulate NG-GNN's graph-based

message passing operations by using their equivariant affine layers to aggregate information across the graph structure encoded in the weight space.

**Proposition D.20.** *Let $K \subset \mathcal{V}^{c_{\mathrm{in}}}$ be a compact set. Then* $\mathrm{NG}_{\mathrm{eq}}^{A} \preceq_{K} \mathrm{DWSNets}_{\mathrm{eq}}^{A}$.

*Proof.* It suffices to show that any single NG layer (defined in equation 32, equation 33, and equation 34) can be implemented by a DWSNet. Let $\mathsf{G}_{\mathrm{NG}} = (N, E)$ be the NG graph as in Definition D.11, and fix an NG layer $\mathcal{L} = (\phi_m, \phi_h, \phi_e)$.

Recall that the weight space $\mathcal{V}^{c_{\mathrm{in}}}$ is defined as:

$$\mathcal{V}^{c_{\mathrm{in}}} := \Big(\bigoplus_{\ell=1}^{L} \mathcal{W}_{\ell}^{c_{\mathrm{in}}}\Big) \oplus \Big(\bigoplus_{\ell=1}^{L} \mathcal{B}_{\ell}^{c_{\mathrm{in}}}\Big) \cong \bigoplus_{\ell=1}^{L} \big(\mathbb{R}^{d_\ell \times d_{\ell-1} \times c_{\mathrm{in}}} \oplus \mathbb{R}^{d_\ell \times c_{\mathrm{in}}}\big). \tag{93}$$

By the DWS basis characterization (Navon et al., 2023), any DWS affine layer is a linear combination of basis maps of four types: $\mathcal{W}_\ell \to \mathcal{W}_{\ell'}$, $\mathcal{B}_\ell \to \mathcal{B}_{\ell'}$, $\mathcal{W}_\ell \to \mathcal{B}_{\ell'}$, and $\mathcal{B}_\ell \to \mathcal{W}_{\ell'}$ (for $0 \le \ell, \ell' \le L$), plus a bias term. These basis maps extend naturally to $c_{\mathrm{in}}$ feature channels by acting identically on each channel. By Corollary D.15, DWSNets can implement any MLP applied pointwise on feature channels.

Our DWSNet construction is a composition of four maps, $M_4 \circ L_3 \circ M_2 \circ L_1$:

(a) **Feature preparation** ($L_1$): Construct edge and node features using DWS affine layers, storing them in weight and bias tensors. We use a single weight entry to store features for both directions of each edge pair.
(b) **Message & edge update** ($M_2$): Apply the NG MLPs $\phi_m$ and $\phi_e$ pointwise on feature channels to compute messages and updated edge features (realizable by Corollary D.15).
(c) **Message aggregation** ($L_3$): Aggregate messages via DWS pooling: $\mathcal{W}_\ell \to \mathcal{B}_\ell$ for forward edges and $\mathcal{W}_{\ell+1} \to \mathcal{B}_\ell$ for backward edges.
(d) **Node update** ($M_4$): Apply the NG MLP $\phi_h$ pointwise on feature channels to compute updated node features $h'_v$ (realizable by Corollary D.15).

We now show that each map $L_1, M_2, L_3, M_4$ can be realized by a DWSNet.

**(a) Feature preparation ($L_1$).** Recall that edge and node features are defined as follows. For an edge $e = (\nu_i^{(\ell-1)}, \nu_j^{(\ell)})$:

$$\boldsymbol{e}_e := \big[p(e),\ \mathrm{layer}(e),\ \mathrm{dir}(e)\big] \in \mathbb{R}^{d_e}, \quad \text{where } d_e := c_{\mathrm{in}} + L + 2,$$

with $p(e) = (\boldsymbol{W}_\ell)_{i,j,:} \in \mathbb{R}^{c_{\mathrm{in}}}$ (edge parameter), $\mathrm{layer}(e) = \boldsymbol{e}_\ell^{(L)} \in \mathbb{R}^L$ (one-hot layer encoding), and $\mathrm{dir}(e) \in \{\boldsymbol{e}_1^{(2)}, \boldsymbol{e}_2^{(2)}\}$ (forward/backward indicator). For a node $\nu_i^{(\ell)}$:

$$\boldsymbol{h}_{\nu_i^{(\ell)}} := \big[\mathrm{layer}(\nu_i^{(\ell)}),\ \mathrm{ntype}(\nu_i^{(\ell)}),\ (\boldsymbol{b}_\ell)_{i,:}\big] \in \mathbb{R}^{d_h}, \quad \text{where } d_h := (L+1) + |\mathcal{T}_{\mathrm{NG}}| + c_{\mathrm{in}},$$

with $\mathrm{layer}(\nu_i^{(\ell)}) = \boldsymbol{e}_\ell^{(L+1)}$ (node layer encoding), $\mathrm{ntype}(\nu_i^{(\ell)}) \in \mathbb{R}^{|\mathcal{T}_{\mathrm{NG}}|}$ (node type—shared for hidden neurons, unique for boundary neurons), and $(\boldsymbol{b}_\ell)_{i,:}$ (the associated bias parameter).

We store edge features for both directions and the incident node features in the weight tensors, while also storing node features in the bias tensors for use in subsequent steps. Define the feature preparation map $L_1 \colon V^{c_{\mathrm{in}}} \to V^{c_1}$, where $c_1 := 2d_e + 2d_h$, by:

$$\{\boldsymbol{W}_\ell^{(1)}, \boldsymbol{b}_\ell^{(1)}\}_{\ell=0}^{L} := L_1\big(\{\boldsymbol{W}_\ell, \boldsymbol{b}_\ell\}_{\ell=0}^{L}\big),$$

$$(\boldsymbol{W}_\ell^{(1)})_{i,j,:} := \big[\boldsymbol{e}_e,\ \boldsymbol{e}_{\bar{e}},\ \boldsymbol{h}_{\nu_i^{(\ell-1)}},\ \boldsymbol{h}_{\nu_j^{(\ell)}}\big] \in \mathbb{R}^{c_1}, \quad \text{where } e = (\nu_i^{(\ell-1)}, \nu_j^{(\ell)}),\ \bar{e} = (\nu_j^{(\ell)}, \nu_i^{(\ell-1)}),$$

$$(\boldsymbol{b}_\ell^{(1)})_{j,:} := \big[\boldsymbol{0}_{2d_e + d_h},\ \boldsymbol{h}_{\nu_j^{(\ell)}}\big] \in \mathbb{R}^{c_1}.$$

We now show that $L_1$ is a DWS $G$-equivariant affine layer, i.e., $L_1 \in \mathrm{Hom}_G(V^{c_{\mathrm{in}}}, V^{c_1})$, by analyzing each component of the output.

Expanding the definition, the output weight tensor at layer $\ell \in [L]$ is:

$$(\boldsymbol{W}_\ell^{(1)})_{i,j,:} = \Big[ \underbrace{(\boldsymbol{W}_\ell)_{i,j,:}}_{(i)}, \ \underbrace{\boldsymbol{e}_\ell^{(L)}, \boldsymbol{e}_1^{(2)}}_{(iv)}, \ \underbrace{(\boldsymbol{W}_\ell)_{i,j,:}}_{(i)}, \ \underbrace{\boldsymbol{e}_\ell^{(L)}, \boldsymbol{e}_2^{(2)}}_{(iv)},$$
$$\underbrace{\boldsymbol{e}_{\ell-1}^{(L+1)}, \ \mathrm{ntype}(\nu_i^{(\ell-1)})}_{(iv)}, \ \underbrace{(\boldsymbol{b}_{\ell-1})_{i,:}}_{(ii)}, \ \underbrace{\boldsymbol{e}_\ell^{(L+1)}, \ \mathrm{ntype}(\nu_j^{(\ell)})}_{(iv)}, \ \underbrace{(\boldsymbol{b}_\ell)_{j,:}}_{(iii)} \Big].$$

We verify that each component is affine and $G$-equivariant:

(i) **Weight-to-weight identity** $\mathcal{W}_\ell \to \mathcal{W}_\ell$ (Table 5 in (Navon et al., 2023)): The term $(\boldsymbol{W}_\ell)_{i,j,:}$ is linear in the input. For equivariance, let $\tau = (\tau_1, \ldots, \tau_{L-1}) \in G$. The group action gives $(\tau \cdot \boldsymbol{W}_\ell)_{\tau_{\ell-1}(i), \tau_\ell(j),:} = (\boldsymbol{W}_\ell)_{i,j,:}$, which matches precisely the action on the output at position $(\tau_{\ell-1}(i), \tau_\ell(j))$.

(ii) **Row-broadcast** $\mathcal{B}_{\ell-1} \to \mathcal{W}_\ell$ (Table 8 in (Navon et al., 2023)): The term $(\boldsymbol{b}_{\ell-1})_{i,:}$ is linear in $\boldsymbol{b}_{\ell-1}$. The same bias vector is broadcast to all columns $j$, so the output at $(i,j)$ depends only on row index $i$. Under $\tau \in G$, both $(\boldsymbol{b}_{\ell-1})_i$ and the output row index transform by $\tau_{\ell-1}$, preserving equivariance.

(iii) **Column-broadcast** $\mathcal{B}_\ell \to \mathcal{W}_\ell$ (Table 8 in (Navon et al., 2023)): The term $(\boldsymbol{b}_\ell)_{j,:}$ is linear in $\boldsymbol{b}_\ell$. The same bias vector is broadcast to all rows $i$, so the output at $(i,j)$ depends only on column index $j$. Under $\tau \in G$, both $(\boldsymbol{b}_\ell)_j$ and the output column index transform by $\tau_\ell$, preserving equivariance.

(iv) **Affine bias terms**: The layer encodings $\boldsymbol{e}_\ell^{(L)}$, direction encodings $\boldsymbol{e}_1^{(2)}, \boldsymbol{e}_2^{(2)}$, layer indices $\boldsymbol{e}_{\ell-1}^{(L+1)}, \boldsymbol{e}_\ell^{(L+1)}$, and node types $\mathrm{ntype}(\cdot)$ are all constant (affine with zero linear part). In addition, $\mathrm{ntype}$ is equivariant because:
  - At hidden layers ($0 < \ell < L$), all neurons share the same type $\mathrm{ntype}(\nu_i^{(\ell)}) = \boldsymbol{e}_{\mathrm{hidden}}^{(|\mathcal{T}_{\mathrm{NG}}|)}$, so the output is constant across permutations.
  - At boundary layers ($\ell \in \{0, L\}$), the group $G$ acts trivially on input/output neurons, so boundary-specific encodings are $G$-invariant.

For the bias tensor $(\boldsymbol{b}_\ell^{(1)})_{j,:} = [\boldsymbol{0}_{2d_e+d_h}, \boldsymbol{h}_{\nu_j^{(\ell)}}]$, we expand:

$$(\boldsymbol{b}_\ell^{(1)})_{j,:} = \Big[ \underbrace{\boldsymbol{0}_{2d_e+d_h}}_{(iv)}, \ \underbrace{\boldsymbol{e}_\ell^{(L+1)}, \ \mathrm{ntype}(\nu_j^{(\ell)})}_{(iv)}, \ \underbrace{(\boldsymbol{b}_\ell)_{j,:}}_{(v)} \Big].$$

(v) **Bias-to-bias identity** $\mathcal{B}_\ell \to \mathcal{B}_\ell$ (Table 6 in (Navon et al., 2023)): The term $(\boldsymbol{b}_\ell)_{j,:}$ is copied via the identity. Under $\tau \in G$, both the input $(\boldsymbol{b}_\ell)_j$ and output $(\boldsymbol{b}_\ell^{(1)})_j$ transform by $\tau_\ell$, preserving equivariance.

Since $L_1$ is a concatenation of affine $G$-equivariant maps, we have $L_1 \in \mathrm{Hom}_G(V^{c_{\mathrm{in}}}, V^{c_1})$.

**(b) Message & edge update** ($M_2$). In the NG architecture (Definition D.11), the functions $\phi_m$ and $\phi_e$ are MLPs equation 32–equation 34. By Corollary D.15, we can apply them pointwise on feature channels, yielding:

$$\big( \boldsymbol{m}_e, \boldsymbol{e}_e' \big) = \big( \phi_m(\boldsymbol{h}_u, \boldsymbol{h}_v, \boldsymbol{e}_e), \ \phi_e(\boldsymbol{h}_u, \boldsymbol{h}_v, \boldsymbol{e}_e) \big), \quad \forall e = (u, v) \in E,$$

where $\boldsymbol{m}_e \in \mathbb{R}^{d_{\mathrm{msg}}}$ is the message and $\boldsymbol{e}_e' \in \mathbb{R}^{d_e}$ is the updated edge feature.

We use the feature vector stored in $\boldsymbol{W}_\ell^{(1)}$ to compute messages and edge updates for both edge directions.

Define the map $M_2\colon V^{c_1} \to V^{c_2}$, where $c_2 := 2(d_e + d_{\mathrm{msg}}) + d_h$, by:

$$\{\boldsymbol{W}_\ell^{(2)}, \boldsymbol{b}_\ell^{(2)}\}_{\ell=0}^L := M_2\big(\{\boldsymbol{W}_\ell^{(1)}, \boldsymbol{b}_\ell^{(1)}\}_{\ell=0}^L\big),$$

$$(\boldsymbol{W}_\ell^{(2)})_{i,j,:} := (\phi_m, \phi_e)^\oplus\big((\boldsymbol{W}_\ell^{(1)})_{i,j,:}\big) = (\phi_m, \phi_e)^\oplus\big(\boldsymbol{e}_e,\, \boldsymbol{e}_{\bar{e}},\, \boldsymbol{h}_{\nu_i^{(\ell-1)}},\, \boldsymbol{h}_{\nu_j^{(\ell)}}\big)$$

$$= \Big[ \underbrace{\phi_m(\boldsymbol{h}_{\nu_i^{(\ell-1)}}, \boldsymbol{h}_{\nu_j^{(\ell)}}, \boldsymbol{e}_e)}_{\boldsymbol{m}_e},\; \underbrace{\phi_e(\boldsymbol{h}_{\nu_i^{(\ell-1)}}, \boldsymbol{h}_{\nu_j^{(\ell)}}, \boldsymbol{e}_e)}_{\boldsymbol{e}_e'},$$

$$\underbrace{\phi_m(\boldsymbol{h}_{\nu_j^{(\ell)}}, \boldsymbol{h}_{\nu_i^{(\ell-1)}}, \boldsymbol{e}_{\bar{e}})}_{\boldsymbol{m}_{\bar{e}}},\; \underbrace{\phi_e(\boldsymbol{h}_{\nu_j^{(\ell)}}, \boldsymbol{h}_{\nu_i^{(\ell-1)}}, \boldsymbol{e}_{\bar{e}})}_{\boldsymbol{e}_{\bar{e}}'},\; \boldsymbol{h}_{\nu_j^{(\ell)}} \Big] \in \mathbb{R}^{c_2},$$

$$(\boldsymbol{b}_\ell^{(2)})_{j,:} := \big[\boldsymbol{0}_{2(d_e+d_{\mathrm{msg}})},\, \boldsymbol{h}_{\nu_j^{(\ell)}}\big] \in \mathbb{R}^{c_2}.$$

Here $(\phi_m, \phi_e)^\oplus$ denotes the pointwise application of the NG MLPs $\phi_m$ and $\phi_e$ to both edge directions, selecting the appropriate inputs from the feature vector. The node features $\boldsymbol{h}_{\nu_j^{(\ell)}}$ are preserved via the bias-to-bias identity $\mathcal{B}_\ell \to \mathcal{B}_\ell$.

**(c) Message aggregation** ($L_3$). For each node $\nu_j^{(\ell)} \in N$, we aggregate messages from incident edges. Recall that $(\boldsymbol{W}_\ell^{(2)})_{i,j,:}$ stores the messages $\boldsymbol{m}_e$ (forward) and $\boldsymbol{m}_{\bar{e}}$ (backward), along with the updated edge features $\boldsymbol{e}_e', \boldsymbol{e}_{\bar{e}}'$.

Define the map $L_3\colon V^{c_2} \to V^{c_3}$, where $c_3 := 2d_e + d_h + d_{\mathrm{msg}}$, by:

$$\{\boldsymbol{W}_\ell^{(3)}, \boldsymbol{b}_\ell^{(3)}\}_{\ell=0}^L := L_3\big(\{\boldsymbol{W}_\ell^{(2)}, \boldsymbol{b}_\ell^{(2)}\}_{\ell=0}^L\big),$$

$$(\boldsymbol{W}_\ell^{(3)})_{i,j,:} := \big[\boldsymbol{e}_e',\, \boldsymbol{e}_{\bar{e}}',\, \boldsymbol{0}_{d_h+d_{\mathrm{msg}}}\big] \in \mathbb{R}^{c_3},$$

$$(\boldsymbol{b}_\ell^{(3)})_{j,:} := \big[\boldsymbol{0}_{2d_e},\, \boldsymbol{h}_{\nu_j^{(\ell)}},\, \boldsymbol{s}_{\nu_j^{(\ell)}}\big] \in \mathbb{R}^{c_3},$$

where the aggregated message for node $\nu_j^{(\ell)}$ is:

$$\boldsymbol{s}_{\nu_j^{(\ell)}} = \underbrace{\sum_{i=1}^{d_{\ell-1}} \boldsymbol{m}_e}_{\text{forward from layer } \ell-1} + \underbrace{\sum_{k=1}^{d_{\ell+1}} \boldsymbol{m}_{\bar{e}'}}_{\text{backward from layer } \ell+1} \in \mathbb{R}^{d_{\mathrm{msg}}},$$

with $e = (\nu_i^{(\ell-1)}, \nu_j^{(\ell)})$ and $\bar{e}' = (\nu_j^{(\ell)}, \nu_k^{(\ell+1)})$.

This is realized via the following DWS operations:

(i) **Weight-to-weight identity** $\mathcal{W}_\ell \to \mathcal{W}_\ell$: Preserves the updated edge features $\boldsymbol{e}_e', \boldsymbol{e}_{\bar{e}}'$.
(ii) **Column-pool** $\mathcal{W}_\ell \to \mathcal{B}_\ell$ (Table 7 in (Navon et al., 2023)): Sums $\boldsymbol{m}_e$ over index $i$ to aggregate forward messages.
(iii) **Row-pool** $\mathcal{W}_{\ell+1} \to \mathcal{B}_\ell$ (Table 7 in (Navon et al., 2023)): Sums $\boldsymbol{m}_{\bar{e}'}$ over index $k$ to aggregate backward messages.
(iv) **Bias-to-bias identity** $\mathcal{B}_\ell \to \mathcal{B}_\ell$: Preserves the node feature $\boldsymbol{h}_{\nu_j^{(\ell)}}$ from $\boldsymbol{b}_\ell^{(2)}$.

At boundary layers ($\ell = 0$ or $\ell = L$), only one edge direction exists, so the corresponding pooling is omitted.

**(d) Node update** ($M_4$). The NG function $\phi_h$ is an MLP equation 33. Define $M_4\colon V^{c_3} \to V^{c_4}$, where $c_4 := 2d_e + d_h'$, by:

$$\{\boldsymbol{W}_\ell^{(4)}, \boldsymbol{b}_\ell^{(4)}\}_{\ell=0}^L := M_4\big(\{\boldsymbol{W}_\ell^{(3)}, \boldsymbol{b}_\ell^{(3)}\}_{\ell=0}^L\big),$$

$$(\boldsymbol{W}_\ell^{(4)})_{i,j,:} := \big[\boldsymbol{e}_e',\, \boldsymbol{e}_{\bar{e}}',\, \boldsymbol{0}_{d_h'}\big] \in \mathbb{R}^{c_4},$$

$$(\boldsymbol{b}_\ell^{(4)})_{j,:} := \big[\boldsymbol{0}_{2d_e},\, \phi_h\big(\boldsymbol{h}_{\nu_j^{(\ell)}}, \boldsymbol{s}_{\nu_j^{(\ell)}}\big)\big] =: \big[\boldsymbol{0}_{2d_e},\, \boldsymbol{h}_{\nu_j^{(\ell)}}'\big] \in \mathbb{R}^{c_4}.$$

The updated edge features $\boldsymbol{e}_e', \boldsymbol{e}_{\bar{e}}'$ are preserved via the weight-to-weight identity, while the updated node features $\boldsymbol{h}_{\nu_j^{(\ell)}}'$ are stored in the bias.

Finally, since the NG functions $\phi_m$, $\phi_e$, and $\phi_h$ are MLPs, by Corollary D.15 they can be applied pointwise on feature channels, and all other operations are DWS linear maps. Therefore, the composition $M_4 \circ L_3 \circ M_2 \circ L_1$ exactly implements the NG layer using a constant-depth DWSNet. By lemma D.13 which shows that layer-wise approximation implies network approximation, we conclude that $\mathrm{NG}^A_{\mathrm{eq}} \preceq_K \mathrm{DWSNets}^A_{\mathrm{eq}}$. $\qquad\square$

### D.5. Expressive power of NFT

In this subsection, we compare the expressive power of the Neural Functional Transformer (NFT) architecture (Zhou et al., 2023b) to the other permutation-equivariant weight-space models considered in this section, namely DWS, NP-NFN, HNP-NFN, GMN, and NG-GNN. In the main theorem of this section (Theorem D.6), we show that these architectures all have the same expressive power. Here, we prove Proposition 5.3, showing that without any assumptions on the input space, NFTs differ in their expressive power from all of the above weight-space networks, and that under a general-position assumption on the inputs, this gap disappears and NFTs become equivalent in expressive power to the other permutation-equivariant models. For convenience, we split the proposition into two parts and prove each one separately. We note that the proofs in this subsection rely on results proved later in the appendix; however, we present them here to maintain a natural and coherent narrative.

**Proposition D.21.** *There exists an architecture $A = (\boldsymbol{d}, \sigma)$, and a compact set of weights $K \subseteq \mathcal{V}_A$ such that*

$$\mathrm{NFT}^A_{\mathrm{eq}} \not\preceq_K \mathrm{DWSNets}^A_{\mathrm{eq}} \tag{94}$$

*(Definition D.3).*

*Proof.* By Remark D.5, showing the current statement is equivalent to showing that $\mathcal{N}^{\mathrm{NFT}}_{\mathrm{inv}}(K) \neq \mathcal{N}^{\mathrm{DWS}}_{\mathrm{inv}}(K)$ and $\mathcal{N}^{\mathrm{NFT}}_{\mathrm{equiv}}(K) \neq \mathcal{N}^{\mathrm{DWS}}_{\mathrm{equiv}}(K)$ for some compact set $K$, where all terms $\mathcal{N}^{\mathrm{NFT}}_{\mathrm{inv}}(K), \mathcal{N}^{\mathrm{DWS}}_{\mathrm{inv}}(K), \mathcal{N}^{\mathrm{NFT}}_{\mathrm{equiv}}(K), \mathcal{N}^{\mathrm{DWS}}_{\mathrm{equiv}}(K)$ are defined in Definition 5.1.

Let $A = ((1, 4, 4, 1), \mathrm{ReLU})$ and let $\mathcal{V} = \mathcal{V}_A$. Define $\boldsymbol{v}^1, \boldsymbol{v}^2 \in \mathcal{V}$ by $\boldsymbol{v}^i = (\boldsymbol{W}^i_1, \boldsymbol{b}^i_1, \boldsymbol{W}^i_2, \boldsymbol{b}^i_2, \boldsymbol{W}^i_3, \boldsymbol{b}^i_3)$, $i \in [2]$, where

$$\boldsymbol{W}^1_1 = \boldsymbol{W}^2_1 = \mathbf{1}_{4 \times 1}, \qquad \boldsymbol{b}^1_1 = \boldsymbol{b}^2_1 = 0, \qquad \boldsymbol{W}^1_3 = \boldsymbol{W}^2_3 = \mathbf{1}_{1 \times 4}, \qquad \boldsymbol{b}^1_2 = \boldsymbol{b}^2_2 = 0, \qquad \boldsymbol{b}^1_3 = \boldsymbol{b}^2_3 = 0. \tag{95}$$

Finally, define the second-layer weight matrices by

$$\boldsymbol{W}^1_2 = \begin{pmatrix} 1 & 1 & 0 & 0 \\ 0 & 1 & 1 & 0 \\ 0 & 0 & 1 & 1 \\ 1 & 0 & 0 & 1 \end{pmatrix}, \qquad \boldsymbol{W}^2_2 = \begin{pmatrix} 1 & 1 & 0 & 0 \\ 1 & 1 & 0 & 0 \\ 0 & 0 & 1 & 1 \\ 0 & 0 & 1 & 1 \end{pmatrix}. \tag{96}$$

See Figure 3 for a visualization of these networks. First, it is shown in the proof of Proposition E.8 that any invariant DWS model $\Phi : \mathcal{V} \to \mathbb{R}$ satisfies $\Phi(\boldsymbol{v}^1) = \Phi(\boldsymbol{v}^2)$. We now construct an invariant NFT model $\Psi$ such that $\Psi(\boldsymbol{v}^1) \neq \Psi(\boldsymbol{v}^2)$. This shows that for any compact $K$ containing both $\boldsymbol{v}^1, \boldsymbol{v}^2$ it holds that $\mathcal{N}^{\mathrm{NFT}}_{\mathrm{inv}}(K) \neq \mathcal{N}^{\mathrm{DWS}}_{\mathrm{inv}}(K)$.

Recalling the definition of the NFT architecture (Definition D.12), we set the first component of our NFT network, composed of maps PE, MLP and LAYERENC to be simply the identity map (this can be achieved by using MLP to ignore the concatenated positional encoding of PE and setting the learned positional encodings of LAYERENC to zero). We then define the first block $\mathrm{Block}_1$ of $\Psi$ by choosing its self-attention layer to consist only of the row-wise attention update applied to the weight matrix $\boldsymbol{W}_2$, while leaving all other weight and bias terms unchanged. Moreover, we take identity maps for the query, key, and value projections, i.e.,

$$\theta_Q = \theta_K = \theta_V = \mathrm{Id}. \tag{97}$$

Formally, for $\boldsymbol{v} = (\boldsymbol{W}_1, \boldsymbol{b}_1, \dots, \boldsymbol{W}_3, \boldsymbol{b}_3)$, writing $\mathrm{Block}_1(\boldsymbol{v}) = (\boldsymbol{W}^*_1, \boldsymbol{b}^*_1, \dots, \boldsymbol{W}^*_3, \boldsymbol{b}^*_3)$, we have

$$\boldsymbol{b}^*_\ell = \boldsymbol{b}_\ell \quad \forall \ell \in [3], \qquad \boldsymbol{W}^*_\ell = \boldsymbol{W}_\ell \quad \forall \ell \in \{1, 3\}, \tag{98}$$

and the only updated term is $\boldsymbol{W}_2$, given entrywise by

$$(\boldsymbol{W}^*_2)_{i,j} = \sum_{j'=1}^4 \alpha_i(j') (\boldsymbol{W}_2)_{i,j'}, \tag{99}$$

where the attention weights are

$$\alpha_i(j) = \frac{\exp\big(\langle (\boldsymbol{W}_2)_{i,:}, (\boldsymbol{W}_2)_{j,:} \rangle\big)}{\sum_{j'=1}^{4} \exp\big(\langle (\boldsymbol{W}_2)_{i,:}, (\boldsymbol{W}_2)_{j',:} \rangle\big)}. \tag{100}$$

After applying $\mathrm{Block}_1$ to $\boldsymbol{v}^1, \boldsymbol{v}^2$ all weight and bias features remain unchanged except $\boldsymbol{W}_2^1, \boldsymbol{W}_2^2$ which become

$$\boldsymbol{W}_2^1 = \begin{pmatrix} 0.73 & 0.73 & 0.27 & 0.27 \\ 0.27 & 0.73 & 0.73 & 0.27 \\ 0.27 & 0.27 & 0.73 & 0.73 \\ 0.73 & 0.27 & 0.27 & 0.73 \end{pmatrix}, \qquad \boldsymbol{W}_2^2 = \begin{pmatrix} 0.88 & 0.88 & 0.12 & 0.12 \\ 0.88 & 0.88 & 0.12 & 0.12 \\ 0.12 & 0.12 & 0.88 & 0.88 \\ 0.12 & 0.12 & 0.88 & 0.88 \end{pmatrix}. \tag{101}$$

We then define the second block $\mathrm{Block}_2$ of $\Psi$ to apply a pointwise MLP to each weight- and bias-feature vector independently. Specifically, we choose the MLP so that for scalar inputs it satisfies

$$x < 0.8 \;\Rightarrow\; \mathrm{MLP}(x) = 0, \qquad x > 0.85 \;\Rightarrow\; \mathrm{MLP}(x) = 1. \tag{102}$$

In this block, we disable the self-attention update by setting all value projections $\theta_V$ to zero and relying solely on the residual (skip) connection. Consequently, applying $\mathrm{Block}_2$ to $\boldsymbol{v}^1$ and $\boldsymbol{v}^2$ leaves all weights and biases unchanged except for $\boldsymbol{W}_2^1$ and $\boldsymbol{W}_2^2$, which become

$$\boldsymbol{W}_2^1 = \begin{pmatrix} 0 & 0 & 0 & 0 \\ 0 & 0 & 0 & 0 \\ 0 & 0 & 0 & 0 \\ 0 & 0 & 0 & 0 \end{pmatrix}, \qquad \boldsymbol{W}_2^2 = \begin{pmatrix} 1 & 1 & 0 & 0 \\ 1 & 1 & 0 & 0 \\ 0 & 0 & 1 & 1 \\ 0 & 0 & 1 & 1 \end{pmatrix}. \tag{103}$$

Finally, we apply an invariant pooling operator that averages all weight-space entries. This can be implemented by setting the key and query projections in the final cross-attention pooling layer CA of $\Psi$ to zero, so that the learned token attends uniformly to all terms. This yields

$$\Psi(\boldsymbol{v}^1) = \frac{8}{33}, \qquad \Psi(\boldsymbol{v}^2) = \frac{16}{33}. \tag{104}$$

In particular, $\Psi(\boldsymbol{v}^1) \neq \Psi(\boldsymbol{v}^2)$, and therefore

$$\mathcal{N}_{\mathrm{inv}}^{\mathrm{NFT}}(K) \neq \mathcal{N}_{\mathrm{inv}}^{\mathrm{DWS}}(K). \tag{105}$$

Moreover, the final head of $\Psi$ consists of summation pooling, which is also available within the DWS framework. Therefore, if there existed a DWS-equivariant model that approximates $\Psi$ up to (and excluding) this head, then composing it with summation pooling would yield an invariant DWS model separating $\boldsymbol{v}^1$ and $\boldsymbol{v}^2$, a contradiction. Hence,

$$\mathcal{N}_{\mathrm{equiv}}^{\mathrm{NFT}}(K) \neq \mathcal{N}_{\mathrm{equiv}}^{\mathrm{DWS}}(K), \tag{106}$$

which completes the proof. $\qquad\square$

**Proposition D.22.** *For any architecture $A = (\boldsymbol{d}, \sigma)$ and any compact set of weights $K \subseteq \mathcal{V}_A \setminus \mathcal{E}_A$,*

$$\mathrm{NFT}_{\mathrm{eq}}^A \simeq_K \mathrm{DWSNets}_{\mathrm{eq}}^A \tag{107}$$

*(Definition D.3).*

*Proof.* By Remark D.5, showing the current statement is equivalent to showing that $\mathcal{N}_{\mathrm{inv}}^{\mathrm{NFT}}(K) = \mathcal{N}_{\mathrm{inv}}^{\mathrm{DWS}}(K)$ and $\mathcal{N}_{\mathrm{equiv}}^{\mathrm{NFT}}(K) = \mathcal{N}_{\mathrm{equiv}}^{\mathrm{DWS}}(K)$ for any compact set $K \subseteq \mathcal{V}_A \setminus \mathcal{E}_A$. First, since $K \subseteq \mathcal{V}_A \setminus \mathcal{E}_A$, Theorems 7.4 and 6.3 show that any invariant/equivariant map can be approximated on $K$ to any precision by DWS models, thus

$$\mathcal{N}_{\mathrm{inv}}^{\mathrm{NFT}}(K) \subseteq \mathcal{N}_{\mathrm{inv}}^{\mathrm{DWS}}(K) \qquad \mathcal{N}_{\mathrm{equiv}}^{\mathrm{NFT}}(K) \subseteq \mathcal{N}_{\mathrm{equiv}}^{\mathrm{DWS}}(K). \tag{108}$$

To prove the reverse inclusion, we rely on the proof of Theorem 7.4 from Appendix F. As stated in Section F.1 the proof shows that, in order to achieve universality, it suffices for DWS models to have access to only a *restricted collection of*

*linear update operations*. We now go over these update primitives one by one, restate them for convenience, and show that each of them can be implemented using NFT-style updates. Since these updates can subsequently be *stacked* by composing multiple NFT blocks (and, when needed, using multi-headed attention), this yields a direct simulation of any DWS model constructed from these primitives. In particular, NFT models can therefore approximate any invariant/equivariant function on $K$ to arbitrary precision, completing the proof.

Concretely, for $\boldsymbol{v} \in \mathcal{V}^c$ written as $\boldsymbol{v} = (\boldsymbol{W}_1, \boldsymbol{b}_1, \ldots, \boldsymbol{W}_L, \boldsymbol{b}_L)$, the update primitives of interest are:

- Pointwise affine update.

- Global summation operator.

- Bias summation operator.

- Lower Weight-to-bias operator.

- Upper weight-to-bias operator.

- First layer per-neuron operator.

- Last layer per-neuron operator.

**Pointwise affine update.** This operator applies the same affine map to each weight and bias feature vector, using a matrix $\boldsymbol{A} \in \mathbb{R}^{c \times c'}$ and a vector $\boldsymbol{u} \in \mathbb{R}^{c'}$. Concretely, for every layer $\ell$ and all valid indices $i, j$, we update

$$(\boldsymbol{W}_\ell)_{i,j,:} \longmapsto (\boldsymbol{W}_\ell)_{i,j,:}\boldsymbol{A} + \boldsymbol{u}, \qquad (\boldsymbol{b}_\ell)_{i,:} \longmapsto (\boldsymbol{b}_\ell)_{i,:}\boldsymbol{A} + \boldsymbol{u}. \qquad (109)$$

This update is directly realizable by the pointwise MLP component of NFT (see Definition D.12 Equation 38).

**Global summation operator.** This operator aggregates all weight and bias feature vectors in $\boldsymbol{v}$ via a global sum, and then broadcasts the resulting vector back to every entry. Namely, define the global summary

$$\boldsymbol{s}(\boldsymbol{v}) \coloneqq \sum_\ell \sum_{i,j} (\boldsymbol{W}_\ell)_{i,j,:} + \sum_\ell \sum_i (\boldsymbol{b}_\ell)_{i,:} \in \mathbb{R}^c. \qquad (110)$$

Then, for every layer $\ell$ and all valid indices $i, j$, we update

$$(\boldsymbol{W}_\ell)_{i,j,:} \longmapsto \boldsymbol{s}(\boldsymbol{v}), \qquad (\boldsymbol{b}_\ell)_{i,:} \longmapsto \boldsymbol{s}(\boldsymbol{v}). \qquad (111)$$

This update can be implemented using the self-attention mechanism of NFT. Specifically, we set all key, query, and value projections to zero except for the value projection used to compute the KV3 term (see Definition D.12, Eq. 46), which we set to the identity. This yields a uniform average of all tokens in $\boldsymbol{v}$. Since the architecture (and hence the number of tokens) is fixed, we can convert this average into the desired sum by applying a subsequent pointwise MLP.

Finally, NFT blocks are equipped with a residual connection. To realize a *pure* broadcast update of the form above, we use multi-headed attention to simultaneously compute (i) the identity map $\boldsymbol{v} \mapsto \boldsymbol{v}$ and (ii) the residual update $\boldsymbol{v} \mapsto \boldsymbol{v} + \mathrm{ATTN}(\boldsymbol{v})$, and then apply a pointwise MLP to combine these channels and extract

$$\boldsymbol{v} + \mathrm{ATTN}(\boldsymbol{v}) - \boldsymbol{v} = \mathrm{ATTN}(\boldsymbol{v}), \qquad (112)$$

thus effectively canceling the residual and implementing the global summation operator.

**Bias summation operator.** Fix a layer index $\ell$. The $\ell$-th bias summation operator aggregates the bias vectors of layer $\ell$ by summation, broadcasts the result across the bias positions of that layer, and sets all other weight and bias entries to zero. Concretely, for every layer $\ell' \neq \ell$ and all valid indices $i, j$, we set

$$(\boldsymbol{W}_{\ell'})_{i,j,:} \longmapsto 0, \qquad (\boldsymbol{b}_{\ell'})_{i,:} \longmapsto 0. \qquad (113)$$

For the selected layer $\ell$, we set all weight entries to zero and replace each bias entry by the sum of all biases in that layer:

$$(\boldsymbol{W}_\ell)_{i,j,:} \longmapsto 0, \qquad (\boldsymbol{b}_\ell)_{i,:} \longmapsto \sum_{i'} (\boldsymbol{b}_\ell)_{i',:}. \qquad (114)$$

We now show how to implement this operator using NFT updates. We begin with a pointwise MLP that pads every token with an additional coordinate, set to zero. Next, we use the learned layer encoding (see Definition D.12, Eq. 39) to modify this last coordinate so that it equals 1 *exactly* for bias tokens belonging to layer $\ell$, and remains 0 for all other tokens. Applying another pointwise MLP, we can then set every token whose last coordinate is 0 to the zero vector, while leaving the remaining tokens unchanged.

Finally, we apply self-attention with all key/query/value projections set to zero except for the value projection used in the $KV3$ term (see Definition D.12, Eq. 46), which we set to the identity. This causes every token to receive the same broadcasted summary, namely the sum of the surviving bias tokens, which equals $\sum_{i'} (\boldsymbol{b}_\ell)_{i',:}$ by construction.

To remove the residual contribution and ensure that only the intended bias tokens remain nonzero, we use multi-headed attention as before: one head computes the identity map $\boldsymbol{v} \mapsto \boldsymbol{v}$, and a second head computes the attention update $\boldsymbol{v} \mapsto \boldsymbol{v} + \text{ATTN}(\boldsymbol{v})$. We then apply a pointwise MLP to the concatenation of these outputs, which outputs 0 whenever the last coordinate of the identity part is 0, and otherwise outputs

$$\boldsymbol{v} + \text{ATTN}(\boldsymbol{v}) - \boldsymbol{v} \;=\; \text{ATTN}(\boldsymbol{v}). \tag{115}$$

This completes the implementation. This

**Lower weight-to-bias operator.** This operator replaces each weight feature vector by the bias feature vector of the *lower* neuron connected to that weight, while leaving all bias terms unchanged. Concretely, for every layer $\ell$ and all valid indices $i, j$, we update

$$(\boldsymbol{W}_\ell)_{i,j,:} \;\longmapsto\; (\boldsymbol{b}_{\ell-1})_{i,:}, \qquad (\boldsymbol{b}_\ell)_{i,:} \;\longmapsto\; (\boldsymbol{b}_\ell)_{i,:}. \tag{116}$$

We now show how to implement this operator using NFT updates. As before, we begin with a pointwise MLP that pads every token with an additional coordinate, initialized to zero. We then use the learned layer encoding (see Definition D.12, Eq. 39) to modify this last coordinate so that it equals 1 for all bias tokens and remains 0 for all weight tokens. Applying another pointwise MLP, we can set every token whose last coordinate is 0 to the zero vector, while leaving the remaining tokens unchanged. In particular, this operation zeroes out all weight feature vectors and keeps only the bias tokens.

Next, we apply self-attention with all key/query/value projections set to zero except for the value projection used in the $KV1$ term (see Definition D.12, Eq. 44), which we set to the identity. Since the KV1 aggregation for a weight token $(\boldsymbol{W}_\ell)_{i,j,:}$ averages over the bias token $(\boldsymbol{b}_{\ell-1})_{i,:}$ together with the $d_\ell$ incoming weight tokens (which are now all zero), the resulting attention output is

$$(\boldsymbol{W}_\ell)_{i,j,:} \;\longmapsto\; \frac{1}{d_\ell + 1}(\boldsymbol{b}_{\ell-1})_{i,:}, \qquad (\boldsymbol{b}_\ell)_{i,:} \;\longmapsto\; 0. \tag{117}$$

Finally, to ensure the correct treatment of bias tokens and to cancel the residual connection, we use multi-headed attention as before: one head computes the identity map $\boldsymbol{v} \mapsto \boldsymbol{v}$, and a second head computes the residual update $\boldsymbol{v} \mapsto \boldsymbol{v} + \text{ATTN}(\boldsymbol{v})$. We then apply a pointwise MLP to the concatenation of these two heads, which outputs the identity head whenever the last coordinate of the identity head equals 1 (i.e., for bias tokens), and otherwise outputs the rescaled attention term

$$(d_\ell + 1) \cdot \big(\boldsymbol{v} + \text{ATTN}(\boldsymbol{v}) - \boldsymbol{v}\big) \;=\; (d_\ell + 1) \cdot \text{ATTN}(\boldsymbol{v}). \tag{118}$$

This produces exactly $(\boldsymbol{b}_{\ell-1})_{i,:}$ at every weight position $(\boldsymbol{W}_\ell)_{i,j,:}$ while leaving the bias tokens unchanged, completing the implementation.

**Upper weight-to-bias operator.** This operator is analogous to the lower weight-to-bias operator, but replaces each weight feature vector by the bias feature vector of the *upper* neuron incident to that weight, while leaving all bias terms unchanged. Concretely, for every layer $\ell$ and all valid indices $i, j$, we update

$$(\boldsymbol{W}_\ell)_{i,j,:} \;\longmapsto\; (\boldsymbol{b}_\ell)_{j,:}, \qquad (\boldsymbol{b}_\ell)_{i,:} \;\longmapsto\; (\boldsymbol{b}_\ell)_{i,:}. \tag{119}$$

The implementation follows *exactly* the construction used for the lower weight-to-bias operator, with the sole modification that the attention update is computed via the KV2 aggregation (see Definition D.12, Eq. 45) in place of KV1. This yields the desired broadcast of $(\boldsymbol{b}_\ell)_{j,:}$ to every weight token $(\boldsymbol{W}_\ell)_{i,j,:}$, while preserving all bias tokens.

**First-layer per-neuron operator.** Fix an input neuron index $i' \in [d_0]$. This operator preserves exactly the first-layer weights incident to neuron $i'$, and sets all other weight and bias terms to zero. Concretely, for all $i \in [d_0]$ and $j \in [d_1]$, we update

$$(\boldsymbol{W}_1)_{i,j,:} \longmapsto \begin{cases} (\boldsymbol{W}_1)_{i,j,:}, & \text{if } i = i', \\ 0, & \text{otherwise,} \end{cases} \tag{120}$$

and map every remaining weight and bias token in $\boldsymbol{v}$ to 0.

To implement this operator with NFT updates, we recall that NFT uses the positional encoding PE of NP-NFN (see Definition D.9), which appends to each token a type-dependent identifier. In particular, each first-layer weight token $(\boldsymbol{W}_1)_{i,j,:}$ is concatenated with an embedding that depends only on the input-neuron index $i$, and thus uniquely tags all weights incident to neuron $i$. We can therefore apply a pointwise MLP that keeps exactly the tokens with the tag corresponding to $i'$ and maps all others to zero, realizing the desired per-neuron restriction.

**Last-layer per-neuron update.** This operator preserves only the bias terms associated with a specified output neuron neuron in the last layer, and sets all other weights and bias terms to zero. Concretely, for fixed $j' \in [d_L]$ and all $j \in [d_L]$, we apply

$$(\boldsymbol{b}_L)_{j,:} \longmapsto \begin{cases} (\boldsymbol{b}_L)_{j,:}, & \text{if } j = j', \\ 0, & \text{otherwise.} \end{cases} \tag{121}$$

All remaining weight and bias terms are mapped to zero.

The operator may be implemented by NFT updates in the exact same way as the first-layer per-neuron update, using the NP-NFN positional encoding along with an MLP update.

Having implemented all required update primitives, we conclude that NFT models can simulate any DWS model constructed from these updates, completing the proof.

$\square$

# E. Expressive power of permutation-invariant weight-space networks

## E.1. Hierarchy of weight-space equivalence relations

In order to characterize the expressive power of weight-space networks, we begin by analyzing their ability to separate points in weight space. To this end, we introduce three natural equivalence relations on weight space: functional equivalence, DWS equivalence, and $G$-equivalence.

**Definition E.1** (Functional equivalence). Let $\boldsymbol{v}, \boldsymbol{v}' \in \mathcal{V}$. We say that $\boldsymbol{v}$ and $\boldsymbol{v}'$ are *functionally equivalent*, and write $\boldsymbol{v} \sim_{\text{func}} \boldsymbol{v}'$, if they realize the same function, i.e.,

$$f_{\boldsymbol{v}}(\boldsymbol{x}) = f_{\boldsymbol{v}'}(\boldsymbol{x}) \quad \forall \boldsymbol{x} \in \mathbb{R}^{d_0}. \tag{122}$$

**Definition E.2** ($G$-equivalence). Let $\boldsymbol{v}, \boldsymbol{v}' \in \mathcal{V}$. We say that $\boldsymbol{v}$ and $\boldsymbol{v}'$ are *$G$-equivalent*, and write $\boldsymbol{v} \sim_G \boldsymbol{v}'$, if there exists $g \in G$ such that

$$\boldsymbol{v}' = \rho(g)\,\boldsymbol{v}. \tag{123}$$

**Definition E.3** (DWS-equivalence). Let $\boldsymbol{v}, \boldsymbol{v}' \in \mathcal{V}$. We say that $\boldsymbol{v}$ and $\boldsymbol{v}'$ are *DWS-equivalent*, and write $\boldsymbol{v} \sim_{\text{DWS}} \boldsymbol{v}'$, if for all invariant DWS networks $\Phi : \mathcal{V} \to \mathbb{R}^n$ it holds that

$$\Phi(\boldsymbol{v}) = \Phi(\boldsymbol{v}'). \tag{124}$$

We say that $\boldsymbol{v}$ and $\boldsymbol{v}'$ are *DWS-separable* if $\boldsymbol{v} \not\sim_{\text{DWS}} \boldsymbol{v}'$.

Our next theorem characterizes the relationship between the three equivalence relations introduced above.

**Theorem E.4** (Hierarchy of equivalence relations). *Let $\mathcal{V}$ be a weight space. Viewed as subsets of $\mathcal{V} \times \mathcal{V}$, the equivalence relations defined above satisfy the strict inclusions*

$$\sim_G \subsetneq \sim_{\text{DWS}} \subsetneq \sim_{\text{func}}. \tag{125}$$

To prove Theorem E.4, we decompose the argument into a sequence of propositions establishing each inclusion and the strictness of the containments.

**Proposition E.5.** *Let $\mathcal{V}$ be a weight space. For all $\boldsymbol{v}, \boldsymbol{v}' \in \mathcal{V}$,*

$$\boldsymbol{v} \sim_G \boldsymbol{v}' \implies \boldsymbol{v} \sim_{\mathrm{DWS}} \boldsymbol{v}'. \tag{126}$$

*Proof.* The claim follows directly from the equivariance of DWSNets. If $\boldsymbol{v} \sim_G \boldsymbol{v}'$, then there exists $g \in G$ such that $\boldsymbol{v}' = \rho(g)\,\boldsymbol{v}$. As shown in (Navon et al., 2023), DWS networks are $G$-equivariant; that is, for any invariant DWS network $\Phi : \mathcal{V} \to \mathbb{R}^n$ it holds that

$$\Phi(\boldsymbol{v}') = \Phi(\rho(g)\,\boldsymbol{v}) = \Phi(\boldsymbol{v}). \tag{127}$$

Therefore, $\boldsymbol{v} \sim_{\mathrm{DWS}} \boldsymbol{v}'$, completing the proof. $\square$

**Proposition E.6.** *Let $\mathcal{V}$ be a weight space. For all $\boldsymbol{v}, \boldsymbol{v}' \in \mathcal{V}$,*

$$\boldsymbol{v} \sim_{\mathrm{DWS}} \boldsymbol{v}' \implies \boldsymbol{v} \sim_{\mathrm{func}} \boldsymbol{v}'. \tag{128}$$

*Proof.* We prove the contrapositive. Suppose that $\boldsymbol{v} \not\sim_{\mathrm{func}} \boldsymbol{v}'$. Then there exists $\boldsymbol{x}^* \in \mathbb{R}^{d_0}$ such that

$$f_{\boldsymbol{v}}(\boldsymbol{x}^*) \neq f_{\boldsymbol{v}'}(\boldsymbol{x}^*). \tag{129}$$

As shown in (Navon et al., 2023), the DWS architecture can be extended to operate on weight–input pairs, yielding networks of the form

$$\Phi : \mathcal{V} \oplus \mathbb{R}^{d_0} \to \mathbb{R}^n, \tag{130}$$

where the group $G$ acts trivially on the input space $\mathbb{R}^{d_0}$. Concretely, the input vector $\boldsymbol{x} \in \mathbb{R}^{d_0}$ is broadcast to a matrix $\boldsymbol{X} \in \mathbb{R}^{d_0 \times d_1}$ and concatenated with the first-layer weights, producing an augmented representation in $\mathcal{V}^2$; the network then proceeds as usual, by composing pointwise nonlinearities with affine $G$-equivariant maps of the form $\mathcal{V}^c \to \mathcal{V}^{c'}$. Moreover, (Navon et al., 2023) show that for any compact set $K \subset \mathcal{V} \oplus \mathbb{R}^{d_0}$, there exists such an extended DWS network $\Phi^*$ satisfying

$$\Phi^*(\boldsymbol{v}, \boldsymbol{x}) = f_{\boldsymbol{v}}(\boldsymbol{x}) \quad \text{for all } (\boldsymbol{v}, \boldsymbol{x}) \in K. \tag{131}$$

Fix a compact set $K \subseteq \mathcal{V}$ containing both $\boldsymbol{v}$ and $\boldsymbol{v}'$, and consider the compact set $K^* = K \times \{\boldsymbol{x}^*\}$. Let $\Phi^*$ be an extended DWS network realizing $f_{\boldsymbol{v}}(\boldsymbol{x})$ on $K^*$, and define

$$\Phi(\cdot) \coloneqq \Phi^*(\cdot, \boldsymbol{x}^*) : \mathcal{V} \to \mathbb{R}^m. \tag{132}$$

By construction, $\Phi$ is a composition of pointwise nonlinearities and affine $G$-equivariant maps acting on $\mathcal{V}$, and therefore constitutes a standard DWS network.

Consequently,

$$\Phi(\boldsymbol{v}) = f_{\boldsymbol{v}}(\boldsymbol{x}^*) \neq f_{\boldsymbol{v}'}(\boldsymbol{x}^*) = \Phi(\boldsymbol{v}'), \tag{133}$$

showing that $\boldsymbol{v}$ and $\boldsymbol{v}'$ are separable by a DWS network. Hence, $\boldsymbol{v} \not\sim_{\mathrm{DWS}} \boldsymbol{v}'$, which establishes the contrapositive and completes the proof. $\square$

**Proposition E.7.** *There exists a weight space $\mathcal{V}$ and weights $\boldsymbol{v}, \boldsymbol{v}' \in \mathcal{V}$ such that*

$$\boldsymbol{v} \sim_{\mathrm{func}} \boldsymbol{v}', \qquad \boldsymbol{v} \not\sim_{\mathrm{DWS}} \boldsymbol{v}'. \tag{134}$$

*Proof.* We exhibit a concrete counterexample. Let $A = ((1, 2, 1), \mathrm{ReLU})$ and let $\mathcal{V} = \mathcal{V}_A$. Define $\boldsymbol{v} = (\boldsymbol{W}_1, \boldsymbol{b}_1, \boldsymbol{W}_2, \boldsymbol{b}_2) \in \mathcal{V}$ by

$$\boldsymbol{W}_1 = \begin{pmatrix} 1 \\ 0 \end{pmatrix}, \qquad \boldsymbol{b}_1 = \begin{pmatrix} 0 \\ 0 \end{pmatrix}, \qquad \boldsymbol{W}_2 = \begin{pmatrix} 1 & 0 \end{pmatrix}, \qquad \boldsymbol{b}_2 = 0. \tag{135}$$

A direct computation shows that the function realized by $\boldsymbol{v}$ satisfies

$$f_{\boldsymbol{v}}(x, y) = \mathrm{ReLU}(x). \tag{136}$$

Fix a scaling factor $\lambda > 0$ with $\lambda \neq 1$, and define $\boldsymbol{v}' = (\boldsymbol{W}_1', \boldsymbol{b}_1', \boldsymbol{W}_2', \boldsymbol{b}_2') \in \mathcal{V}$ by

$$\boldsymbol{W}_1' = \begin{pmatrix} \lambda \\ 0 \end{pmatrix}, \qquad \boldsymbol{b}_1' = \begin{pmatrix} 0 \\ 0 \end{pmatrix}, \qquad \boldsymbol{W}_2' = \begin{pmatrix} \lambda^{-1} & 0 \end{pmatrix}, \qquad \boldsymbol{b}_2' = 0. \tag{137}$$

By the positive homogeneity of the ReLU activation, the rescaling of the first layer by $\lambda$ and of the second layer by $\lambda^{-1}$ leaves the realized function unchanged. Consequently,

$$f_{\boldsymbol{v}'}(x, y) = \mathrm{ReLU}(x) = f_{\boldsymbol{v}}(x, y), \tag{138}$$

and therefore $\boldsymbol{v} \sim_{\mathrm{func}} \boldsymbol{v}'$.

We now show that $\boldsymbol{v}$ and $\boldsymbol{v}'$ are not DWS-equivalent. Define a map $\Phi : \mathcal{V} \to \mathbb{R}$ by

$$\Phi(\boldsymbol{v}) \coloneqq \boldsymbol{W}_1^\top \mathbf{1}, \tag{139}$$

where $\mathbf{1}$ denotes the all-ones vector of appropriate dimension. This map is linear and $G$-invariant, and hence constitutes a valid (single-layer) DWS network. Evaluating $\Phi$ at $\boldsymbol{v}$ and $\boldsymbol{v}'$ yields

$$\Phi(\boldsymbol{v}) = 1, \qquad \Phi(\boldsymbol{v}') = \lambda. \tag{140}$$

Since $\lambda \neq 1$, it follows that $\Phi(\boldsymbol{v}) \neq \Phi(\boldsymbol{v}')$, and thus $\boldsymbol{v} \not\sim_{\mathrm{DWS}} \boldsymbol{v}'$. $\qquad\square$

**Proposition E.8.** *There exists a weight space $\mathcal{V}$ and weights $\boldsymbol{v}, \boldsymbol{v}' \in \mathcal{V}$ such that*

$$\boldsymbol{v} \sim_{\mathrm{DWS}} \boldsymbol{v}', \qquad \boldsymbol{v} \not\sim_G \boldsymbol{v}'. \tag{141}$$

*Proof.* Let $A = ((1, 4, 4, 1), \mathrm{ReLU})$ and let $\mathcal{V} = \mathcal{V}_A$. Define $\boldsymbol{v}, \boldsymbol{v}' \in \mathcal{V}$ by $\boldsymbol{v} = (\boldsymbol{W}_1, \boldsymbol{b}_1, \boldsymbol{W}_2, \boldsymbol{b}_2, \boldsymbol{W}_3, \boldsymbol{b}_3)$ and $\boldsymbol{v}' = (\boldsymbol{W}_1', \boldsymbol{b}_1', \boldsymbol{W}_2', \boldsymbol{b}_2', \boldsymbol{W}_3', \boldsymbol{b}_3')$ where

$$\boldsymbol{W}_1 = \boldsymbol{W}_1' = \mathbf{1}_{4 \times 1}, \qquad \boldsymbol{b}_1 = \boldsymbol{b}_1' = 0, \qquad \boldsymbol{W}_3 = \boldsymbol{W}_3' = \mathbf{1}_{1 \times 4}, \qquad \boldsymbol{b}_2 = \boldsymbol{b}_2' = 0, \qquad \boldsymbol{b}_3 = \boldsymbol{b}_3' = 0. \tag{142}$$

Here, $\mathbf{1}_{4 \times 1}$ denotes the $4 \times 1$ column vector of ones and $\mathbf{1}_{1 \times 4}$ denotes the $1 \times 4$ row vector of ones. Finally, define the second-layer weight matrices by

$$\boldsymbol{W}_2 = \begin{pmatrix} 1 & 1 & 0 & 0 \\ 0 & 1 & 1 & 0 \\ 0 & 0 & 1 & 1 \\ 1 & 0 & 0 & 1 \end{pmatrix}, \qquad \boldsymbol{W}_2' = \begin{pmatrix} 1 & 1 & 0 & 0 \\ 1 & 1 & 0 & 0 \\ 0 & 0 & 1 & 1 \\ 0 & 0 & 1 & 1 \end{pmatrix}. \tag{143}$$

First, suppose toward a contradiction that $\boldsymbol{v} \sim_G \boldsymbol{v}'$. By Definition 3.2, this implies that there exist permutation matrices $\boldsymbol{P}_1, \boldsymbol{P}_2 \in \mathbb{R}^{4 \times 4}$ such that

$$\boldsymbol{W}_2 = \boldsymbol{P}_2^\top \boldsymbol{W}_2' \boldsymbol{P}_1. \tag{144}$$

However, $\boldsymbol{W}_2$ has rank 3, whereas $\boldsymbol{W}_2'$ has rank 2. Since left and right multiplication by permutation matrices preserves matrix rank, the above equality cannot hold. We conclude that $\boldsymbol{v} \not\sim_G \boldsymbol{v}'$.

We now show that $\boldsymbol{v} \sim_{\mathrm{DWS}} \boldsymbol{v}'$, i.e., that $\Phi(\boldsymbol{v}) = \Phi(\boldsymbol{v}')$ for every DWS network $\Phi$. By Theorem 5.2, it suffices to prove that $\Psi(\boldsymbol{v}) = \Psi(\boldsymbol{v}')$ for every NG-GNN model $\Psi$.

Recall that an NG-GNN applies a message-passing graph neural network to the computational graph induced by the underlying MLP architecture (see (Kofinas et al., 2024) for details). A direct computation shows that the computational graphs induced by $\boldsymbol{v}$ and $\boldsymbol{v}'$ are indistinguishable by the Weisfeiler–Lehman test (see Figure 3 for an illustration). As shown in (Morris et al., 2019), this implies that no message-passing neural network can separate the two graphs. Consequently, $\Psi(\boldsymbol{v}) = \Psi(\boldsymbol{v}')$ for all NG-GNN models $\Psi$, and therefore $\boldsymbol{v} \sim_{\mathrm{DWS}} \boldsymbol{v}'$, completing the proof.

$\qquad\square$

### E.2. Universal approximation of function-space functionals

In this section, we leverage the analysis of the equivalence relations $\sim_{\text{func}}$ and $\sim_{\text{DWS}}$ developed in Appendix E.1 to prove Theorem 6.1, thus establishing the universality of weight-space networks with respect to the class of continuous function-space functionals. For completeness, we restate the theorem here.

**Theorem E.9.** *Let $K \subseteq \mathcal{V}$ be compact. Then any function-space functional $\Psi : \mathcal{C}(X, \mathbb{R}^{d_L}) \to \mathbb{R}^n$ can be approximated on $K$ by permutation-invariant weight-space networks.*

*Proof.* Let $A = (\boldsymbol{d}, \sigma)$ be an MLP architecture with corresponding weight space $\mathcal{V}$ and let $\Psi : \mathcal{C}(X, \mathbb{R}^{d_L}) \to \mathbb{R}^n$ be continuous. From Theorem 5.2, it suffices to show that, for any compact set $K \subseteq \mathcal{V}$, the map $\Psi^*$ defined by

$$\Psi^*(\boldsymbol{v}) = \Psi(f_{\boldsymbol{v}}) \tag{145}$$

can be uniformly approximated on $K$ by some DWS network.

First, $\Psi^*$ is continuous because the realization map is continuous (Proposition C.2) and $\Psi$ is continuous.

Second, by Proposition E.6, for any pair of weights $\boldsymbol{v}, \boldsymbol{v}' \in \mathcal{V}$ such that $f_{\boldsymbol{v}} \neq f_{\boldsymbol{v}'}$, there exists a DWS network $\Phi$ satisfying $\Phi(\boldsymbol{v}) \neq \Phi(\boldsymbol{v}')$. Importantly, the depth of such a separating DWS network depends only on the architecture underlying $\mathcal{V}$, and not on the particular choice of $\boldsymbol{v}, \boldsymbol{v}'$. Thus, since the map $\Psi^*$ is constant on functional-equivalence classes, it is also constant on DWS-equivalence classes, even when restricting to DWS networks of bounded depth.

It was shown in (Pacini et al., 2025a) (Theorem 1) that any continuous function on a compact set that is constant on the equivalence classes induced by equivariant affine models of bounded depth can be uniformly approximated by such models. Applying this result, we conclude that $\Psi^*$ can be approximated arbitrarily well on $K$ by a DWS network, completing the proof. □

### E.3. Universal approximation of permutation-invariant functionals

In this section, we analyze the approximation power of weight-space networks with respect to the class of continuous $G$-invariant maps. We prove Proposition 6.2 and Theorem 6.3, showing that, while weight-space networks are not universal approximators for this class in full generality, they are universal assuming the input is in general position. For the remainder of this section, we focus on DWS networks, since Theorem 5.2 shows that all other prominent architectures are expressively equivalent. For convenience, we restate the theorem below.

**Proposition E.10.** *There exists a compact set $K \subseteq \mathcal{V}$ and a permutation-invariant map $\Psi : K \to \mathbb{R}$ that cannot be approximated on $K$ by invariant weight-space models.*

*Proof.* By Proposition E.8, there exists a weight space $\mathcal{V}$ and a pair of weights $\boldsymbol{v}, \boldsymbol{v}' \in \mathcal{V}$ such that $\boldsymbol{v} \not\sim_G \boldsymbol{v}'$ while $\boldsymbol{v} \sim_{\text{DWS}} \boldsymbol{v}'$. Since $\boldsymbol{v}$ and $\boldsymbol{v}'$ are not $G$-equivalent, their orbits

$$\text{orbit}(\boldsymbol{v}) := \{\rho(g)\,\boldsymbol{v} \mid g \in G\}, \qquad \text{orbit}(\boldsymbol{v}') := \{\rho(g)\,\boldsymbol{v}' \mid g \in G\} \tag{146}$$

are disjoint subsets of $\mathcal{V}$.

Because $\mathcal{V}$ is a finite-dimensional Euclidean space and $\text{orbit}(\boldsymbol{v})$ and $\text{orbit}(\boldsymbol{v}')$ are disjoint compact sets, there exists a continuous function $\Psi : \mathcal{V} \to \mathbb{R}$ such that

$$\Psi(\boldsymbol{u}) > 1 \quad \text{for all } \boldsymbol{u} \in \text{orbit}(\boldsymbol{v}), \qquad \Psi(\boldsymbol{u}) < 0 \quad \text{for all } \boldsymbol{u} \in \text{orbit}(\boldsymbol{v}'). \tag{147}$$

Define the symmetrized function $\Psi^* : \mathcal{V} \to \mathbb{R}$ by

$$\Psi^*(\boldsymbol{u}) := \sum_{g \in G} \Psi(\rho(g)\,\boldsymbol{u}). \tag{148}$$

The function $\Psi^*$ is continuous and $G$-invariant by construction. Moreover, it satisfies

$$\Psi^*(\boldsymbol{u}) > 1 \quad \text{for all } \boldsymbol{u} \in \text{orbit}(\boldsymbol{v}), \qquad \Psi^*(\boldsymbol{u}) < 0 \quad \text{for all } \boldsymbol{u} \in \text{orbit}(\boldsymbol{v}'). \tag{149}$$

In particular,

$$|\Psi^*(\boldsymbol{v}) - \Psi^*(\boldsymbol{v}')| > 1. \tag{150}$$

On the other hand, since $\boldsymbol{v} \sim_{\text{DWS}} \boldsymbol{v}'$, every DWS network $\Phi$ satisfies $\Phi(\boldsymbol{v}) = \Phi(\boldsymbol{v}')$. It follows that no DWS network can approximate the $G$-invariant function $\Psi^*$ on $\{\boldsymbol{v}, \boldsymbol{v}'\}$ with uniform error smaller than $1/2$. This completes the proof. $\qquad\square$

**Theorem E.11.** *Let $K \subseteq \mathcal{V} \setminus \mathcal{E}$ be compact. Then any permutation-invariant map $\Psi : \mathcal{V} \to \mathbb{R}^n$ can be approximated on $K$ by invariant weight-space models.*

**Remark:** The argument given here is conceptually different from the proof sketch in Section 6.2. The sketch was included primarily for intuition: it highlights the key ideas and helps explain why the statement should be true at a high level. However, a cleaner route is to derive the claim as a direct consequence of Theorem 7.4, whose complete proof is provided in Appendix F.1. For this reason, we treat Section 6.2 mainly as intuition-building, and rely on Theorem 7.4 for the formal justification.

*Proof.* The proof uses a reduction to the equivariant setting. We rely on Theorem 7.4, proved in the subsequent section of the appendix, which states that for any weight space $\mathcal{V}$, DWS networks are universal approximators for the class of continuous $G$-*equivariant* maps assuming the input is in general position. Let $\Psi : \mathcal{V} \to \mathbb{R}^n$ be a continuous $G$-invariant function. We define an associated map

$$\Psi^* : \mathcal{V} \to \mathcal{V}^n \tag{151}$$

by broadcasting the vector value $\Psi(\boldsymbol{v})$ uniformly across all weight and bias terms of $\mathcal{V}^n$. Specifically, for $\boldsymbol{v} = (\boldsymbol{W}_1, \boldsymbol{b}_1, \ldots, \boldsymbol{W}_L, \boldsymbol{b}_L)$, we define $\Psi^*(\boldsymbol{v}) = (\boldsymbol{W}_1^*, \boldsymbol{b}_1^*, \ldots, \boldsymbol{W}_L^*, \boldsymbol{b}_L^*)$ by setting, for each $\ell = 1, \ldots, L$,

$$(\boldsymbol{W}_\ell^*)_{i,j,:} := \Psi(\boldsymbol{v}), \qquad (\boldsymbol{b}_\ell^*)_{i,:} := \Psi(\boldsymbol{v}), \tag{152}$$

for all valid indices $i, j$. That is, every coordinate of $\Psi^*(\boldsymbol{v})$ is equal to $\Psi(\boldsymbol{v})$.

Since $\Psi$ is $G$-invariant, for every $g \in G$ and $\boldsymbol{v} \in \mathcal{V}$ we have

$$\Psi^*(\rho(g)\boldsymbol{v}) = \Psi^*(\boldsymbol{v}) = \rho(g)\,\Psi^*(\boldsymbol{v}), \tag{153}$$

where the final equality follows from the fact that the action of $G$ permutes coordinates along which $\Psi^*(\boldsymbol{v})$ is constant. Thus, $\Psi^*$ is a continuous $G$-equivariant map.

By Theorem 7.4, $\Psi^*$ can be approximated by a DWS network $\Phi^*$ over any compact set $K \subseteq \mathcal{V} \setminus \mathcal{E}$. Finally, composing $\Phi^*$ with a fixed linear $G$-invariant pooling operator, which takes the mean over all weight and bias terms, we get a DWS network whose output uniformly approximates $\Psi$. This completes the proof. $\qquad\square$

# F. Expressive power of permutation-equivariant weight-space networks

## F.1. Universal approximation of permutation-equivariant operators

In this section, we prove Proposition 7.3 and Theorem 7.4 showing that, while weight-space networks are not universal approximators for this class in full generality, they are universal for inputs in general position. For the remainder of this section, we focus on DWS networks, since Theorem 5.2 shows that all other prominent architectures are expressively equivalent. For convenience, we restate these results below.

**Proposition F.1.** *There exists a compact set $K \subseteq \mathcal{V}$ and a permutation-equivariant map $\Psi : K \to \mathcal{V}$ that cannot be approximated on $K$ by permutation-equivariant[5] weight-space models.*

*Proof.* We rely on Proposition 6.2, proved in the previous section of the appendix, which states that there exists a weight space $\mathcal{V}$ and a continuous $G$-invariant map $\Psi : \mathcal{V} \to \mathbb{R}$ that cannot be approximated by any DWS network.

We define an associated map

$$\Psi^* : \mathcal{V} \to \mathcal{V} \tag{154}$$

---

[5]In fact, the same argument shows that the statement extends to scale- and permutation-equivariant networks as well.

by broadcasting the scalar value $\Psi(\boldsymbol{v})$ uniformly across all coordinates of $\mathcal{V}$. Specifically, for $\boldsymbol{v} = (\boldsymbol{W}_1, \boldsymbol{b}_1, \ldots, \boldsymbol{W}_L, \boldsymbol{b}_L)$, we define $\Psi^*(\boldsymbol{v}) = (\boldsymbol{W}_1^*, \boldsymbol{b}_1^*, \ldots, \boldsymbol{W}_L^*, \boldsymbol{b}_L^*)$ by setting, for each $\ell = 1, \ldots, L$,

$$(\boldsymbol{W}_\ell^*)_{i,j} := \Psi(\boldsymbol{v}), \qquad (\boldsymbol{b}_\ell^*)_i := \Psi(\boldsymbol{v}), \tag{155}$$

for all valid indices $i, j$. That is, every coordinate of $\Psi^*(\boldsymbol{v})$ is equal to $\Psi(\boldsymbol{v})$.

Since $\Psi$ is $G$-invariant, for every $g \in G$ and $\boldsymbol{v} \in \mathcal{V}$ we have

$$\Psi^*(\rho(g)\boldsymbol{v}) = \Psi^*(\boldsymbol{v}) = \rho(g)\,\Psi^*(\boldsymbol{v}), \tag{156}$$

where the final equality follows from the fact that the action of $G$ permutes coordinates along which $\Psi^*(\boldsymbol{v})$ is constant. Thus, $\Psi^*$ is a continuous $G$-equivariant map.

Assuming there existed a DWS model which was able to approximate $\Psi^*$, by composing that model with a fixed linear $G$-invariant pooling operator, which takes the mean over all output coordinates, we get a DWS network whose output uniformly approximates $\Psi$, which is a contradiction. Thus we are unable to approximate the equivariant continuous map $\Psi^*$ with any DWS model, completing the proof. $\qquad\square$

**Theorem F.2.** *Let $K \subseteq \mathcal{V} \setminus \mathcal{E}$ be compact. Then any permutation-equivariant map $\Psi : \mathcal{V} \to \mathcal{V}$ can be approximated on $K$ by equivariant weight-space models.*

The remainder of this section is devoted to proving the theorem above. The proof proceeds in several stages. First, we introduce a neuron-identification map and show that, on compact sets that exclude $\mathcal{E}$ (a set contained in a finite union of lower-dimensional submanifolds; see Definition 3.4), this map can be approximated arbitrarily well by DWS networks. Next, we use neuron identification to construct an invariant canonization map and show that it too can be approximated by DWS networks. Finally, we prove that any permutation-equivariant map admits a decomposition into the composition of this canonization map with a pointwise function applied in parallel to the resulting feature representation concatenated with the original weights, implying that DWS networks can approximate any such equivariant map.

**Neuron identification map.**

**Definition F.3** (Neuron-identification map)**.** Let $A = (\boldsymbol{d}, \sigma)$ be an architecture with $L$ layers, let $\mathcal{V}_A$ be the corresponding weight-space, and let $\mathcal{E}_A$ denote the corresponding exclusion set (see Definition 3.4). We define the *neuron-identification map*

$$\mathrm{NI} : \mathcal{V}_A \setminus \mathcal{E}_A \longrightarrow \mathcal{V}_A^5 \tag{157}$$

as follows.

Given parameters $\boldsymbol{v} = (\boldsymbol{W}_1, \boldsymbol{b}_1, \ldots, \boldsymbol{W}_L, \boldsymbol{b}_L) \in \mathcal{V}_A \setminus \mathcal{E}_A$, we define

$$\mathrm{NI}(\boldsymbol{v}) := (\boldsymbol{W}_1', \boldsymbol{b}_1', \ldots, \boldsymbol{W}_L', \boldsymbol{b}_L'), \tag{158}$$

where

**Biases.** For $\ell = 1, \ldots, L$ and all $i = 1, \ldots, d_\ell$, we define

$$(\boldsymbol{b}_\ell')_i := \big((\boldsymbol{b}_\ell)_i,\ \ell, 1, 0, \eta_\ell^b(i)\big), \tag{159}$$

where

$$\eta_\ell^b(i) := \begin{cases} \mathrm{rank}_\ell(i), & 1 \le \ell < L, \\ i, & \ell = L. \end{cases} \tag{160}$$

Here $\mathrm{rank}_\ell(i)$ denotes the rank of $(\boldsymbol{b}_\ell)_i$ among the entries of $\boldsymbol{b}_\ell$, i.e.,

$$\mathrm{rank}_\ell(i) := \big|\{\, j \in [d_\ell] : (\boldsymbol{b}_\ell)_j < (\boldsymbol{b}_\ell)_i \,\}\big|. \tag{161}$$

This quantity is well defined on $\mathcal{V}_A \setminus \mathcal{E}_A$, since all bias entries within each layer are distinct.

**Weights.** For $\ell = 1, \ldots, L$ and all valid indices $i, j$, we define

$$(\boldsymbol{W}'_\ell)_{i,j} := \left((\boldsymbol{W}_\ell)_{i,j}, \ell, 0, \eta^w_\ell(i), \bar{\eta}^b_\ell(j)\right), \tag{162}$$

where

$$\eta^w_\ell(i) := \begin{cases} \eta^b_{\ell-1}(i), & 1 < \ell \leq L, \\ i, & \ell = 1. \end{cases} \tag{163}$$

**Lemma F.4** (Approximation of the neuron-identification map by DWS networks)**.** *Let $\mathcal{V}$ be a weight-space with corresponding exclusion set $\mathcal{E}$, and let $K \subseteq \mathcal{V} \setminus \mathcal{E}$ be compact. Then, for every $\epsilon > 0$, there exists a DWS network $\Phi : \mathcal{V} \to \mathcal{V}^5$ such that*

$$\sup_{\boldsymbol{v} \in K} \left\| \Phi(\boldsymbol{v}) - \mathrm{NI}(\boldsymbol{v}) \right\| < \epsilon. \tag{164}$$

*Proof.* We first define an affine $G$-equivariant map

$$L : \mathcal{V}_A \setminus \mathcal{E}_A \longrightarrow \mathcal{V}_A^5. \tag{165}$$

For $\boldsymbol{v} = (\boldsymbol{W}_1, \boldsymbol{b}_1, \ldots, \boldsymbol{W}_L, \boldsymbol{b}_L)$ we set

$$L(\boldsymbol{v}) := (\boldsymbol{W}_1^{(1)}, \boldsymbol{b}_1^{(1)}, \ldots, \boldsymbol{W}_L^{(1)}, \boldsymbol{b}_L^{(1)}), \tag{166}$$

with components defined as follows.

For each layer $\ell = 1, \ldots, L$ and neuron index $i = 1, \ldots, d_\ell$, define

$$(\boldsymbol{b}_\ell^{(1)})_i := \left((\boldsymbol{b}_\ell)_i, \ell, 1, 0, 0\right), \tag{167}$$

Similarly, for each $\ell = 1, \ldots, L$ and all valid indices $i, j$, define

$$(\boldsymbol{W}_\ell^{(1)})_{i,j} := \left((\boldsymbol{W}_\ell)_{i,j}, \ell, 0, 0, 0\right). \tag{168}$$

By construction, $L$ is affine and $G$-equivariant. For each layer $\ell$, define a bias-to-bias map $\theta_\ell : \mathcal{B}_\ell^5 \to \mathcal{B}_\ell^5$ by

$$(\theta_\ell(\boldsymbol{b}_\ell^{(1)}))_i := \begin{cases} (0, 0, 0, 0, \mathrm{rank}_\ell(i)), & 1 \leq \ell < L, \\ (0, 0, 0, 0, i), & \ell = L, \end{cases} \tag{169}$$

The ranking map $\mathrm{rank}_\ell$ is equivariant with respect to neuron permutations within layer $\ell$, and hence so is $\theta_\ell$. Moreover, since $K \subseteq \mathcal{V}_A \setminus \mathcal{E}_A$, all bias values within each layer are pairwise distinct on $K$. Consequently, $\mathrm{rank}_\ell(i)$ is locally constant in a neighborhood of every $\boldsymbol{v} \in K$, and therefore $\mathrm{rank}_\ell$ and consequently $\theta_\ell$ are continuous on $K$.

As shown in (Navon et al., 2023), any affine equivariant map on weight-space decomposes into bias-to-bias, bias-to-weight, and weight-to-weight components. In particular, the bias-to-bias components contain all affine equivariant set functions acting independently on the bias vectors of each layer. Such affine functions coincide with the layers of the DeepSets architectures proposed in (Zaheer et al., 2017), which were shown to be universal approximators of continuous permutation-equivariant set functions on compact domains (Segol & Lipman, 2019). Therefore, by appropriately choosing only the bias-to-bias components, setting all weight-to-weight components to the identity, and all weight-to-bias components to zero, we obtain intermediate representations

$$(\boldsymbol{W}_1^{(m)}, \boldsymbol{b}_1^{(m)}, \ldots, \boldsymbol{W}_L^{(m)}, \boldsymbol{b}_L^{(m)}) \tag{170}$$

satisfying $\forall \ell \in [L]$

$$\boldsymbol{W}_\ell^{(m)} = \boldsymbol{W}_\ell^{(1)}, \tag{171}$$

and

$$\sup_{\boldsymbol{v} \in K} \|(\boldsymbol{b}_\ell^{(m)})_i - (\boldsymbol{b}_\ell^{(1)})_i - \theta_\ell(\boldsymbol{b}_\ell^{(1)})_i\| < \epsilon. \tag{172}$$

Equivalently,

$$\sup_{\boldsymbol{v} \in K} \|(\boldsymbol{b}_\ell^{(m)})_i - \big((\boldsymbol{b}_\ell)_i,\ \ell,\ 1,\ 0,\ \eta_\ell^b(i)\big)\| < \epsilon. \tag{173}$$

Finally, note that concatenating the bias features $(\boldsymbol{b}_{\ell-1})_{i,:}$ to the feature dimension of weights $(\boldsymbol{W}_\ell)i,:,:$, and similarly concatenating $(\boldsymbol{b}_\ell)_{j,:}$ to $(\boldsymbol{W}_{\ell-1})_{:,j,:}$, are affine $G$-equivariant weight-to-bias maps. Applying one final equivariant linear layer therefore can yield a DWS network $\Phi$ with final representation $(\boldsymbol{W}_1^{(m+1)}, \boldsymbol{b}_1^{(m+1)}, \ldots, \boldsymbol{W}_L^{(m+1)}, \boldsymbol{b}_L^{(m+1)})$ which keeps the bias terms the same

$$\boldsymbol{b}_\ell^{(m+1)} = \boldsymbol{b}_\ell^{(m)} \qquad \forall \ell, \tag{174}$$

and augments the weight features by concatenating the relevant neuron identifiers. Specifically, for each layer $\ell$ and all valid indices $i, j, k$,

$$(\boldsymbol{W}_\ell^{(m+1)})_{i,j,k} = \begin{cases} (\boldsymbol{W}_\ell^{(m)})_{i,j,k}, & k = 1, 2, 3, \\ (\boldsymbol{b}_{\ell-1}^{(m)})_{i,5}, & k = 4,\ \ell \neq 1, \\ i, & k = 4,\ \ell = 1, \\ (\boldsymbol{b}_\ell^{(m)})_{j,5}, & k = 5. \end{cases} \tag{175}$$

The first three channels preserve the original weight features, while channels $k = 4$ and $k = 5$ encode the identifiers of the source and target neurons, respectively. All operations are affine and $G$-equivariant functions of the previous representation, as they act uniformly across neurons and respect the permutation structure within each layer.

Combining this construction with the approximation guarantees from the previous steps yields a DWS network $\Phi$ satisfying

$$\sup_{\boldsymbol{v} \in K} \big\|\Phi(\boldsymbol{v}) - \mathrm{NI}(\boldsymbol{v})\big\| < \epsilon. \tag{176}$$

This completes the proof. $\qquad\square$

**Weight-space canonization**

**Definition F.5** (weight-space canonization map). Let $A = (\boldsymbol{d}, \sigma)$ be an architecture with $L$ layers, and let $\mathcal{V}_A, \mathcal{E}_A$ be the corresponding weight-space and exclusion set respectively. We define the *weight-space canonization map*

$$\mathrm{canon} : \mathcal{V}_A \setminus \mathcal{E}_A \longrightarrow \mathcal{V}_A^5 \tag{177}$$

by

$$\mathrm{canon}(\boldsymbol{v}) := \tilde{\mathrm{Flat}}^{-1}\big(\tilde{\mathrm{Sort}}\big(\tilde{\mathrm{Flat}}(\mathrm{NI}(\boldsymbol{v}))\big)\big). \tag{178}$$

Here, $\tilde{\mathrm{Flat}}$ denotes the vectorization operator that flattens all weights and biases in $\mathcal{V}_A^5$ into a matrix in $\mathbb{R}^{n \times 5}$ (see the main paper for the definition of $\tilde{\mathrm{Flat}}$), where

$$n := \sum_{\ell=1}^{L} d_\ell (1 + d_{\ell-1}), \tag{179}$$

and $\tilde{\mathrm{Sort}}$ denotes lexicographic sorting of the rows of $\mathbb{R}^{n \times 5}$ with respect to coordinates 2 through 5, i.e., ignoring the first coordinate.

**Lemma F.6** (Properties of the canonization map). *Let $\mathcal{V}_A$ be a weight-space with corresponding exclusion set $\mathcal{E}_A$, and let* $\mathrm{canon} : \mathcal{V}_A \setminus \mathcal{E}_A \to \mathcal{V}_A^5$ *be the weight-space canonization map. Then,* $\mathrm{canon}$ *is continuous, and for every $\boldsymbol{v} \in \mathcal{V}_A \setminus \mathcal{E}_A$, the following hold:*

1. $\mathrm{canon}(\boldsymbol{v})_1 \in \mathrm{orbit}(\boldsymbol{v})$.

2. $\mathrm{canon}$ *is indeed a canonization map (Definition B.4). That is,* $\mathrm{canon}(\boldsymbol{v}) = \mathrm{canon}(\rho(g)\boldsymbol{v}) \quad \forall g \in G$.

*Proof.* Let $\boldsymbol{v} = (\boldsymbol{W}_1, \boldsymbol{b}_1, \ldots, \boldsymbol{W}_L, \boldsymbol{b}_L)$ and write

$$\text{NI}(\boldsymbol{v}) = (\boldsymbol{W}_1', \boldsymbol{b}_1', \ldots, \boldsymbol{W}_L', \boldsymbol{b}_L'). \tag{180}$$

Recall that in $\text{NI}(\boldsymbol{v})$ the feature coordinates are defined as follows: the second coordinate of $(\boldsymbol{W}_\ell', \boldsymbol{b}_\ell')$ encodes the layer index $\ell$, the third coordinate distinguishes weights from biases (with value 0 for weights and 1 for biases), and the fifth coordinate of $\boldsymbol{b}_\ell'$ encodes the neuron identifier within layer $\ell$. First, as NI is continuous, canon is a composition of continuous maps and so it is continuous.

Additionally, by construction, lexicographic sorting via $\widetilde{\text{Sort}}$ first groups terms by layer index, then places all weight terms of a given layer before the bias terms of that layer. As a result, canon acts only by permuting weight terms within each layer and bias terms within each layer, without mixing across layers or types.

Within a fixed layer $\ell$, the bias vectors $(\boldsymbol{b}_\ell')_{:,5}$ enforce a sorting of neurons according to the standard ordering of the bias values $\boldsymbol{b}_\ell$. Let $\tau_\ell$ denote the permutation that sorts $\boldsymbol{b}_\ell$ in ascending order, and set $g = (\tau_1, \ldots, \tau_L) \in G$.

The fourth and fifth feature coordinates of $\boldsymbol{W}_\ell'$ ensure that weight entries are permuted consistently with the corresponding neuron permutations. As a result, the first feature coordinate of $\text{canon}(\boldsymbol{v})$ satisfies

$$\begin{aligned}
(\boldsymbol{W}_1')_{:,:,1} &= \boldsymbol{P}_{\tau_1}^\top \boldsymbol{W}_1, & (\boldsymbol{b}_1')_{:,1} &= \boldsymbol{P}_{\tau_1}^\top \boldsymbol{b}_1, \\
(\boldsymbol{W}_\ell')_{:,:,1} &= \boldsymbol{P}_{\tau_\ell}^\top \boldsymbol{W}_\ell \boldsymbol{P}_{\tau_{\ell-1}}, & (\boldsymbol{b}_\ell')_{:,1} &= \boldsymbol{P}_{\tau_\ell}^\top \boldsymbol{b}_\ell, \quad \ell = 2, \ldots, L-1, \\
(\boldsymbol{W}_L')_{:,:,1} &= \boldsymbol{W}_L \boldsymbol{P}_{\tau_{L-1}}, & (\boldsymbol{b}_L')_{:,1} &= \boldsymbol{b}_L.
\end{aligned}$$

Therefore, restricting $\text{canon}(\boldsymbol{v})$ to its first feature coordinate yields a representative of the $G$-orbit of $\boldsymbol{v}$, proving the first claim.

For the second claim, observe that canon depends only on the relative ordering of bias values within each layer and applies the induced permutations consistently to the weights. Since applying any group action $\rho(g)$ merely reorders neurons within layers, it does not affect the outcome of the sorting operation. Hence,

$$\text{canon}(\boldsymbol{v}) = \text{canon}(\rho(g)\boldsymbol{v}) \quad \forall g \in G. \tag{181}$$

This completes the proof. $\qquad\square$

**Lemma F.7** (Approximation of the canonization map by DWS networks). *Let $\mathcal{V}_A$ be a weight-space with corresponding exclusion set $\mathcal{E}_A$, and let $K \subseteq \mathcal{V}_A \setminus \mathcal{E}_A$ be compact. Define a map $\Psi : \mathcal{V}_A \longrightarrow \mathcal{V}_A^{n+5}$ where $n = \sum_\ell 5d_\ell(1 + d_{\ell-1})$ as follows: for $\boldsymbol{v} = (\boldsymbol{W}_1, \boldsymbol{b}_1, \ldots, \boldsymbol{W}_L, \boldsymbol{b}_L)$, $\text{NI}(\boldsymbol{v}) = (\boldsymbol{W}_1^*, \boldsymbol{b}_1^*, \ldots, \boldsymbol{W}_L^*, \boldsymbol{b}_L^*)$ let $\Psi(\boldsymbol{v}) = (\boldsymbol{W}_1', \boldsymbol{b}_1', \ldots, \boldsymbol{W}_L', \boldsymbol{b}_L')$*

*where for each layer $\ell$ and all valid indices $i, j$,*

$$(\boldsymbol{W}_\ell')_{i,j,:} := \big((\boldsymbol{W}_\ell^*)_{i,j}, \; \text{Flat}(\text{canon}(\boldsymbol{v}))\big), \tag{182}$$

*and*

$$(\boldsymbol{b}_\ell')_{i,:} := \big((\boldsymbol{b}_\ell^*)_i, \; \text{Flat}(\text{canon}(\boldsymbol{v}))\big), \tag{183}$$

*and* Flat *is the standard flattening operator $\mathcal{V}^5 \to \mathbb{R}^n$. That is, $\Psi$ concatenates the vectorized canonized representation* $\text{Flat}(\text{canon}(\boldsymbol{v}))$ *to every weight and bias entry. Then, for every $\epsilon > 0$, there exists a DWS network*

$$\Phi : \mathcal{V}_A \longrightarrow \mathcal{V}_A^{n+5} \tag{184}$$

*such that*

$$\sup_{\boldsymbol{v} \in K} \big\|\Phi(\boldsymbol{v}) - \Psi(\boldsymbol{v})\big\| < \epsilon. \tag{185}$$

*Proof.* We begin by observing that for any $\boldsymbol{v} \in \mathcal{V}_A$, the map

$$\text{NI}(\boldsymbol{v}) = (\boldsymbol{W}_1^*, \boldsymbol{b}_1^*, \ldots, \boldsymbol{W}_L^*, \boldsymbol{b}_L^*) \tag{186}$$

assigns to each weight and bias entry a unique auxiliary feature vector. Moreover, these auxiliary vectors take values in a finite, discrete set. Specifically,

$$(\boldsymbol{W}_{\ell_1}^*)_{i_1,j_1,1:} \neq (\boldsymbol{W}_{\ell_2}^*)_{i_2,j_2,1:} \neq (\boldsymbol{b}_{\ell_3}^*)_{i_3,1:} \tag{187}$$

for any distinct combinations of indices. All vectors $(\boldsymbol{W}_\ell^*)_{i,j,1:}$ and $(\boldsymbol{b}_\ell^*)_{i,1:}$ take values in the finite index set

$$I = \Big\{ (n_1, n_2, n_3, n_4) \,\big|\, n_1 \in [L], \; n_2 \in \{0, 1\}, \; n_3, n_4 \in \big[ \max_{\ell \in \{1,\ldots,L\}} d_\ell \big] \Big\}. \tag{188}$$

For each $\boldsymbol{n} \in I$, define a continuous function $f_{\boldsymbol{n}} : \mathbb{R}^5 \to \mathbb{R}^5$ by

$$f_{\boldsymbol{n}}(\boldsymbol{x}) = \begin{cases} \boldsymbol{x}, & \|\boldsymbol{x}_{1:} - \boldsymbol{n}\| < 1/4, \\ 0, & \|\boldsymbol{x}_{1:} - \boldsymbol{n}\| > 1/2. \end{cases} \tag{189}$$

By construction, for any term $\boldsymbol{x} = (\boldsymbol{W}_\ell^*)_{i,j,:}$ or $\boldsymbol{x} = (\boldsymbol{b}_\ell^*)_{i,:}$, there exists a unique $\boldsymbol{n} \in I$ such that $f_{\boldsymbol{n}}(\boldsymbol{x}) = \boldsymbol{x}$, while for all other terms $\boldsymbol{x}' \neq \boldsymbol{x}$ we have $f_{\boldsymbol{n}}(\boldsymbol{x}') = 0$.

Since each $f_{\boldsymbol{n}}$ is continuous and all inputs lie in a compact subset of $\mathbb{R}^5$, there exists, for any $\epsilon > 0$, an MLP $M_{\boldsymbol{n}}$ such that

$$\|M_{\boldsymbol{n}}(\boldsymbol{x}) - f_{\boldsymbol{n}}(\boldsymbol{x})\| < \epsilon \tag{190}$$

for all admissible $\boldsymbol{x}$. By Corollary D.15, DWS networks can realize the pointwise application of any MLP to feature channel vectors. In particular, for each $\boldsymbol{n} \in I$, we may realize

$$(\boldsymbol{W}_\ell')_{i,j,:} = M_{\boldsymbol{n}}\big((\boldsymbol{W}_\ell^*)_{i,j,:}\big), \tag{191}$$

$$(\boldsymbol{b}_\ell')_{i,:} = M_{\boldsymbol{n}}\big((\boldsymbol{b}_\ell^*)_{i,:}\big). \tag{192}$$

Additionally, define the equivariant sum-broadcast operator $SB(\boldsymbol{W}_1, \boldsymbol{b}_1, \ldots, \boldsymbol{W}_L, \boldsymbol{b}_L) = (\boldsymbol{W}_1', \boldsymbol{b}_1', \ldots, \boldsymbol{W}_L', \boldsymbol{b}_L')$ by

$$(\boldsymbol{W}_\ell')_{i,j,:} = \Big[ (\boldsymbol{W}_\ell)_{i,j,:}, \sum_{i,j,\ell} (\boldsymbol{W}_\ell)_{i,j,:} + \sum_{i,\ell} (\boldsymbol{b}_\ell)_{i,:} \Big], \tag{193}$$

$$(\boldsymbol{b}_\ell')_{i,:} = \Big[ (\boldsymbol{b}_\ell)_{i,:}, \sum_{i,j,\ell} (\boldsymbol{W}_\ell)_{i,j,:} + \sum_{i,\ell} (\boldsymbol{b}_\ell)_{i,:} \Big], \tag{194}$$

where $[\cdot, \cdot]$ denotes concatenation. This operator is affine and equivariant, and therefore realizable by DWS layers.

Combining $SB$ with the termwise application of each $M_{\boldsymbol{n}}$, we obtain, for every $\boldsymbol{n} \in I$, a DWS-realizable map satisfying

$$(\boldsymbol{W}_\ell')_{i,j,:} = \big[ (\boldsymbol{W}_\ell^*)_{i,j,:}, \boldsymbol{x}_{\boldsymbol{n}} \big], \tag{195}$$

$$(\boldsymbol{b}_\ell')_{i,:} = \big[ (\boldsymbol{b}_\ell^*)_{i,:}, \boldsymbol{x}_{\boldsymbol{n}} \big], \tag{196}$$

where $\boldsymbol{x}_{\boldsymbol{n}}$ is the unique term (either $\boldsymbol{x}_{\boldsymbol{n}} = (\boldsymbol{W}_\ell)_{i,j,:}$ or $\boldsymbol{x}_{\boldsymbol{n}} = (b\ell)_{i,:}$) whose auxiliary feature satisfies $\boldsymbol{x}_{1:} = \boldsymbol{n}$.

Applying this construction for all $\boldsymbol{n} \in I$, concatenating the results in lexicographic order, and choosing $\epsilon$ sufficiently small so that each $M_{\boldsymbol{n}}$ accurately approximates $f_{\boldsymbol{n}}$, we obtain

$$(\boldsymbol{W}_\ell')_{i,j,:} \approx \big( (\boldsymbol{W}_\ell^*)_{i,j}, \; \mathrm{Flat}(\mathrm{canon}(\boldsymbol{v})) \big), \tag{197}$$

and

$$(\boldsymbol{b}_\ell')_{i,:} \approx \big( (\boldsymbol{b}_\ell^*)_i, \; \mathrm{Flat}(\mathrm{canon}(\boldsymbol{v})) \big). \tag{198}$$

This completes the proof. $\qquad\square$

**Lemma F.8** (Reduction of equivariant maps to a canonical form)**.** *Let $\mathcal{V}$ be a weight-space with corresponding exclusion set $\mathcal{E}$, and let $K \subseteq \mathcal{V} \setminus \mathcal{E}$ be compact. Let $\Psi : \mathcal{V} \longrightarrow \mathcal{V}$ be a continuous $G$-equivariant map. Then there exists a continuous function $f : \mathbb{R}^{n+5} \longrightarrow \mathbb{R}$ with $n := \sum_{\ell=1}^L 5d_\ell(1 + d_{\ell-1})$ such that the following holds: for every $\boldsymbol{v} = (\boldsymbol{W}_1, \boldsymbol{b}_1, \ldots, \boldsymbol{W}_L, \boldsymbol{b}_L) \in \mathcal{V}_A \setminus \mathcal{E}_A$, let*

$$\mathrm{NI}(\boldsymbol{v}) = (\boldsymbol{W}_1^*, \boldsymbol{b}_1^*, \ldots, \boldsymbol{W}_L^*, \boldsymbol{b}_L^*), \qquad \Psi(\boldsymbol{v}) = (\boldsymbol{W}_1', \boldsymbol{b}_1', \ldots, \boldsymbol{W}_L', \boldsymbol{b}_L'). \tag{199}$$

*Then, for all layers $\ell$ and all valid indices $i, j$, we have*

$$(\boldsymbol{W}_\ell')_{i,j} = f\big((\boldsymbol{W}_\ell^*)_{i,j,:}, \; \mathrm{Flat}(\mathrm{canon}(\boldsymbol{v}))\big), \tag{200}$$

*and*

$$(\boldsymbol{b}_\ell')_i = f\big((\boldsymbol{b}_\ell^*)_{i,:}, \; \mathrm{Flat}(\mathrm{canon}(\boldsymbol{v}))\big). \tag{201}$$

*Proof.* Fix a term $\boldsymbol{x} = (\boldsymbol{W}_\ell^*)_{i,j,:}$ or $\boldsymbol{x} = (\boldsymbol{b}_\ell^*)_{i,:}$. By construction of NI, the last four coordinates $\boldsymbol{n} := \boldsymbol{x}_{1:}$ encode the exact position of $\boldsymbol{x}$ under the lexicographic sorting used in the definition of $\mathrm{canon}(\boldsymbol{v})$. In particular, knowing $\boldsymbol{n}$ uniquely determines the index set

$$I_{\boldsymbol{n}} = \{i_1, \ldots, i_k\} \tag{202}$$

such that the entries of $\mathrm{Flat}(\mathrm{canon}(\boldsymbol{v}))$ at these indices correspond exactly to $\boldsymbol{x}$. More formally,

$$[\mathrm{Flat}(\mathrm{canon}(\boldsymbol{v}))_{i_1}, \ldots, \mathrm{Flat}(\mathrm{canon}(\boldsymbol{v}))_{i_k}] = \mathrm{Flat}(\boldsymbol{x}). \tag{203}$$

Since $\boldsymbol{n}$ takes values in a finite set, we may define a continuous function $f_1 : \mathbb{R}^5 \longrightarrow \mathbb{R}^n$ such that

$$f_1(\boldsymbol{x}) = e_{\boldsymbol{n}}, \tag{204}$$

where $e_{\boldsymbol{n}} \in \mathbb{R}^n$ is the indicator vector defined by

$$(e_{\boldsymbol{n}})_i = \begin{cases} 1, & i \in I_{\boldsymbol{n}}, \\ 0, & \text{otherwise.} \end{cases} \tag{205}$$

Let $K^* := \mathrm{Flat}(\mathrm{canon}(K)) \subseteq \mathbb{R}^n$. Since $\mathrm{canon}$ and $\mathrm{Flat}$ are continuous and $K$ is compact, $K^*$ is compact as well. Define the function $f_2 : K^* \longrightarrow \mathbb{R}^n$ by

$$f_2(\mathrm{Flat}(\mathrm{canon}(\boldsymbol{v}))) = \mathrm{Flat}(\Psi(\mathrm{canon}(\boldsymbol{v}))). \tag{206}$$

This function is well defined and continuous on $K^*$, and therefore admits a continuous extension to all of $\mathbb{R}^n$, which we again denote by $f_2$. We now define

$$f(\boldsymbol{x}, \mathrm{canon}(\boldsymbol{v})) := \mathrm{Sum}\big(\mathrm{Flat}^{-1}\big(f_1(\boldsymbol{x}) \odot f_2(\mathrm{canon}(\boldsymbol{v}))\big)\big), \tag{207}$$

where $\odot$ denotes elementwise multiplication, and $\mathrm{Sum}$ is the affine equivariant map that sums all terms:

$$\mathrm{Sum}(\boldsymbol{v}) = \sum_{i,j,\ell} (\boldsymbol{W}_\ell)_{i,j,:} + \sum_{i,\ell} (\boldsymbol{b}_\ell)_{i,:}. \tag{208}$$

The function $f$ is continuous as a composition of continuous operations. By construction, for every $\boldsymbol{v} = (\boldsymbol{W}_1, \boldsymbol{b}_1, \ldots, \boldsymbol{W}_L, \boldsymbol{b}_L)$ with $\mathrm{canon}(\boldsymbol{v}) = (\boldsymbol{W}_1^*, \boldsymbol{b}_1^*, \ldots, \boldsymbol{W}_L^*, \boldsymbol{b}_L^*)$ and $\Psi(\mathrm{canon}(\boldsymbol{v})) = (\boldsymbol{W}_1', \boldsymbol{b}_1', \ldots, \boldsymbol{W}_L', \boldsymbol{b}_L')$, we have

$$f((\boldsymbol{W}_\ell^*)_{i,j,:}, \mathrm{Flat}(\mathrm{canon}(\boldsymbol{v}))) = (\boldsymbol{W}_\ell')_{i,j,:}, \tag{209}$$

and

$$f((\boldsymbol{b}_\ell^*)_{i,:}, \mathrm{Flat}(\mathrm{canon}(\boldsymbol{v}))) = (\boldsymbol{b}_\ell')_{i,:}. \tag{210}$$

Define $\Psi^* : \mathcal{V}_A \to \mathcal{V}_A$ by setting, for $\boldsymbol{v} = (\boldsymbol{W}_1, \boldsymbol{b}_1, \ldots, \boldsymbol{W}_L, \boldsymbol{b}_L)$,

$$(\boldsymbol{W}_\ell'')_{i,j} = f\big((\boldsymbol{W}_\ell^*)_{i,j,:}, \mathrm{Flat}(\mathrm{canon}(\boldsymbol{v}))\big), \tag{211}$$

$$(\boldsymbol{b}_\ell'')_i = f\big((\boldsymbol{b}_\ell^*)_{i,:}, \mathrm{Flat}(\mathrm{canon}(\boldsymbol{v}))\big). \tag{212}$$

Since $\mathrm{Flat}(\mathrm{canon}(\boldsymbol{v}))$ is continuous on $K$ and invariant under the action of $G$ (Lemma F.6), the map

$$(\boldsymbol{W}_\ell)_{i,j} \to [(\boldsymbol{W}_\ell^*)_{i,j,:}, \mathrm{Flat}(\mathrm{canon}(\boldsymbol{v}))], \qquad (\boldsymbol{b}_\ell)_i \to [(\boldsymbol{b}_\ell^*)_{i,:}, \mathrm{Flat}(\mathrm{canon}(\boldsymbol{v}))]. \tag{213}$$

is continuous and equivariant. Hence, $\Psi^*$ is continuous and $G$-equivariant as a composition of equivariant maps with a shared continuous function acting independently along the feature dimension. By equations 209, 210, for every $\boldsymbol{v} \in \mathcal{V}_A$,

$$\Psi^*(\mathrm{canon}(\boldsymbol{v})) = \Psi(\mathrm{canon}(\boldsymbol{v})). \tag{214}$$

Since $\Psi^*$ is $G$-equivariant, for any $v \in \mathcal{V}_A$ let $g \in G$ be such that $v = \rho(g)\text{canon}(v)$ (such a $g$ exists by Lemma F.6). Thus

$$
\begin{aligned}
\Psi^*(v) &= \Psi^*(\rho(g)\text{canon}(v)) \\
&= \rho(g)\Psi^*(\text{canon}(v)) \\
&= \rho(g)\Psi(\text{canon}(v)) \\
&= \Psi(\rho(g)\text{canon}(v)) \\
&= \Psi(v).
\end{aligned}
$$

This completes the proof. $\square$

**Proof of Theorem 7.4.**   Using the results developed above, we now prove Theorem 7.4.

*Proof.* Fix a weight-space $\mathcal{V}$, with corresponding set $\mathcal{E}$, a compact set $K \subseteq \mathcal{V}_A \setminus \mathcal{E}_A$, and $\epsilon > 0$. We show that there exists a DWS network $\Phi$ such that

$$
\sup_{v \in K} \|\Phi(v) - \Psi(v)\|_2 < \epsilon. \tag{215}
$$

By Lemma F.8, there exists a continuous function $f : \mathbb{R}^{n+5} \longrightarrow \mathbb{R}$, such that the equivariant map $\Psi$ admits the following pointwise representation. Given $v = (W_1, b_1, \ldots, W_L, b_L)$, and $\text{NI}(v) = (W_1^*, b_1^*, \ldots, W_L^*, b_L^*)$, the map $\Psi' : \mathcal{V} \to \mathcal{V}$, $\Psi'(v) = (W_1', b_1', \ldots, W_L', b_L')$ defined by

$$
(W_\ell')_{i,j} = f\big((W_\ell^*)_{i,j,:}, \ \text{Flat}(\text{canon}(v))\big), \tag{216}
$$

and

$$
(b_\ell')_i = f\big((b_\ell^*)_{i,:}, \ \text{Flat}(\text{canon}(v))\big), \tag{217}
$$

satisfies $\Psi = \Psi'$ on $K$. Equivalently, $\Psi$ can be written as the composition of the map

$$
(W_\ell)_{i,j} \longmapsto [(W_\ell^*)_{i,j,:}, \text{Flat}(\text{canon}(v))], \qquad (b_\ell)_i \longmapsto [(b_\ell^*)_{i,:}, \text{Flat}(\text{canon}(v))], \tag{218}
$$

followed by the application of the shared continuous function $f$ acting independently along the feature dimension. Since $f$ is continuous, there exists an MLP $M$ that approximates $f$ uniformly up to arbitrary precision. By Lemma F.7, the first map above can be approximated uniformly on $K$ by a DWS network. Moreover, by Corollary D.15, DWS architectures can simulate pointwise application of the MLP $M$ by adding a finite number of equivariant layers.

By Lemma D.13, composing these constructions yields a DWS network $\Phi$ such that

$$
\sup_{v \in K} \|\Phi(v) - \Psi(v)\|_2 < \epsilon, \tag{219}
$$

which completes the proof. $\square$

**DWS layers used for maximal expressivity.**   Throughout the proof of DWS universality under the general position assumption, we in fact only used a small subset of the full collection of available DWS updates. In particular, restricting a DWS model to *only* these update primitives still suffices to obtain universality under general position. Concretely, for $v \in \mathcal{V}^c$ written as $v = (W_1, b_1, \ldots, W_L, b_L)$, the update primitives used are:

- Pointwise affine update.

- Global summation operator.

- Bias summation operator.

- Lower weight-to-bias operator.

- Upper weight-to-bias operator.

- First layer per-neuron operator.

- Last layer per-neuron operator.

We define each primitive below. A DWS model obtained by composing these updates, interleaving them with nonlinearities, and allowing concatenation of multiple such operations within each layer already attains maximal expressivity. **Pointwise affine update.** This operator applies the same affine map to each weight and bias feature vector, using a matrix $\boldsymbol{A} \in \mathbb{R}^{c \times c'}$ and a vector $\boldsymbol{u} \in \mathbb{R}^{c'}$. Concretely, for every layer $\ell$ and all valid indices $i, j$, we update

$$(\boldsymbol{W}_\ell)_{i,j,:} \longmapsto (\boldsymbol{W}_\ell)_{i,j,:}\boldsymbol{A} + \boldsymbol{u}, \qquad (\boldsymbol{b}_\ell)_{i,:} \longmapsto (\boldsymbol{b}_\ell)_{i,:}\boldsymbol{A} + \boldsymbol{u}. \tag{220}$$

**Global summation operator.** This operator aggregates all weight and bias feature vectors in $\boldsymbol{v}$ via a global sum, and then broadcasts the resulting vector back to every entry. Namely, define the global summary

$$\boldsymbol{s}(\boldsymbol{v}) \coloneqq \sum_\ell \sum_{i,j} (\boldsymbol{W}_\ell)_{i,j,:} + \sum_\ell \sum_i (\boldsymbol{b}_\ell)_{i,:} \in \mathbb{R}^c. \tag{221}$$

Then, for every layer $\ell$ and all valid indices $i, j$, we update

$$(\boldsymbol{W}_\ell)_{i,j,:} \longmapsto \boldsymbol{s}(\boldsymbol{v}), \qquad (\boldsymbol{b}_\ell)_{i,:} \longmapsto \boldsymbol{s}(\boldsymbol{v}). \tag{222}$$

**Bias summation operator.** Fix a layer index $\ell$. The $\ell$-th bias summation operator aggregates the bias vectors of layer $\ell$ by summation, broadcasts the result across the bias positions of that layer, and sets all other weight and bias entries to zero. Concretely, for every layer $\ell' \neq \ell$ and all valid indices $i, j$, we set

$$(\boldsymbol{W}_{\ell'})_{i,j,:} \longmapsto 0, \qquad (\boldsymbol{b}_{\ell'})_{i,:} \longmapsto 0. \tag{223}$$

For the selected layer $\ell$, we set all weight entries to zero and replace each bias entry by the sum of all biases in that layer:

$$(\boldsymbol{W}_\ell)_{i,j,:} \longmapsto 0, \qquad (\boldsymbol{b}_\ell)_{i,:} \longmapsto \sum_{i'} (\boldsymbol{b}_\ell)_{i',:}. \tag{224}$$

**Lower weight-to-bias operator.** This operator replaces each weight feature vector by the bias feature vector of the *lower* neuron connected to that weight, while leaving all bias terms unchanged. Concretely, for every layer $\ell$ and all valid indices $i, j$, we update

$$(\boldsymbol{W}_\ell)_{i,j,:} \longmapsto (\boldsymbol{b}_{\ell-1})_{i,:}, \qquad (\boldsymbol{b}_\ell)_{i,:} \longmapsto (\boldsymbol{b}_\ell)_{i,:}. \tag{225}$$

**Upper weight-to-bias operator.** This operator is analogous to the lower weight-to-bias operator, but replaces each weight feature vector by the bias feature vector of the *upper* neuron incident to that weight, while leaving all bias terms unchanged. Concretely, for every layer $\ell$ and all valid indices $i, j$, we update

$$(\boldsymbol{W}_\ell)_{i,j,:} \longmapsto (\boldsymbol{b}_\ell)_{j,:}, \qquad (\boldsymbol{b}_\ell)_{i,:} \longmapsto (\boldsymbol{b}_\ell)_{i,:}. \tag{226}$$

**First-layer per-neuron operator.** This operator preserves only the weights associated with a specified input neuron in the first layer, and sets all other weights and bias terms to zero. Concretely, for fixed $i' \in [d_0]$ and all $i \in [d_0], j \in [d_1]$, we apply

$$(\boldsymbol{W}_1)_{i,j,:} \longmapsto \begin{cases} (\boldsymbol{W}_1)_{i,j,:}, & \text{if } i = i', \\ 0, & \text{otherwise.} \end{cases} \tag{227}$$

All remaining weight and bias terms are mapped to zero. **Last-layer per-neuron update.** This operator preserves only the bias terms associated with a specified output neuron neuron in the last layer, and sets all other weights and bias terms to zero. Concretely, for fixed $j' \in [d_L]$ and all $j \in [d_L]$, we apply

$$(\boldsymbol{b}_L)_{j,:} \longmapsto \begin{cases} (\boldsymbol{b}_L)_{j,:}, & \text{if } j = j', \\ 0, & \text{otherwise.} \end{cases} \tag{228}$$

All remaining weight and bias terms are mapped to zero.

## F.2. Universal approximation of function-space operators

In this section, we study the expressive power of weight-space models in approximating continuous function-space operators

$$\Psi : \mathcal{C}(X, Y) \longrightarrow \mathcal{C}(X, Y), \tag{229}$$

where, throughout this section, $X \subseteq \mathbb{R}^n$ is compact and $Y \subseteq \mathbb{R}^m$.

Our first result formalizes Proposition 7.3, showing that when restricted to a fixed input architecture, DWS networks fail to approximate even natural function-space operators.

**Theorem F.9** (Limitations of fixed architecture). *Let $A = (\boldsymbol{d}, \sigma)$ be a fixed MLP architecture with activation $\sigma = \mathrm{ReLU}$, and let $\mathcal{V}_A$ denote its parameter space. Let $X = [0, 1]^n$, and let $\Psi : \mathcal{C}(X, Y) \to \mathcal{C}(X, Y)$ be a continuous function-space operator. Suppose there exist $a < b$ in $[0, 1]$ such that for every $f \in \mathcal{C}(X, Y)$ and every $\boldsymbol{x} \in [a, b]^n$,*

$$\Psi(f)(\boldsymbol{x}) = f\left(\frac{\boldsymbol{x} - a}{b - a}\right), \tag{230}$$

*and moreover for every $f$ there exists $\boldsymbol{x}^* \in X \setminus [a, b]^n$ such that*

$$\Psi(f)(\boldsymbol{x}^*) \neq f\left(\frac{\boldsymbol{x}^* - a}{b - a}\right) \tag{231}$$

*Then there exists a constant $C > 0$ and $\boldsymbol{v} \in \mathcal{V}_A$ such that, for every DWS model $\Phi : \mathcal{V}_A \to \mathcal{V}_A$*

$$\|\Psi(f_{\boldsymbol{v}}) - f_{\Phi(\boldsymbol{v})}\|_\infty \geq C. \tag{232}$$

Note that Theorem F.9 implies that, when operating on weights of a *fixed and small* base architecture, DWS models are inherently unable to approximate several natural function-space operators. Intuitively, the theorem captures the following obstruction: the operator $\Psi$ enforces a prescribed *rescaling behavior* on a subdomain $[a, b]^n$, while simultaneously requiring the output to exhibit *new non-trivial behavior* outside this subdomain. This combination cannot, in general, be realized by applying a continuous map directly to the weights of a fixed architecture.

A concrete example arises when a function $f$ encodes a natural image or a 3D scene using a neural implicit representation (e.g., INRs or NeRFs). In this setting, a natural transformation is a *zoom-out* operator: the original scene is preserved at a smaller spatial scale, while new regions of the scene, absent from the input, are introduced. In the notation of the theorem, the "preserved" part corresponds to the box $[a, b]^n$, on which the output must agree with a rescaled version of the input. At the same time, introducing new regions requires the output to behave *non-trivially* outside of $[a, b]^n$, as captured by equation 231. Theorem F.9 therefore shows that such zoom-out operators cannot be approximated by DWS models acting on the weights of a fixed small architecture.

Another implication concerns function-level domain adaptation. Suppose that a function $f$ corresponds to a global minimizer of a loss defined over some training set, and we wish to map it to a function $\Psi(f)$ that is a global minimizer of a *different* loss which incorporates additional data points not seen during training. This necessarily requires extending the behavior of $f$ in regions of the domain that were previously unconstrained. As an immediate corollary of Theorem F.9, operators $\Psi : \mathcal{C}([0, 1]^n) \to \mathcal{C}([0, 2]^n)$ that preserve the input function on the unit box but extend it in a non-trivial way to the larger domain cannot be approximated by DWS models operating on fixed architectures. Indeed, such extensions can be reframed as zoom-out transformations, and hence fall under the same obstruction. In particular, this shows that function-level domain adaptation of this form cannot, in general, be learned by DWS models acting on small fixed architectures.

While the previous theorem shows that in general, DWS networks are not universal approximators of the class of continuous function-space operators, our second result shows that this limitation disappears once we allow DWS to operate on large enough architectures. In particular, by choosing a large enough weight-space architecture, DWS models can approximate any continuous function-space operator up to arbitrary precision. The next theorem formalizes this and provides a precise analogue of Theorem 7.2.

**Theorem F.10** (Universal approximation with growing capacity). *Let $\Psi : \mathcal{C}(X, Y) \to \mathcal{C}(X, Y)$ be a continuous function-to-function map and let $K \subseteq \mathcal{C}(X, Y)$ be a compact set of functions. For every $\epsilon > 0$ there exists $\delta > 0$ and an architecture $A$ such that:*

*1. There exists a compact set $K' \subseteq \mathcal{V}$ which is a $\delta$-approximator of $K$ (see Definition B.5)*

*2. For any such set $K'$ there exists a DWS network $\Phi : \mathcal{V}_A \to \mathcal{V}_A$ such that, for every $\boldsymbol{v} \in K'$,*

$$\| f_{\Phi(\boldsymbol{v})} - \Psi(f_{\boldsymbol{v}}) \|_\infty < \epsilon. \tag{233}$$

### F.2.1. PROOF OF THEOREM F.9

For the remainder of this subsection, we fix $X = [0, 1]^n$. Before proving Theorem F.9, we introduce a convenient class of functions that captures the geometric complexity of ReLU networks with a fixed architecture.

**Definition F.11** (Piecewise-affine functions with bounded complexity). For integers $M, R \in \mathbb{N}$, we denote by $\mathrm{PL}_{M,R}$ the set of all functions $f \in \mathcal{C}(X, Y)$ for which there exist convex polytopes $P_1, \ldots, P_m \subseteq X$ and affine functions $A_1, \ldots, A_m : X \to Y$ such that:

1. $1 \le m \le M$;

2. for each $j$, the polytope $P_j$ can be written as

$$P_j = \mathrm{conv}\{\boldsymbol{v}_{j,1}, \ldots, \boldsymbol{v}_{j,r_j}\} \tag{234}$$

   for some $\boldsymbol{v}_{j,\ell} \in X$ with $1 \le r_j \le R$;

3. the interiors are pairwise disjoint and cover $X$,

$$X = \bigcup_{j=1}^m P_j, \qquad \mathrm{int}(P_i) \cap \mathrm{int}(P_j) = \emptyset \text{ for } i \ne j; \tag{235}$$

4. on each polytope $P_j$ the function $f$ is affine, i.e.

$$f(x) = A_j(x) \qquad \forall x \in P_j. \tag{236}$$

We refer to $P_1, \ldots, P_m$ as the *linearity regions* of $f$.

The next lemma shows that, for a fixed ReLU architecture, there are uniform bounds on both the number of linearity regions induced by each network and the number of extreme points of each such region. The arguments used in this lemma are standard (see e.g. (Montúfar et al., 2014; Raghu et al., 2017; Serra et al., 2018)), but we include them for completeness.

**Lemma F.12** (Geometric complexity of fixed ReLU architectures). *Let $A = (\boldsymbol{d}, \sigma)$ be an architecture with $\sigma = \mathrm{ReLU}$, and let $\mathcal{V}_A$ denote its parameter space. Then there exist integers $M_A, R_A \in \mathbb{N}$ such that, for every $\boldsymbol{v} \in \mathcal{V}_A$, the realized function $f_{\boldsymbol{v}}$ belongs to $\mathrm{PL}_{M_A, R_A}$.*

*Proof.* Before applying any layer, the domain $X = [0, 1]^n$ is a single convex polytope on which the identity map $h^{(0)}(\boldsymbol{x}) = \boldsymbol{x}$ is affine. Hence at depth $0$ we have a single region with $2^n$ vertices.

Consider now the effect of applying one ReLU neuron to a single convex polytope $P \subseteq X$. The neuron computes

$$\boldsymbol{x} \mapsto \sigma(\boldsymbol{W}^\top \boldsymbol{x} + \boldsymbol{b}). \tag{237}$$

The hyperplane

$$\mathcal{H} = \{\boldsymbol{x} : \boldsymbol{W}^\top \boldsymbol{x} + \boldsymbol{b} = 0\} \tag{238}$$

can intersect $P$ and partition it into at most two convex polytopes, corresponding to the regions where the neuron is active and inactive, respectively. Moreover, every vertex of these two polytopes is either

1. an original vertex of $P$, or

2. an intersection of $\mathcal{H}$ with an edge of $P$.

If $P$ has $R$ vertices, it has at most $R(R-1)/2$ edges. Therefore, after the split, each of the resulting polytopes has at most

$$R + \frac{R(R-1)}{2} = \frac{R(R+1)}{2} \tag{239}$$

vertices. Thus, a single ReLU neuron applied to a polytope with $R$ vertices produces at most two polytopes, each with at most $\frac{R(R+1)}{2}$ vertices.

We now propagate these bounds through the network. Consider a fixed layer $\ell$. Each pre-activation in this layer is a linear combination of the $d_{\ell-1}$ outputs of the previous layer. By induction, we may assume that every such output is affine on at most $M_{\ell-1}$ regions, and every region has at most $R_{\ell-1}$ vertices.

In order to describe the linearity regions of a pre-activation in layer $\ell$, observe that a point $x$ belongs to a region of affinity when all summands appearing in the linear combination are affine at $x$. Therefore each linearity region of the pre-activation is obtained as an intersection of $d_{\ell-1}$ regions, one from each of the pieces of the previous layer.

Hence the total number of such intersections is bounded by

$$M_{\ell-1}^{d_{\ell-1}}, \tag{240}$$

and each intersection can be written as the intersection of at most $d_{\ell-1}$ polytopes, each with at most $R_{\ell-1}$ vertices. This yields a uniform bound of

$$R_{\ell-1}^{d_{\ell-1}} \tag{241}$$

vertices for every resulting region.

Consequently, every pre-activation in layer $\ell$ is affine on at most $M_{\ell-1}^{d_{\ell-1}}$ regions, each of which is a convex polytope with at most $R_{\ell-1}^{d_{\ell-1}}$ vertices.

Applying this reasoning layer by layer shows, by induction over the depth of the network, that there exist integers $M_A, R_A$ depending only on the architecture $A$ and the input dimension $n$ such that, for every choice of weights $v$, the function $f_v$ is affine on each of at most $M_A$ convex polytopes, each of which can be written as the convex hull of at most $R_A$ vertices. That is,

$$f_v \in \mathrm{PL}_{M_A, R_A}, \tag{242}$$

completing the proof.

$\square$

The previous lemma shows that, for a fixed architecture, all realizable functions lie in some class $\mathrm{PL}_{M,R}$ of uniformly bounded geometric complexity. Our next lemma shows that $\mathrm{PL}_{M,R}$ is a closed set in $\mathcal{C}(X, Y)$ with respect to the supremum norm.

**Lemma F.13** (Piecewise-affine classes are closed sets). *For integers $M, R \in \mathbb{N}$, the set $\mathrm{PL}_{M,R} \subseteq \mathcal{C}(X, Y)$ is closed with respect to the supremum norm.*

*Proof.* Let $(f_k)$ be a sequence in $\mathrm{PL}_{M,R}$ with $f_k \to f$ uniformly on $X$. For each $k$, there exist:

- an integer $1 \leq m_k \leq M$,

- convex polytopes $P_1^{(k)}, \ldots, P_{m_k}^{(k)} \subseteq X$,

- affine functions $A_1^{(k)}, \ldots, A_{m_k}^{(k)}$,

such that:

1. $X = \bigcup_{j=1}^{m_k} P_j^{(k)}$ and $\mathrm{int}(P_i^{(k)}) \cap \mathrm{int}(P_j^{(k)}) = \emptyset$ for $i \neq j$;

2. each $P_j^{(k)}$ can be written as

$$P_j^{(k)} = \text{conv}\{\boldsymbol{v}_{j,1}^{(k)}, \ldots, \boldsymbol{v}_{j,r_j^k}^{(k)}\}, \tag{243}$$

with $1 \le r_j^k \le R$ and $\boldsymbol{v}_{j,\ell}^{(k)} \in X$;

3. $f_k(\boldsymbol{x}) = A_j^{(k)}(\boldsymbol{x})$ for all $\boldsymbol{x} \in P_j^{(k)}$.

We first standardize the description so that each $f_k$ uses exactly $M$ regions and exactly $R$ vertices per region. If $m_k < M$, we add polytopes $P_j^{(k)}$ with empty interiors that sit on the boundary of non-degenerate polytopes, and choose the corresponding affine maps $A_j^{(k)}$ to be the same as for the non-degenerate polytope selected. If a polytope $P_j^{(k)}$ has fewer than $R$ vertices, i.e., $P_j^{(k)} = \text{conv}\{\boldsymbol{v}_{j,1}^{(k)}, \ldots, \boldsymbol{v}_{j,r_j^k}^{(k)}\}$ with $r_j^k < R$, we repeat vertices so that we obtain a representation

$$P_j^{(k)} = \text{conv}\{\boldsymbol{v}_{j,1}^{(k)}, \ldots, \boldsymbol{v}_{j,R}^{(k)}\}. \tag{244}$$

Thus, without loss of generality, we may assume that for each $k$:

- we have polytopes $P_1^{(k)}, \ldots, P_M^{(k)}$ that cover $X$, and whose interiors are pairwise disjoint;

- each $P_j^{(k)}$ is given by

$$P_j^{(k)} = \text{conv}\{\boldsymbol{v}_{j,1}^{(k)}, \ldots, \boldsymbol{v}_{j,R}^{(k)}\}, \tag{245}$$

for some $\boldsymbol{v}_{j,\ell}^{(k)} \in X$;

- $f_k(\boldsymbol{x}) = A_j^{(k)}(\boldsymbol{x})$ for all $\boldsymbol{x} \in P_j^{(k)}$.

Since $X$ is compact, so is the Cartesian product $X^{MR}$. Thus the finite collection of vertices $(\boldsymbol{v}_{j,\ell}^{(k)})_{1 \le j \le M, 1 \le \ell \le R}$ admits a convergent subsequence. Passing to this subsequence and renaming indices, we may assume that

$$\boldsymbol{v}_{j,\ell}^{(k)} \to \boldsymbol{v}_{j,\ell} \in X \quad \text{as } k \to \infty, \qquad \forall j, \ell. \tag{246}$$

For each $j$, define the limiting vertex set $V_j := \{\boldsymbol{v}_{j,1}, \ldots, \boldsymbol{v}_{j,R}\}$ and the limiting polytope

$$P_j := \text{conv}(V_j). \tag{247}$$

By continuity of the convex hull operator in the Hausdorff metric, $P_j^{(k)} \to P_j$ in Hausdorff distance as $k \to \infty$, for each $j$.

Fix an index $j$ such that $\text{int}(P_j) \ne \emptyset$. Let $\boldsymbol{x}_0 \in \text{int}(P_j)$. There exists $\rho > 0$ such that the ball $B(\boldsymbol{x}_0, \rho)$ is contained in $P_j$. Since $P_j^{(k)} \to P_j$ in Hausdorff distance, for all sufficiently large $k$ we have

$$B(\boldsymbol{x}_0, \rho/2) \subseteq P_j^{(k)}. \tag{248}$$

Thus, on $B(\boldsymbol{x}_0, \rho/2)$ we have $f_k = A_j^{(k)}$, and $f_k$ converges uniformly to $f$. Since a sequence of affine maps uniformly converges to $f$ on $B(\boldsymbol{x}_0, \rho/2)$, $f$ is itself affine on $B(\boldsymbol{x}_0, \rho/2)$. By connectedness and convexity of $\text{int}(P_j)$, the same affine representation extends to all of $\text{int}(P_j)$, and by continuity to $P_j$.

We now prove that $P_1, \ldots, P_M$ satisfy $X = \bigcup_{j=1}^M P_j$ and $\text{int}(P_i) \cap \text{int}(P_j) = \emptyset$ for $i \ne j$. First, since $P_1^{(k)}, \ldots, P_M^{(k)}$ always cover $X$, for each $\boldsymbol{x} \in X$ there exists an integer $i$ such that $\boldsymbol{x} \in P_i^{(k)}$ for infinitely many values of $k$. Since $P_i^{(k)}$ converges to $P_i$ in the Hausdorff metric, we get that $\boldsymbol{x} \in P_i$, thus $X = \bigcup_{j=1}^M P_j$. Second, assume for some $i, j$ we have $\text{int}(P_i) \cap \text{int}(P_j) \ne \emptyset$, then there exists some ball $B(\boldsymbol{x}, \rho) \subseteq \text{int}(P_i) \cap \text{int}(P_j)$. Since $P_j^{(k)} \to P_j$, $P_i^{(k)} \to P_i$ in Hausdorff distance, for all sufficiently large $k$ we have

$$B(\boldsymbol{x}_0, \rho/2) \subseteq P_j^{(k)} \cap P_i^{(k)}. \tag{249}$$

This contradicts the definition of $P_1^{(k)}, \ldots, P_m^{(k)}$ as a cover with disjoint interiors.

We have thus shown $f$ is affine on polytopes $P_1, \ldots P_M$ which all have at most $R$ vertices, cover $X$ and have disjoint interiors, completing the proof.

$\square$

We are now ready to prove Theorem F.9. The key idea is that, for a fixed architecture, the family of realizable functions has uniformly bounded geometric complexity (Lemma F.12), and this property is preserved under any DWS map as its output is weights of the same architecture. In contrast, the function-space operator $\Psi$ specified in Theorem F.9 produces functions whose behavior on $X$ cannot be matched arbitrarily well by such a family, as it requires adding more nonlinearity regions.

*Proof of Theorem F.9.* Fix an architecture $A$ with corresponding weight space $\mathcal{V}_A$. From Lemma F.12 we know there exist constants $M, R$ such that for every $\boldsymbol{v} \in \mathcal{V}_A$, $f_{\boldsymbol{v}} \in \mathrm{PL}_{M,R}$. Choose $M, R$ to be the minimal choices for such constants. Recall that the operator $\Psi$ in Theorem F.9 satisfies, by assumption, that there exist $a, b \in [0, 1]$ with $a < b$ such that for all $f$ and all $\boldsymbol{x} \in [a, b]^n$,

$$\Psi(f)(\boldsymbol{x}) = f\left(\frac{\boldsymbol{x} - a}{b - a}\right), \tag{250}$$

while for each $f \in \mathcal{C}(X, Y)$ there exists an $\boldsymbol{x}^* \in X \setminus [a, b]^n$ such that

$$\Psi(f)(\boldsymbol{x}^*) \neq f\left(\frac{\boldsymbol{x}^* - a}{b - a}\right). \tag{251}$$

Choose weights $\boldsymbol{v} \in \mathcal{V}_A$ such that $f_{\boldsymbol{v}}$ has exactly $M$ nonlinearity regions on $X$ denoted by $P_1, \ldots, P_M$. The function $\Psi(f_{\boldsymbol{v}})$ is an affine rescaling of $f_{\boldsymbol{v}}$ on the smaller cube $[a, b]^n$ and so it exactly $M$ nonlinearity regions there. Assume now that $\Psi(f_{\boldsymbol{v}}) \in \mathrm{PL}_{M,R}$; then $\Psi(f_{\boldsymbol{v}})$ must have exactly $M$ nonlinearity regions in the larger cube $X = [0, 1]^n$ each intersecting the smaller cube $[a, b]^n$. Denote these regions by $Q_1, \ldots, Q_M$ and let $\boldsymbol{x}^*$ satisfy equation equation 251 and $\boldsymbol{x}^* \in Q_i$. Since $\Psi(f_{\boldsymbol{v}})$ is affine in $Q_i$ and $Q_i$ intersects $[a, b]^n$ we have from equation equation 250 that

$$\Psi(f_{\boldsymbol{v}})(\boldsymbol{x}^*) = f_{\boldsymbol{v}}\left(\frac{\boldsymbol{x}^* - a}{b - a}\right), \tag{252}$$

which is a contradiction, thus $\Psi(f_{\boldsymbol{v}}) \notin \mathrm{PL}_{M,R}$. By Lemma F.13 we know that $\mathrm{PL}_{M,R}$ is closed with respect to the $\| \cdot \|_\infty$ norm and since $\Psi(f_{\boldsymbol{v}}) \notin \mathrm{PL}_{M,R}$ we have $C := d(\Psi(f_{\boldsymbol{v}}), \mathrm{PL}_{M,R}) > 0$. Since the set of functions realizable by architecture $A$ is a subset of $\mathrm{PL}_{M,R}$, equation equation 232 holds for any DWS model $\Phi$ completing the proof.

$\square$

We remark that the argument above suggests a broader phenomenon. In fact, there is a wide class of continuous function-space operators that we believe DWS models are unable to approximate when restricted to a fixed architecture. Examples include operators that sharpen contrast in pixel space for INRs, or operators that significantly modify decision boundaries in classifiers to accommodate for new data points, among many others.

### F.2.2. PROOF OF THEOREM F.10

To begin the proof, we first show that, in a neighborhood of any compact family of functions, the identity map can be uniformly approximated by a continuous function-space operator, implemented by neural networks of sufficiently large architecture. To formalize this neighborhood, we introduce the $\epsilon$-ball around a compact set.

**Definition F.14.** Let $K$ be a compact subset of a normed space $\mathcal{U}$, and let $\epsilon > 0$. We define the $\epsilon$-ball around $K$ as

$$B_\epsilon(K) := \{\boldsymbol{u} \in \mathcal{U} \mid \exists \boldsymbol{u}' \in K \text{ such that } \|\boldsymbol{u} - \boldsymbol{u}'\| < \epsilon\}. \tag{253}$$

**Lemma F.15.** *Let $K \subseteq \mathcal{C}(X, Y)$ be a compact set of functions. Then, for every $\epsilon > 0$, there exist an architecture $A = (\boldsymbol{d}, \sigma)$ and a continuous map $I : B_{\epsilon/4}(K) \to \mathcal{V}_A$ such that, for every $f \in K$,*

$$\|f - f_{I(f)}\|_\infty < \epsilon. \tag{254}$$

*Proof.* Let us fix $\epsilon > 0$. First, since $K$ is compact, there exist $f_1, \ldots, f_m \in K$ such that for every $f \in K$ there exists $i$ with

$$\|f - f_i\|_\infty < \epsilon/4. \tag{255}$$

Define open sets

$$U_i := \{f \in K : \|f - f_i\|_\infty < \epsilon/2\}. \tag{256}$$

The family $\{U_i\}_{i=1}^m$ is an open cover of $B_{\epsilon/4}(K)$. Since $B_{\epsilon/4}(K)$ is an open subset of a metric space, it is Hausdorff and paracompact. Therefore, it admits a continuous partition of unity $\{\varphi_i\}_{i=1}^m$ such that each $\varphi_i : B_{\epsilon/4}(K) \to [0,1]$ is continuous, $\operatorname{supp}(\varphi_i) \subseteq U_i$, and $\sum_{i=1}^m \varphi_i(f) = 1$ for all $f \in B_{\epsilon/4}(K)$.

Define

$$\Lambda(f) := \sum_{i=1}^m \varphi_i(f) \, f_i. \tag{257}$$

Then $\Lambda : B_{\epsilon/4}(K) \to \mathcal{C}(X, Y)$ is continuous, and for any $f \in B_{\epsilon/4}(K)$,

$$\|\Lambda(f) - f\|_\infty \leq \sum_{i=1}^m \varphi_i(f)\|f_i - f\|_\infty = \sum_{f \in U_i} \varphi_i(f)\|f_i - f\|_\infty \leq \sum_{f \in U_i} \varphi_i(f) \cdot \epsilon/2 \leq \epsilon/2. \tag{258}$$

Now, by universal approximation of MLPs, for each $i$ there exists an MLP with parameters $\boldsymbol{v}_i$ (possibly different architectures) such that

$$\|f_i - f_{\boldsymbol{v}_i}\|_\infty < \epsilon/2. \tag{259}$$

Choose a single architecture $A$ large enough so that each of these networks can be embedded as a subnetwork with zero-padded weights. We abuse notation and let $\boldsymbol{v}_i \in \mathcal{V}_A$ denote those embedded parameters. Then

$$\|f_i - f_{\boldsymbol{v}_i}\|_\infty < \epsilon/2 \quad \text{for all } i. \tag{260}$$

Define

$$\Xi(f) := \sum_{i=1}^m \varphi_i(f) \, f_{\boldsymbol{v}_i}. \tag{261}$$

Then $\Xi$ is continuous and

$$\|\Xi(f) - \Lambda(f)\|_\infty \leq \epsilon/2, \tag{262}$$

so

$$\|\Xi(f) - f\|_\infty \leq \epsilon. \tag{263}$$

Let $\mathcal{U} = \operatorname{span}\{f_{\boldsymbol{v}_1}, \ldots, f_{\boldsymbol{v}_m}\}$ and choose a basis $g_1, \ldots, g_r$ consisting of some of these functions.

Write

$$\Xi(f) = \sum_{j=1}^r c_j(f) g_j, \tag{264}$$

where since $\mathcal{U}$ is a finite-dimensional linear space, $c_j(f)$ are well defined and depend continuously on $f$. Construct a slightly larger architecture $\tilde{A}$ consisting of $r$ parallel copies of $A$ whose outputs feed into a final linear layer. Fix the subnetworks so that they compute $g_1, \ldots, g_r$, and the final layer parameters vary and may encode the coefficients $c_1, \ldots, c_r$.

Thus we can define a continuous affine map

$$L : \mathbb{R}^r \to \mathcal{V}_{\tilde{A}} \tag{265}$$

such that, for $\alpha = (\alpha_1, \ldots, \alpha_r)$,

$$f_{L(\alpha)} = \sum_{j=1}^r \alpha_j g_j. \tag{266}$$

Define
$$I(f) := L(c_1(f), \ldots, c_r(f)). \tag{267}$$

Then $I$ is continuous and
$$f_{I(f)} = \Xi(f), \tag{268}$$

so
$$\|f - f_{I(f)}\|_\infty < \epsilon. \tag{269}$$

Renaming $\tilde{A}$ as $A$ finishes the proof.

$\square$

The next lemma shows that continuous maps from function space to weight space can be realized, on the relevant set, by continuous equivariant maps that operate purely in weight space.

**Lemma F.16.** *Let $A = (d, \sigma)$ be an architecture with corresponding weight space $\mathcal{V}_A$, symmetry group $G_A$ and exclusion set $\mathcal{E}_A$ (see Definition 3.4), and let $\Psi : \mathcal{C}(X, Y) \to \mathcal{V}_A$ be continuous. Then, for any compact set $K^* \subseteq \mathcal{V}_A \setminus \mathcal{E}_A$ , there exists a continuous map*
$$\Psi^* : \bigcup_{g \in G} (g \cdot K^*) \to \mathcal{V}_A \tag{270}$$

*such that the following holds for all $v \in \bigcup_{g \in G} \rho(g) K^*$:*

1. *Equivariance:*
$$\Psi^*(\rho(g)v) = \rho(g)\Psi^*(v) \quad \forall g \in G_A. \tag{271}$$

2. *Functional equality:*
$$f_{\Psi^*(v)} = f_{\Psi(v)}. \tag{272}$$

*Proof.* For simplicity, assume without loss of generality that $K^* = \bigcup_{g \in G} \rho(g)K^*$. By Lemma F.6, there exists a continuous map $\mathrm{canon} : K^* \to K^*$ such that for every $v \in K^*$ and every $g \in G$:

$$\mathrm{canon}(v) = \mathrm{canon}(\rho(g)v) \quad \text{and} \quad \mathrm{canon}(v) \in \mathrm{orbit}(v). \tag{273}$$

Moreover, when constructing $\mathrm{canon}$, we arranged that, for each $v \in K^*$, there exists a *unique* $g_v \in G$ such that

$$\mathrm{canon}(v) = \rho(g_v)v. \tag{274}$$

For any $h \in G$ and $v' := h \cdot v$, we thus have

$$\mathrm{canon}(v) = \rho(g_{v'})v' = \rho(g_{v'})\rho(h)v. \tag{275}$$

From the uniqueness of $g_v$ we thus have
$$g_{\rho(h)v} = g_v h^{-1}. \tag{276}$$

Define $\Psi^* : K^* \to \mathcal{V}_A$ by
$$\Psi^*(v) := \rho((g_v)^{-1})\Psi\big(f_{\mathrm{canon}(v)}\big). \tag{277}$$

This is well-defined because $g_v \in G$ is uniquely determined by $v$. Additionally, as the maps $v \to \mathrm{canon}(v)$, $v \to f_v$ (where continuity follows from Proposition C.2) and $v \to g_v$ are all continuous on $K^*$, $\Psi^*$ is also continuous. Moreover, since $\mathrm{canon}(v) \in \mathrm{orbit}(v)$, we have $f_v = f_{\mathrm{canon}(v)}$ and so

$$\Psi^*(v) = \rho((g_v)^{-1})\Psi(f_v). \tag{278}$$

Since $\Psi^*(\boldsymbol{v}) \in \mathrm{orbit}(\Psi(f_{\boldsymbol{v}}))$, functional equality holds. Finally, from equation 276 we have

$$\Psi^*(\rho(h)\boldsymbol{v}) = \rho((g_{\rho(h)\boldsymbol{v}})^{-1})\Psi\big(f_{\mathrm{canon}(\rho(h)\boldsymbol{v})}\big) = \rho((g_{\boldsymbol{v}}h^{-1})^{-1})\Psi\big(f_{\mathrm{canon}(\boldsymbol{v})}\big) = \rho(h)\rho(g_{\boldsymbol{v}})^{-1}\Psi\big(f_{\mathrm{canon}(\boldsymbol{v})}\big) = \rho(h)\Psi^*(\boldsymbol{v}).$$
(279)

which is exactly the equivariance property, completing the proof.

$\square$

We are now ready for our final proof.

*Proof of Theorem F.10.* First, since $\Psi$ is continuous and $K$ is compact, the image $\Psi(K)$ is compact as well. Additionally, for any $\epsilon > 0$, there exists $\delta > 0$ such that
$$\Psi\big(B_\delta(K)\big) \subseteq B_{\epsilon/4}\big(\Psi(K)\big).$$
(280)

Now, Lemma F.15 states there exists a continuous map $I : B_{\epsilon/4}\big(\Psi(K)\big) \to \mathcal{V}_A$ such that, for every $f \in B_{\epsilon/4}\big(\Psi(K)\big)$,
$$\|f - f_{I(f)}\|_\infty < \epsilon.$$
(281)

Thus, the function $I \circ \Psi : B_\delta(K) \to \mathcal{V}_A$ is continuous and satisfies

$$\|f_{I \circ \Psi(f)} - \Psi(f)\|_\infty < \epsilon.$$
(282)

Now, from Lemma F.16 we get that for any compact set $K^* \subseteq \mathcal{V}_A \setminus \mathcal{E}_A$ that is a $\delta$ approximation of $K$, there exists a continuous equivariant map $\Psi^* : \bigcup_{g \in G} \rho(g)K^* \to \mathcal{V}_A$ such that

$$f_{\Psi^*(\boldsymbol{v})} = f_{I \circ \Psi(f_{\boldsymbol{v}})} \quad \forall \boldsymbol{v} \in \bigcup_{g \in G} \rho(g)K^*.$$
(283)

This means that for every $\boldsymbol{v} \in K^*$

$$\|f_{\Psi^*(\boldsymbol{v})} - \Psi(f_{\boldsymbol{v}})\|_\infty < \epsilon.$$
(284)

Since $\Psi^*$ is equivariant and continuous and $K^* \subseteq \mathcal{V}_A \setminus \mathcal{E}_A$, Theorem 7.4 implies that we can approximate $\Psi^*$ using a DWS network to any precision, completing the proof.

$\square$

# G. Exclusion Sets

Throughout our theoretical analysis, we frequently impose a general position (GP) assumption on the input weights, namely that all inputs $\boldsymbol{v}$ lie outside a designated exclusion set $\mathcal{E}$. For simplicity, our discussion primarily uses the bias exclusion set from Definition 3.4, denoted in this section by $\mathcal{E}^b$ (Since this is the only section discussing multiple exclusion sets, we omit the superscript $b$ elsewhere in the paper for notational simplicity). The condition $\boldsymbol{v} \notin \mathcal{E}^b$ requires that, within each hidden layer, all neuron biases are pairwise distinct. This is a natural assumption, as violations correspond to degenerate parameter configurations that are unlikely to arise under random initialization or stochastic training dynamics.

That said, the bias exclusion set is less suitable in settings where biases are shared (e.g., equivariant parameter sharing) or absent altogether, as in transformer or CNN weight-space architectures. Fortunately, our analysis relies only on more abstract properties of the exclusion set.

Specifically, we require the existence of a continuous canonization map (Definition B.4)

$$\mathrm{canon} : \mathcal{V} \setminus \mathcal{E} \to \mathcal{V}^d$$
(285)

such that, for every $\boldsymbol{v} \in \mathcal{V} \setminus \mathcal{E}$,

$$\text{canon}(\boldsymbol{v})_1 \in \text{orbit}(\boldsymbol{v}), \tag{286}$$

or equivalently,

$$\text{canon}(\boldsymbol{v})_1 = \rho(g^{\boldsymbol{v}})\boldsymbol{v} \tag{287}$$

for some permutation $g^{\boldsymbol{v}} = (\tau_1^{\boldsymbol{v}}, \ldots, \tau_L^{\boldsymbol{v}}) \in G$ depending on the input $\boldsymbol{v}$. Given such a canonization map, we further define an associated equivariant encoding that augments each parameter with its canonicalized representation and permutation-index information.

**Definition G.1** (equivariant canonization encoding map). For canonization map $\text{canon} : \mathcal{V} \setminus \mathcal{E} \to \mathcal{V}^d$ satisfying Equation 286, we define the associated equivariant canonization encoding map as follows.

For an input $\boldsymbol{v} = (\boldsymbol{W}_1, \boldsymbol{b}_1, \ldots, \boldsymbol{W}_L, \boldsymbol{b}_L)$, let $\widetilde{\text{canon}}(\boldsymbol{v}) = (\boldsymbol{W}_1', \boldsymbol{b}_1', \ldots, \boldsymbol{W}_L', \boldsymbol{b}_L')$, where for each layer $\ell$ and all valid indices $i, j$,

$$(\boldsymbol{W}_\ell')_{i,j,:} := \Big( (\boldsymbol{W}_\ell)_{i,j}, \; \ell, \; \tau_\ell^{\boldsymbol{v}}(i), \; \tau_{\ell-1}^{\boldsymbol{v}}(j), \; 1, \; \text{Flat}(\text{canon}(\boldsymbol{v})) \Big), \tag{288}$$

and

$$(\boldsymbol{b}_\ell')_{i,:} := \Big( (\boldsymbol{b}_\ell)_i, \; \ell, \; \tau_\ell^{\boldsymbol{v}}(i), \; 0, \; 0, \; \text{Flat}(\text{canon}(\boldsymbol{v})) \Big). \tag{289}$$

Our proofs require that $\widetilde{\text{canon}}$ can be approximated arbitrarily well on compact subsets of $\mathcal{V} \setminus \mathcal{E}$ by weight-space networks.

As shown in Lemmas F.6 and F.7, the bias exclusion set admits a continuous canonization map satisfying Equation 286, whose associated equivariant canonization encoding map satisfies the above approximation property. Importantly, this is the only property of the bias exclusion set used throughout our analysis. Consequently, the bias exclusion set may be replaced by any exclusion set satisfying the same conditions, yielding a broad family of valid alternatives. We next present one such family.

**Definition G.2** ($p$-norm exclusion set). For a given architecture $A$ and $p > 0$, define

$$\mathcal{E}_A^p := \Big\{ \boldsymbol{v} \in \mathcal{V}_A \; \Big| \; \exists \ell \in [L-1], \; \exists 1 \le i < j \le d_\ell \text{ such that } \|(\boldsymbol{W}_\ell)_{i,:}\|_p = \|(\boldsymbol{W}_\ell)_{j,:}\|_p \Big\}. \tag{290}$$

Equivalently, $\boldsymbol{v} \notin \mathcal{E}_A^p$ if, within every hidden layer, the rows of each weight matrix have pairwise distinct $p$-norms. As with the bias exclusion set, $\mathcal{E}_A^p$ has Lebesgue measure zero, and the condition $\boldsymbol{v} \notin \mathcal{E}_A^p$ therefore holds almost surely under random initialization or stochastic training dynamics.

Moreover, as the following proposition shows, each $\mathcal{E}_A^p$ admits a continuous canonization map satisfying Equation 286, whose associated equivariant canonization encoding map can be approximated arbitrarily well on compact sets by weight-space networks. Consequently, all of our theoretical results remain valid when the GP assumption $\boldsymbol{v} \notin \mathcal{E}$ is replaced by $\boldsymbol{v} \notin \mathcal{E}_A^p$ for any $p > 0$.

**Proposition G.3.** *Let $\mathcal{V}_A$ be a weight-space with associated exclusion set $\mathcal{E}_A^p$ for some $p > 0$. Then there exists a continuous canonization map $\text{canon} : \mathcal{V}_A \setminus \mathcal{E}_A^p \to \mathcal{V}_A^d$ satisfying Equation 286, such that, for every compact set $K \subseteq \mathcal{V}_A \setminus \mathcal{E}_A^p$, the associated equivariant canonization encoding map $\widetilde{\text{canon}}$ can be approximated arbitrarily well on $K$ by weight-space networks.*

*Proof.* The proof follows the same construction as Lemmas F.6 and F.7, with only a minor modification to the ranking function used to canonize neurons (Equation 161).

In the bias exclusion set construction, neuron ordering was determined by sorting the bias coordinates via

$$\text{rank}_\ell(i) := \big| \{ j \in [d_\ell] : (\boldsymbol{b}_\ell)_j < (\boldsymbol{b}_\ell)_i \} \big|. \tag{291}$$

Here, we instead define the ordering using the $p$-norms of the incoming weight vectors:

$$\text{rank}_\ell(i) := \left| \{ j \in [d_\ell] : \|(\boldsymbol{W}_\ell)_{i,:}\|_p^p < \|(\boldsymbol{W}_\ell)_{j,:}\|_p^p \} \right|. \tag{292}$$

Since $\boldsymbol{v} \notin \mathcal{E}_A^p$, these quantities are pairwise distinct within each layer, so the above defines a valid permutation ranking of the neurons.

Moreover, this ranking can be computed by standard weight-space architectures such as DWS networks. Indeed, one may first apply the pointwise transformation

$$\boldsymbol{W}_\ell \mapsto |\boldsymbol{W}_\ell|^p, \tag{293}$$

followed by an equivariant aggregation layer that computes, for each neuron, the sum of its incoming transformed weights,

$$\sum_k |(\boldsymbol{W}_\ell)_{i,k}|^p. \tag{294}$$

The remainder of the canonization construction then proceeds exactly as in Lemmas F.6 and F.7. Consequently, the associated equivariant canonization encoding map can again be approximated arbitrarily well on compact sets by weight-space networks. $\qquad\square$

### G.1. Empirical validation of the general position assumption

The GP assumption $\boldsymbol{v} \notin \mathcal{E}$ underlies our universality results for both invariant functionals (Theorem 6.3) and equivariant operators (Theorem 7.4), while Proposition 6.2 shows that expressivity may degrade once this assumption fails. Although $\mathcal{E}$ has Lebesgue measure zero and is therefore null under any absolutely continuous weight distribution, the practical relevance of the assumption is a separate empirical question: how often do trained weights actually lie inside an exclusion set? In this subsection we address this question on a standard weight-space dataset and use the multi-exclusion-set theory developed above to obtain a more refined picture.

**Setup.** We use the MNIST INR dataset of Navon et al. (2023), which consists of 60,000 pretrained INRs, one per MNIST training image. Each INR is an MLP with 3 layers and widths $\boldsymbol{d} = (2, 32, 32, 1)$. For a quantization level of $q$ bits we apply uniform mid-rise bias quantization to each layer independently: the bias entries of layer $\ell$ are mapped to the closest bin of $2^q$ bins built over $[\min(\boldsymbol{b}_\ell), \max(\boldsymbol{b}_\ell)]$. The unquantized network is treated as the $q = \infty$ baseline.[6]

For each quantized network $\boldsymbol{v}$ and each exclusion set $\mathcal{E}_\star$ under consideration, we record whether $\boldsymbol{v} \notin \mathcal{E}_\star$ (i.e., whether $\boldsymbol{v}$ is in GP with respect to $\mathcal{E}_\star$), and report the fraction of the 60,000 INRs that satisfy this condition.

**Bias exclusion set vs. quantization.** We first measure the GP rate with respect to the bias exclusion set $\mathcal{E}^b$ of Definition 3.4, as a function of the bias quantization precision $q$. The results are summarized in Figure 5. In the unquantized regime essentially all networks satisfy GP (100% at full precision, 97.98% at 16-bit), confirming that GP is the typical case under standard training dynamics. Violations begin to appear at 12-bit (72.66% of INRs still in GP) and become severe at 8-bit and below, where the GP rate collapses to below 1%. The renderings below the curve show that the quantized INRs continue to represent the original image reasonably well throughout this range, indicating that the loss of GP at low precision is not merely an artifact of the network ceasing to be a useful representation. Together with Proposition 6.2, this provides empirical support for the prediction that aggressive bias quantization is a regime in which fixed-architecture weight-space networks may suffer from expressivity limitations.

**Combining exclusion sets** As shown in Proposition G.3, the bias exclusion set $\mathcal{E}^b$ is only one element of a much broader family of admissible exclusion sets: each $p$-norm exclusion set $\mathcal{E}^p$ (Definition G.2) admits a continuous canonization map approximable by weight-space networks, and our universality results hold under any of them. Since our proofs only require the existence of *some* exclusion set $\mathcal{E}_\star$ with $\boldsymbol{v} \notin \mathcal{E}_\star$, a natural empirical question is the fraction of weights that lie outside the *union of complements*, i.e. the fraction of $\boldsymbol{v}$ for which $\boldsymbol{v} \notin \mathcal{E}_\star$ for at least one $\mathcal{E}_\star$ in our family.

---

[6]We focus on bias quantization since the bias exclusion set $\mathcal{E}^b$ of Definition 3.4 is determined by bias values; an analogous protocol on weight rows is reflected in the $p$-norm exclusion sets reported below.

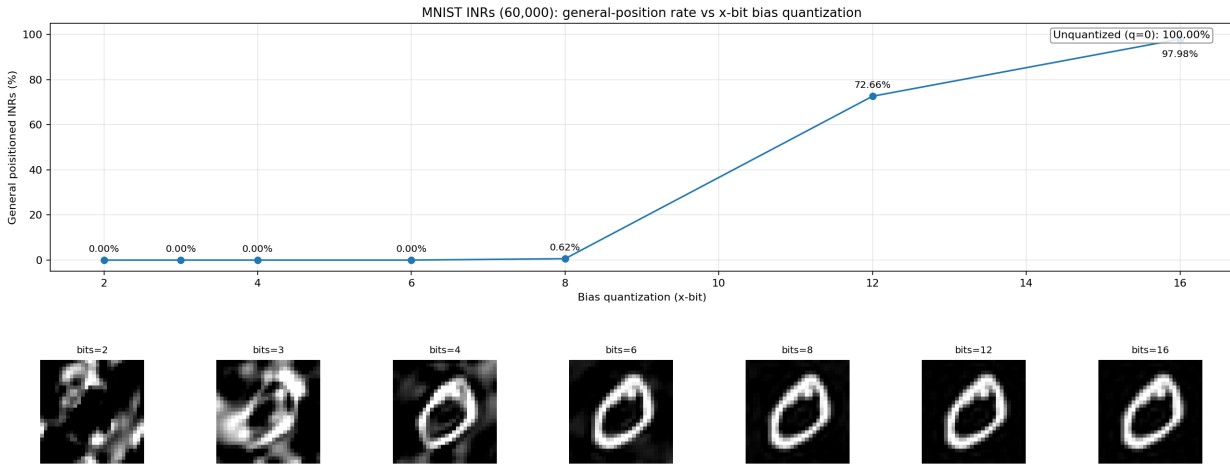

*Figure 5.* Percentage of MNIST INRs satisfying the bias-based GP assumption $\boldsymbol{v} \notin \mathcal{E}^b$ (Definition 3.4), as a function of bias quantization precision $q$. The bottom row shows reconstructions of a single representative INR at each quantization level. At standard precision GP holds for nearly all networks; the GP rate begins to drop at 12-bit and collapses by 8-bit, while the underlying INR remains a visually faithful representation of the digit.

Concretely, we consider the increasing family of GP conditions

$$\mathrm{GP}_P(\boldsymbol{v}) \coloneqq \left(\boldsymbol{v} \notin \mathcal{E}^b\right) \vee \bigvee_{p \in \{1,\ldots,P\}} \left(\boldsymbol{v} \notin \mathcal{E}^p\right), \qquad P \in \{0,1,2,3,4,5\}, \tag{295}$$

where $P = 0$ refers to the bias exclusion set criterion of Definition 3.4 and increasing $P$ progressively incorporates the $\ell_p$-norm criteria of Definition G.2. A network is declared to satisfy $\mathrm{GP}_P$ if it is in general position with respect to *at least one* of the exclusion sets in the family. Since our theory applies separately under each such exclusion set, $\mathrm{GP}_P$ is a sufficient condition for the conclusions of Theorems 6.3 and 7.4 to hold.

Figure 6 reports $\mathrm{GP}_P$ rates on the same quantized MNIST INRs. Two qualitative observations emerge. First, broadening the family of exclusion sets meaningfully improves the GP rate at intermediate precision: at $q = 8$ bits, the bias criterion alone is satisfied by only $0.62\%$ of networks, while the combined criterion $\mathrm{GP}_5$ raises this to roughly $80\%$. Second, the improvement saturates rapidly with $P$, and the qualitative dependence on quantization precision is preserved: at $q \leq 6$ bits, even the combined family $\mathrm{GP}_5$ leaves a substantial fraction of networks unaccounted for.

**Takeaways.** The two experiments above provide an initial practical picture of the GP assumption. (i) For standard MLP weights trained at typical precision, GP holds for essentially all networks already under the simplest bias-based exclusion set, so our universality results apply directly. (ii) In the extreme low-precision regime, neither a single nor a combined finite family of measure-zero exclusion sets can cover the data. This suggests that learning over heavily quantized weight spaces is a setting in which more expressive architectures may be needed, and we view a thorough investigation of this regime as an interesting direction for future work.

## H. Transformer-Weight-Space Architectures

In this work, we primarily focus on weight-space architectures operating on MLP parameters. This choice is motivated both by clarity of presentation, given our already dense technical development, and by the fact that MLP-based weight-space architectures are currently the most mature and extensively studied setting. Moreover, MLPs remain fundamental building blocks in many modern neural architectures.

Nevertheless, weight-space architectures for transformers, CNNs, RNNs, and other neural network families have also been proposed (Kofinas et al., 2024; Lim et al., 2023; Zhou et al., 2023a). In this section, we demonstrate that our analysis naturally extends beyond the MLP setting. In particular, we consider Graph Meta-Networks (GMNs) (Lim et al., 2023) operating on transformer weight spaces and establish universality results in both the invariant and equivariant settings under a suitable general-position assumption. Importantly, the required modifications to the proofs of Theorems 6.3 and 7.4 are minimal.

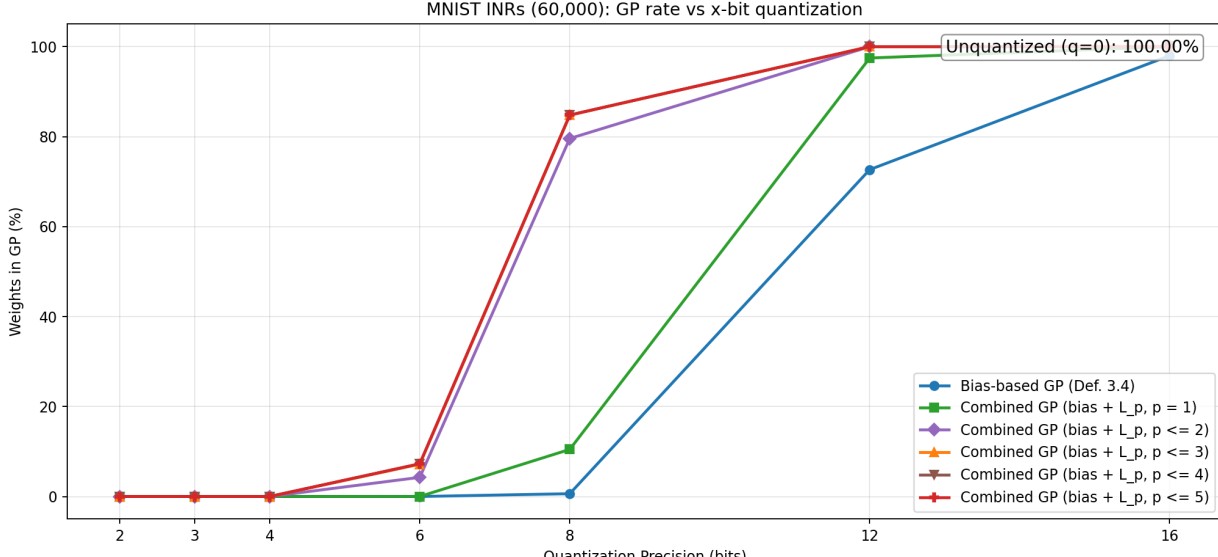

*Figure 6.* Percentage of MNIST INRs satisfying the combined GP criterion $\mathrm{GP}_P$ from Equation equation 295, evaluated on a monotonically growing family of exclusion sets that begins with the bias-based criterion of Definition 3.4 and progressively adds the $\ell_p$-norm criteria of Definition G.2 for $p \in \{1, 2, 3, 4, 5\}$. A network is marked as in GP if it lies outside at least one exclusion set in the family. Enlarging the family substantially improves the GP rate at intermediate precision, but the dependence on quantization persists and the improvement plateaus with $P$.

We begin by defining an appropriate exclusion set for transformer weight spaces. Recall that a transformer consists of four main component types: MLP blocks, multi-head attention layers, normalization layers, and residual connections. GMNs encode transformer weights as parameter graphs, where each parameter corresponds to an edge equipped with structural features encoding the layer depth and component type.

**Definition H.1** (Transformer exclusion set). Let $A$ be a transformer architecture composed of $L$ components, where each component is either an MLP block, a multi-head attention layer, a normalization layer, or a residual connection. The transformer exclusion set is defined by

$$\mathcal{E}_A = \bigcup_{\ell=1}^{L} \mathcal{E}_\ell, \tag{296}$$

where for each $\ell \in [L]$:

1. If component $\ell$ is an MLP block, then $\mathcal{E}_\ell$ is defined exactly as in Definition 3.4.

2. If component $\ell$ is a multi-head attention layer with hidden dimension $d$, then

$$\mathcal{E}_\ell = \mathcal{E}_\ell^V \cup \mathcal{E}_\ell^O, \tag{297}$$

   where

$$\mathcal{E}_\ell^V := \left\{ \boldsymbol{v} \in \mathcal{V}_A \;\middle|\; \exists\, 1 \le i < j \le d \text{ such that } \sum_k \left|(\boldsymbol{W}_V^1)_{i,k}\right| = \sum_k \left|(\boldsymbol{W}_V^1)_{j,k}\right| \right\}, \tag{298}$$

   with $\boldsymbol{W}_V^1$ denoting the value projection matrix of the first attention head, and

$$\mathcal{E}_\ell^O := \left\{ \boldsymbol{v} \in \mathcal{V}_A \;\middle|\; \exists\, 1 \le i < j \le d \text{ such that } \sum_k \left|(\boldsymbol{W}_O)_{i,k}\right| = \sum_k \left|(\boldsymbol{W}_O)_{j,k}\right| \right\}, \tag{299}$$

   where $\boldsymbol{W}_O$ denotes the output projection matrix of the attention layer.

3. If component $\ell$ is a residual connection or normalization layer, then

$$\mathcal{E}_\ell = \emptyset. \tag{300}$$

The above definition can be viewed as a combination of several valid exclusion-set constructions discussed in Appendix G. Importantly, $\mathcal{E}_A$ remains a finite union of lower-dimensional sets and therefore has measure zero. Consequently, transformer weights achieved by training with random initialization or stochastic gradient decent lie outside $\mathcal{E}_A$ with probability one.

We now show that, under the general-position assumption $\boldsymbol{v} \notin \mathcal{E}$, GMNs are universal approximators of equivariant [7] operators on transformer weight spaces.

**Theorem H.2.** *Let $\mathcal{V}$ denote a transformer weight space, and let $K \subset \mathcal{V} \setminus \mathcal{E}_A$ be compact. Then every equivariant $\Psi : \mathcal{V} \to \mathcal{V}^n$ can be approximated arbitrarily well on $K$ by GMN networks.*

*Proof.* The proof follows the same overall strategy as in the MLP setting, with only minor modifications to the canonization procedure.

Recall that the proof of Theorem 7.4 relies on the existence of a continuous canonization map (Definition B.4)

$$\mathrm{canon} : \mathcal{V} \setminus \mathcal{E} \to \mathcal{V}^d \tag{301}$$

satisfying

$$\mathrm{canon}(\boldsymbol{v})_1 \in \mathrm{orbit}(\boldsymbol{v}), \tag{302}$$

or equivalently,

$$\mathrm{canon}(\boldsymbol{v})_1 = \rho(g^{\boldsymbol{v}})\boldsymbol{v} \tag{303}$$

for some permutation $g^{\boldsymbol{v}} \in G$ depending continuously on $\boldsymbol{v}$. Moreover, the proof requires that the associated equivariant canonization encoding map (Definition G.1)

$$\mathrm{can\bar{o}n} : \mathcal{V} \setminus \mathcal{E} \to \mathcal{V}^{d^*} \tag{304}$$

can itself be approximated by GMNs on compact subsets. Thus, it suffices to establish analogous properties in the transformer setting. The construction of the canonization map proceeds exactly as in Lemmas F.6 and F.7, with the only modification being the ranking function used to order neurons (Equation 161).

In the MLP case, neurons were ordered using the bias-based ranking

$$\mathrm{rank}_\ell(i) := \left| \{ j \in [d_\ell] : (\boldsymbol{b}_\ell)_j < (\boldsymbol{b}_\ell)_i \} \right|. \tag{305}$$

For transformer parameter graphs, we retain the same ranking for nodes corresponding to MLP components. We now extend this ranking to nodes associated with multi-head attention layers.

Recall that in the GMN parameter graph representation, a multi-head attention layer is encoded as two consecutive graph layers. The incoming nodes of the first graph layer inherit a canonical ordering from the preceding component and therefore require no additional processing.[8] For the outgoing nodes of the first graph layer (equivalently, the incoming nodes of the second graph layer), we define

$$\mathrm{rank}_\ell(i) := \left| \left\{ j \in [d] : \sum_k \left| (\boldsymbol{W}_V^1)_{i,k} \right| < \sum_k \left| (\boldsymbol{W}_V^1)_{j,k} \right| \right\} \right|. \tag{306}$$

Similarly, for the outgoing nodes of the second graph layer, we define

$$\mathrm{rank}_\ell(i) := \left| \left\{ j \in [d] : \sum_k \left| (\boldsymbol{W}_O)_{i,k} \right| < \sum_k \left| (\boldsymbol{W}_O)_{j,k} \right| \right\} \right|. \tag{307}$$

Since $\boldsymbol{v} \notin \mathcal{E}_A$, all quantities above are pairwise distinct within each layer. Consequently, these rankings induce valid and continuous neuron orderings.

---

[7]Here equivariance is taken with respect to the group $G$ of automorphisms shared by all transformer parameter graphs of a given transformer architectures. For more details on these automorphisms see Lim et al. (2023).

[8]If the first component of the transformer is a multi-head attention layer, then these nodes are never permuted by the group action and no canonization is necessary.

Residual connections only introduce additional edges into the parameter graph and therefore require no additional ordering. Likewise, each normalization layer contributes only a single marked node, which is already canonically identified.

It remains to show that these rankings can be computed by weight-space architectures such as GMNs. For MLP components, this follows exactly as in Lemma F.7. For attention components, we first apply the pointwise transformation

$$\boldsymbol{W}_V^1 \mapsto |\boldsymbol{W}_V^1|, \qquad \boldsymbol{W}_O \mapsto |\boldsymbol{W}_O|, \tag{308}$$

followed by an equivariant aggregation layer computing

$$\sum_k \left|(\boldsymbol{W}_V^1)_{i,k}\right| \qquad \text{and} \qquad \sum_k \left|(\boldsymbol{W}_O)_{i,k}\right|. \tag{309}$$

The remainder of the construction is identical to the MLP case. In particular, the resulting canonization map is continuous, and the associated equivariant canonization encoding map can be approximated arbitrarily well on compact subsets by GMN networks. The conclusion therefore follows exactly as in the proof of Theorem 7.4. $\square$

As an immediate corollary, we obtain universality for invariant functionals on transformer weight spaces.

**Corollary H.3.** *Let $\mathcal{V}$ be a transformer weight space, and let $K \subset \mathcal{V} \setminus \mathcal{E}_A$ be a compact set. Then every invariant functional $\Psi : \mathcal{V} \to \mathbb{R}^n$ can be approximated arbitrarily well on $K$ by GMN networks.*

*Proof.* The proof is identical to that of Theorem E.11, using the standard reduction from the equivariant setting to the invariant setting. $\square$

These results demonstrate that our framework naturally extends beyond MLP weight spaces and applies to substantially richer neural architectures, including transformers.

## I. Experimental Details

In this section we provide full experimental details for the experiments reported in Section 8. All experiments are run on a single NVIDIA A100-SXM4 GPU with 40 GB of memory. Code, configurations, and logs are available at https://github.com/dayanadir/capacity_increase_inr_editing_experiment.

### I.1. OCE

All permutation-equivariant weight-space networks considered in this work naturally support outputs in the product weight space $\mathcal{V}^d$, rather than only in $\mathcal{V}$. Motivated by Theorem 7.2, we leverage this property to increase the effective output capacity of existing weight-space architectures.

Given any weight-space architecture, its $k$-OCE variant is obtained by modifying the output space from $\mathcal{V}$ to $\mathcal{V}^d$. We then interpret an output $\boldsymbol{v} = (\boldsymbol{v}_1, \ldots, \boldsymbol{v}_k) \in \mathcal{V}^k$ as an ensemble of $k$ neural networks sharing the same architecture. Thus, instead of predicting a single output network, the modified model predicts an ensemble of $k$ networks.

Given an input $x$, the final prediction is obtained by averaging the outputs of the predicted networks:

$$f_{\boldsymbol{v}}(x) = \frac{1}{k} \sum_{i=1}^k f_{\boldsymbol{v}_i}(x). \tag{310}$$

The parameter $k$ controls the effective output capacity of the model and serves as a hyperparameter. We evaluate OCE on two representative permutation-equivariant weight-space architectures: DWSNet (Navon et al., 2023) and GMN (Lim et al., 2023). For each netowrk we sweep $k \in \{1, 2, 4, 8\}$. To isolate the effect of *output-side* capacity from sheer parameter growth, we shrink the internal hidden widths so that the total parameter count of each $k > 1$ configuration does not exceed that of the $k = 1$ baseline.

## I.2. INR dilation task

**Task.** The task is the MNIST INR *dilation* benchmark introduced by Zhou et al. (2023a). Each datapoint consists of an input INR $f_v$ representing an MNIST digit and a target image $I^{\text{dil}} \in \mathbb{R}^{H \times W}$ obtained by applying a morphological *dilation* to the original digit. The goal is to learn a weight-space operator $\Phi : \mathcal{V}_A \to \mathcal{V}_A$ such that the INR realized by the predicted weights renders the dilated image, i.e. $\mathcal{R}(\Phi(v)) \approx I^{\text{dil}}$. This is a *function-space operator* (item 3 of Definition 4.1): the target depends only on the function realized by the input weights, not on their specific parameterization. As such, it falls precisely under the setting analyzed in Section 7.1.

**Input architecture.** The input INRs are taken from the publicly released MNIST INR dataset of Navon et al. (2023). Each INR is a SIREN (Sitzmann et al., 2020) with $L = 3$ layers and widths $d = (2, 32, 32, 1)$, so that

$$\mathcal{V}_A \cong \mathbb{R}^{32 \times 2} \oplus \mathbb{R}^{32} \oplus \mathbb{R}^{32 \times 32} \oplus \mathbb{R}^{32} \oplus \mathbb{R}^{1 \times 32} \oplus \mathbb{R}^1.$$

The dataset contains one pretrained INR per MNIST image; we use the standard training/testing split inherited from MNIST.

**Target generation.** For each MNIST image $I$, we generate the target $I^{\text{dil}}$ by applying a $3 \times 3$ morphological dilation (`cv2.dilate` with a $3 \times 3$ kernel of ones and a single iteration). The operator acts at the pixel level on the original $28 \times 28$ MNIST image and is held fixed across the entire dataset.

**Loss.** Models are trained with the rendered-image MSE loss

$$\mathcal{L}(v, I^{\text{dil}}) = \left\| \text{RENDER}(\Phi(v)) - I^{\text{dil}} \right\|_2^2,$$

where $\text{RENDER}(\cdot)$ evaluates the predicted INR on the $28 \times 28$ pixel grid using a vectorized SIREN renderer. We follow the residual parameterization used in Navon et al. (2023) and predict an additive update $\Delta v$ scaled by a learnable scalar; the final prediction is $\Phi(v) = v + s \cdot \Delta v$. Test performance is reported as the rendered MSE on the held-out MNIST INR test split.

## I.3. Baselines

The main results in Section 8 contrast our capacity-increased models with two groups of baselines:

**Same-architecture baselines ($k = 1$).** The DWS and GMN configurations at $k = 1$ serve as the natural baselines for the capacity-increase ablation, since they match the output architecture to the input architecture exactly. To rule out that improvements stem from extra capacity rather than from increased output capacity in particular, the $k > 1$ variants are constrained to use *no more* parameters than these baselines. The reported numbers for DWS ($k = 1$) and GMN ($k = 1$) are reproduced from Gelberg et al. (2026) under our training setup.

**Prior weight-space methods.** In Table 1 we also compare against published numbers for prior weight-space methods on the same MNIST INR dilation benchmark: NFT (Zhou et al., 2023b), NP-NFN, NG-GNN-64, and NG-T-64 (Kofinas et al., 2024), ScaleGMN-B (Kalogeropoulos et al., 2024), and ScaleGMN combined with GradMetaNet++ (Gelberg et al., 2026). These numbers are taken verbatim from the references and serve to place our capacity-increased models in the context of the published state of the art.

## I.4. Data preparation

**Splits.** We use the standard MNIST train/test split. From the training INRs we hold out a validation set of 5000 examples for model selection; test performance is reported on the official MNIST INR test split.

## I.5. Training details

**Optimization.** All models are trained for 150 epochs using AdamW with learning rate $10^{-3}$, weight decay $10^{-3}$, and batch size 32. We use a scheduler that halves the learning rate after 5 epochs without improvement on the training loss, with a floor of $10^{-4}$.

**Random seeds and reporting.** Each configuration is trained with three random seeds. For each run we select the epoch achieving the best validation MSE and report the corresponding test MSE. We then report the mean and standard deviation across the three seeds.

*Table 2.* Ablation of ensemble size on the MNIST INR dilation benchmark averaged over 3 random seeds. The best results are highlighted in bold.

| | DWS | | GMN | |
|---|---|---|---|---|
| **Output feat. dim** | **MSE** $(10^{-2})$ | $\Delta\%$ | **MSE** $(10^{-2})$ | $\Delta\%$ |
| 1 (baseline) | $2.29 \pm 0.01$ | – | $1.96 \pm 0.02$ | – |
| 2 | $2.01 \pm 0.01$ | $-12\%$ | $1.69 \pm 0.06$ | $-14\%$ |
| 4 | $1.58 \pm 0.00$ | $-31\%$ | $1.19 \pm 0.03$ | $-39\%$ |
| 8 | $\mathbf{1.36 \pm 0.03}$ | $-41\%$ | $\mathbf{1.06 \pm 0.13}$ | $-46\%$ |

**Compute.** A full training run takes between 4 and 10 hours on a single NVIDIA A100-SXM4-40GB GPU, depending on the network and the value of $k$. The total compute used to produce Tables 2 and 1 is approximately 200 A100-GPU-hours.

### I.6. Additional results and extended discussion

We study two main questions: (i) how the output feature dimension $k$, corresponding to the number of predicted ensemble models, affects performance, and (ii) how OCE compares to existing state-of-the-art weight-space approaches on function-space operator tasks.

To study the effect of output capacity, we evaluate OCE with varying output feature dimensions $k$ on two representative permutation-equivariant weight-space architectures: DWS and GMN. Results are reported in Table 2. Across both architectures, increasing the output feature dimension consistently improves performance. These findings are aligned with our theoretical results: Proposition 7.1 shows that existing weight-space networks are fundamentally limited when constrained to output networks with the same architecture as the input, while Theorem 7.2 suggests that increasing output capacity can overcome this limitation. Importantly, to ensure a fair comparison across different values of $d$, all models were adjusted to maintain approximately the same total number of parameters. Consequently, the observed improvements cannot be attributed simply to larger model size, but rather to the increased expressivity afforded by higher-capacity output representations.

We additionally compare DWS+OCE and GMN+OCE with output feature dimension $d = 8$ against prior state-of-the-art weight-space models. Several competing approaches leverage auxiliary information unavailable to OCE, including gradient information (Gelberg et al., 2026) and probing-based features (Kofinas et al., 2024). Despite relying only on standard weight-space inputs, GMN+OCE achieves state-of-the-art performance, improving over the previous GMN baseline by up to $34\%$ (Table 1).

Taken together, these results suggest that the output capacity of weight-space networks may constitute a significant bottleneck for function-space operator tasks. More broadly, they support the practical relevance of Theorem 7.2, indicating that mapping to larger output architectures can serve as a useful design principle for future weight-space models.

