# OpenReview forum: "On the Expressive Power of Permutation-Equivariant Weight-Space Networks"
_ICML.cc/2026/Conference — ICML 2026 spotlight_

### Official Review · Reviewer_M5tJ · 2026-02-27

**Soundness:** 4
**Presentation:** 4
**Significance:** 3
**Originality:** 4
**Overall Recommendation:** 5
**Confidence:** 3

**Summary:**

In this paper, the authors study theoretically the expressive power of permutation-equivariant models in weight space learning.

To do so, they first categorise target functions to be approximated in four categories, depending on their input and output spaces. Then, they show that the set of permutation-equivariant weight space models they study are equivalent in terms of their expressive power. Based on this finding, and on the pre-existing proof that one of the models in the equivalence class can approximate the forward pass of the neural network studied, they further show that these expressively equivalent models are, under some conditions, universal approximators for all four function categories.

**Compliance With Llm Reviewing Policy:**

Affirmed.

**Final Justification:**

This paper presents a solid theoretical exploration of the expressivity of equivariant weight space models. Like mentioned in my original review, I found this paper original, very well-written (in particular for such a theory-heavy work) and significant to the field of weight space learning. My concerns were mostly about limitations in scope of this work, and lack of generality of the original definition of the General Position assumption.

The authors made a very good rebuttal, and engaged well with the questions of the different reviewers, including mine. They have addressed my concerns, and I believe this has significantly improved the quality of this paper. For this reason, I am happy to increase my score from **4: Weak accept** to **5: Accept**.

**Key Questions For Authors:**

**Q1 - Scope:** Related to my points W1 and W2, what do the authors think about the applicability of their findings to more practical settings where models studied go beyond the MLP, or may not fulfill the general position assumption?

**Q2 - Relation to non-equivariant models:** What would be the expressive power of weight space models that are not equivariant to permutations, such as in Schürholt et al. [3]? Would the authors expect such models to be more, less or equivalently expressive, and why?

**Limitations:**

While the authors list a number of interesting and adequate directions for future work not directly covered in this paper, my opinion is that they do not sufficiently discuss limitations in terms of scope or applicability to practical settings. Because these limitations are insufficiently discussed, the practical impact and utility of this paper remains unclear.

**Strengths And Weaknesses:**

**S1 - Originality:** I think this work is both original and impactful, as the theoretical exploration of the expressivity of weight-space models has not, to the best of my knowledge, been studied. Characterising the possibilities, limitations and equivalences offered by different approaches is a very valuable contribution to the field of weight space learning.

**S2 - Significance:** Showing that the set of weight space models studied are expressively equivalent, and that this equivalence class is under mild conditions an universal approximator for all categories of functions studied is a major result. This will help the development of weight space learning by showing that differences in performance between these models is a question of training and design, not expressive power.

**S3 - Presentation:** The overall organisation of the paper, the introduction of core concepts and notations, as well as the writing are all very good. Proof sketches in the main text are very helpful in understanding what concepts are being used, while the more formal proof itself is reserved to the Appendix. Unfortunately, the length and amount of proofs make it difficult to check all of them carefully.

**W1 - Scope:** This paper’s scope is limited to the study of weight space models operating on MLP weights. While it is understandable to focus on those if they are the most theoretically studied, recent empirical developments have gone beyond the MLP architecture. For example, Kofinas et al. [1] work on CNNs, Lim et al. [2] on both CNNs and Transformers, Schürholt et al. [3] on ResNet. Although it is true that convolutional layers can be represented using dense layers with more parameters, I would argue that in most of such cases the general position assumption would not hold because of shared bias parameters. In addition, any neural network architectures using max-pooling or normalisation layers cannot be represented as a MLP. Because of this, I believe the scope of validity of the proofs in this paper may not extend to those more complex architectures, which may limit the applicability of the findings in this paper to empirical settings.

**W2 - General position:** Although the authors describe the exclusion set corresponding to the general position assumption as unlikely, I would argue the contrary. In addition to the case of convolutional layers raised in W1, this could easily arise if a single layer in the model does not contain bias terms (or equivalently, they are all zero), an example being residual connections. Besides, proofs on approximation universality with regard to permutation-invariant functionals and permutation-equivariant operators base themselves on the general position assumption to build a canonisation map based on the sorting of the bias parameters. Because of the existence of such a canonisation map, I would assume the equivalence class of models approximating such functions to be very large, and to extend well beyond equivariant architectures by simply working on a canonical version of the MLP studied. It would be helpful to discuss the prevalence of the general position assumption in real-world models, or to explore how the results might extend to cases where it does not hold.

**W3 - Discussion:** I feel like the contributions made by this paper are insufficiently discussed by the authors. Because of the purely theoretical nature of this work, my opinion is that the authors could discuss practical implications more. In particular, related to my two previous points, I think that discussing the application scope of this work would be helpful. Finally, it would be helpful in contextualising this work better to discuss links with different weight space methods, in particular the ones that are not permutation-equivariant.

Overall, I think this paper is a solid contribution to the field of weight space learning, but am somewhat concerned about the scope in which it is valid. I look forward to the discussion period to read the authors’ response to my concerns.

**References**

[1] Kofinas, M., Knyazev, B., Zhang, Y., Chen, Y., Burghouts, G. J., Gavves, E., Snoek, C. G., and Zhang, D. W. Graph neural networks for learning equivariant representations of neural networks. arXiv preprint arXiv:2403.12143, 2024.

[2] Lim, D., Maron, H., Law, M. T., Lorraine, J., & Lucas, J. (2023). Graph metanetworks for processing diverse neural architectures. arXiv preprint arXiv:2312.04501.

[3] Schürholt, K., Mahoney, M. W., & Borth, D. (2024). Towards scalable and versatile weight space learning. arXiv preprint arXiv:2406.09997.

---

> ### Author Rebuttal · Authors · 2026-03-30
>
> We thank the reviewer for their positive assessment, highlighting our work as “original and impactful,” a “very valuable contribution”, and noting its strong organization and clarity. We address their comments below.
>
>
> **On the general position (GP) assumption and the scope of our work (W1, W2, Q1):**
> We thank the reviewer for raising these important points. We agree that representing complex layers (e.g., convolutional or residual) as dense layers may introduce duplicate weights and biases. In such cases, Prop. 6.2 implies that WS networks processing these architectures may exhibit expressivity limitations.
>
>
> However, as we validate below, standard MLPs (only linear layers, biases and activations), under standard initialization, satisfy GP with high probability.
> Additionally, modern WS architectures that operate on richer layer classes (e.g., [3]) do not use dense computational graph encodings, but represent models as parameter graphs where each weight corresponds to a single edge.  In this setting, GP naturally extends to components such as attention, residual, and convolutional layers, allowing our framework to cover broader architectures. Indeed, as noted in Footnote 2 (page 4), the GP definition used in the paper is chosen for simplicity, and our theory extends to a broad class of alternatives that do not rely on bias terms.
>
>
> Extending the definition of GP allows us to extend our theory to more modern architectures. To support this, we will include in the revision an analogue of Thm. 6.3 **establishing universality under GP for GMNs [3] operating on transformer weights**, via the same proof strategy. Due to space constraints, we refer to our response to Reviewer Lro2 for a proof sketch and discussion, and to Reviewer r4xm for alternative GP definitions.
>
>
> **On practical implications and empirical validation (W3):**
> While primarily theoretical, our work yields concrete practical guidelines, which we will emphasize in the revision. Below we highlight two key insights; for further details, see our response to Reviewer HYpN.
>
>
> - **Increasing output architecture capacity.**
> As shown by  Prop. 7.1, weight-space networks are limited in approximating function-space operators since their output weights must match the architecture of the input. Thm. 7.2 shows that this limitation can be mitigated by increasing architecture size, motivating WS models that map to larger output architectures than their inputs. We test a simple and general modification to any weight space network: use output dimension k>1 and interpret the output as an ensemble of $k$ MLPs, increasing the output capacity, with predictions averaged across the ensemble. Evaluating on the prominent  MNIST INR dilation benchmark [2] for k=1,2,4,8 ([seen here](https://anonymous.4open.science/r/anonymusicml2026-3A06/experiment_results_mnist_inr_dilation.png)) shows significant improvements as k increases, with up to a 46% MSE reduction. Notably, by using  k=8 **we achieve SOTA performance**, while **maintaining comparable compute and parameter counts**.
>
>
> -  **Expressivity limitations under quantization.**
> In Prop. 6.2, we show that expressivity degrades when inputs are not in GP.  We hypothesized that while GP almost surely holds for standard MLPs, it may fail in quantized networks, causing expressivity limitations. We test this on the MNIST INR dataset and measure GP violations across quantization levels. Results ([seen here](https://anonymous.4open.science/r/anonymusicml2026-3A06/gp_rate_vs_xbit_bias_quantization.png)) show **all networks satisfy GP at high precision**, but violations begin to appear at 12-bit and grow pronounced at 8-bit, motivating more expressive architectures for these regimes.
>
>
> **On the relation to non-equivariant models (Q2, W3):**
> We thank the reviewer for raising this question. Equivariant models are typically preferred for processing data with symmetries due to their strong generalization properties, though they may have limited expressivity. In contrast, non-equivariant approaches (e.g., [1]) often process these inputs with standard universal architectures (e.g., MLPs or Transformers). These methods are thus inherently universal but sacrifice inductive bias.
>
>
> In our setting, we show that equivariant weight-space models do not suffer from this limitation and are universal in the relevant regimes under GP. The main limitation arises in the function-space operator setting, where expressivity is constrained. As shown above, increasing capacity in this regime leads to significant performance gains.
>
>
> We believe these additions clarify applicability and strengthen practical relevance.  We hope our response addresses the reviewer’s concerns and would appreciate a reconsideration of the score.
>
> [1] Schürholt et al. *Towards scalable and versatile weight space learning*.
>
> [2]  Zhou et al. *Permutation equivariant neural functionals*.
>
> [3] Lim et al. *Graph Metanetworks for Processing Diverse Neural Architectures*.

---

> > ### Author Rebuttal · Reviewer_M5tJ · 2026-04-01
> >
> > I thank the authors for their detailed rebuttal and the additional elements provided. Please find my response to the individual elements discussed below.
> >
> > **GP assumption:** This point has been raised by multiple reviewers, and I thank the authors for their clarification. In my opinion, Definition 3.4 and the related definition of GP in the original submission have led to some misunderstandings, since the authors use the non-uniqueness of bias terms as the core definition of the exclusion set. My current understanding is that the GP assumption is defined by the existence of a continuous canonicalisation map which can be approximated by a DWS model. I think that this submission could be made clearer by using such a definition instead, and using the bias-uniqueness approach as an example. This would further help understanding what kind of models follow the GP assumption, as one would simply need to find a fitting canonicalisation map. Finally, I understand that proving that a canonicalisation map does *not* exist is much more difficult, and that finding which models do *not* follow the GP assumption is equivalently difficult.
> >
> > **Increasing output architecture capacity:** This is an interesting, practical result. Thank you for providing it.
> >
> > **Expressivity limitations under quantization:** I find this example challenging to interpret, in relation to the previous paragraph regarding the GP assumption. Is my understanding correct that the quantity measured in this Figure is the GP violation with regard to Definition 3.4, i.e. the proportion of models with non-unique bias parameters? In that case, this experiment mostly measures violations to this specific exclusion set and corresponding canonicalisation map. Does this experiment reveal anything about the expressivity of DWS models for models with quantised biases? For example, if using the $\\mathcal{l}\_p$-norm definition of GP in the response to reviewer r4xm, wouldn’t the GP assumption still hold for approximately 100% of those same models? In such cases, the expressivity of the DWS models would not be altered despite the numerous violations of the exclusion set from Definition 3.4.
> >
> > **Relation to non-equivariant models:** Thank you for this clarification which helps me understand the impact of this paper better. I would have a follow-up question. The GP assumption, from my current understanding, is equivalent to the existence of the canonicalisation map. In such settings, would using this canonicalisation map on the input $K$ before using a symmetry-unaware weight space model render it equivariant? Would it follow that this combination of canonicalisation and symmetry-unaware weight space models would also be equivalent in expressive power with the other equivariant approaches studied in this submission?
> >
> > Since I still have some follow-up questions, in particular those relating to the GP assumption, I am not yet ready to update my score. I look forward to the response from the authors for some more explanations and clarifications, and will reconsider my scoring after reading it.

---

> > > ### Author Response · Authors · 2026-04-02
> > >
> > > We thank the reviewer for their meaningful engagement and insightful comments, which we address below:
> > >
> > > **GP assumption:** We agree that our original formulation may have caused confusion. In the revision, we will restructure this part as follows:
> > > - We will revise Definition 3.4 to explicitly define the unique-bias exclusion set, rather than presenting it as the general definition of GP.
> > > - We will include additional examples of exclusion sets (e.g., the p-norm exclusion set) to illustrate the flexibility of the framework.
> > > - We will clarify that, as noted by the reviewer, our results apply to any exclusion set whose complement admits a continuous canonicalisation map that can be uniformly approximated by equivariant weight space models.
> > >
> > > We believe this revised presentation will help with clarity, while still emphasizing that standard MLP weights are naturally in GP under concrete and simple conditions. Finally, note that Proposition 6.2 implies that there exists no continuous canonicalisation map over the entire weight space which can be implemented by WS models. As the reviewer notes, characterizing optimal exclusion sets (e.g., minimal sets for which such maps exist) is a challenging problem, which we leave for future work.
> > >
> > >
> > > **GP violation experiment:**  We confirm that the reported metric measures GP violations with respect to Definition 3.4. This experiment serves two purposes. First, it shows that in standard high-precision regimes, GP is essentially never violated, suggesting that WS networks do not suffer from expressivity limitations in this setting. Second, it provides a preliminary empirical signal that low-quantization regimes may exhibit such limitations.
> > >
> > > To further support this, we provide additional analysis  [here](https://anonymous.4open.science/r/anonymusicml2026-3A06/extended_gp_rate_vs_xbit_quantization.png) measuring GP violations as a function of precision across multiple exclusion sets. Specifically, for each graph we consider a monotonically increasing family of GP conditions, and mark a weight as “in GP” if it satisfies at least one of them. We begin with Definition 3.4 and progressively incorporate $p$-norm–based GP conditions for $p \le 1,2,3,4,5$.
> > >
> > > We find that, while combining multiple GP conditions reduces the overall violation rate, the same dependence on quantization persists, and the improvement plateaus as more conditions are added. This suggests that, in low-precision regimes, it may be impossible to identify an exclusion set avoided by all weights.
> > >
> > >
> > > While this analysis does not constitute definitive evidence of real-world expressivity limitations in low-precision regimes, we believe that it, together with Proposition 6.2, provides consistent preliminary evidence in this direction. A more thorough investigation is left for future work.
> > >
> > >
> > > **Canonicalization networks:** We thank the reviewer for the opportunity to clarify this point. Applying a canonicalization map followed by a symmetry-unaware model is a classical approach for learning over symmetric data. This pipeline yields an invariant model, and since the underlying symmetry-unaware models are typically universal, universality of the overall construction follows.
> > >
> > > However, while such approaches improve over fully symmetry-unaware models, prior empirical work (e.g., table 1 in [1] and figure 5 in [2]) shows they still generalize worse than symmetry-aware architectures that incorporate parameter sharing or message passing. While a rigorous explanation of this remains an open problem, a common hypothesis is that canonicalization can distort the geometry of the space (e.g., mapping nearby points to distant ones), thereby discarding useful structure.
> > >
> > >
> > > This aligns with the fact that WS models leveraging symmetry-aware mechanisms (e.g., parameter sharing in DWS/NFN and message passing in GMN/NG-GNN) achieve SOTA  across many common WS tasks, making the study of their expressive power essential. Finally, although our universality proofs rely on constructing a canonicalization map, these constructions are not necessarily optimal, and in practice, simpler or more learnable implementations may suffice for many WS tasks.
> > >
> > >
> > > [1] Navon et al. Equivariant Architectures for Learning in Deep Weight Spaces, ICML 2023.
> > >
> > >
> > > [2] Qi et al.  PointNet: Deep Learning on Point Sets for 3D Classification and Segmentation, CVPR 2017.

---

### Official Review · Reviewer_r4xm · 2026-03-06

**Soundness:** 3
**Presentation:** 2
**Significance:** 3
**Originality:** 3
**Overall Recommendation:** 5
**Confidence:** 3

**Summary:**

The paper covers universal approximation results for permutation equivariant neural networks operating on weight spaces. First, a range of weight space networks are proven to have equivalent expressive powers. Second, four settings for weight space networks are discussed in terms of universal approximability: functionals on the function space implemented by the weights, permutation invariant functions of the weights, function to function operators mapping from the function space implemented by the weights and permutation equivariant maps from weights to weights. In these four settings, weight space networks are shown to be universal approximators.

**Compliance With Llm Reviewing Policy:**

Affirmed.

**Final Justification:**

The paper provides several new results on the universal approximation properties of equivariant weight space networks. The presentation is well laid out and I believe that the results will be of interest to the community.

**Key Questions For Authors:**

In the conclusion, there is a claim that the paper “provides principled guidance” for the design of weight space networks. It would be helpful to be more explicit about what this guidance is.

**Limitations:**

Yes

**Strengths And Weaknesses:**

Strengths:
  - The paper provides a large set of new results that extend the knowledge about universal approximation properties of weight space networks from what has been previously covered.
  - While the proofs are relegated to a 50 page appendix, the main paper does provide plausible proof sketches and arguments for building intuition.

Weaknesses:
  - The proof sketches do not always correspond to the actual proofs. The remark on line 2428 explicitly explains this discrepancy, but this should be mentioned in the main paper, not deep in the appendix.
  - The “general position” property in Def 3.4, required for some results, excludes cases where any values in a bias vector are equal. This is claimed to be natural as such bias vectors have measure zero. However, it should be noted that in practice such “degenerate” weights are often reached, e.g. in duplicated or dead neuron configurations.
  - It would be interesting to know if one of the equivalent weight space architectures discussed in the first part of the paper should be preferred over others in practice, but this is not answered by the presented theory.

---

> ### Author Rebuttal · Authors · 2026-03-30
>
> We thank the reviewer for their positive assessment, highlighting both the breadth of new results in our paper, and the clarity of the proof sketches in conveying intuition. We address their comments and questions below.
>
> **On the gap between proof sketches and full proofs:**
> We appreciate this comment and agree that the distinction should be made clearer in the main text, and we will revise the paper accordingly.
>
> **On the general position (GP) property:**
> We agree that understanding our results in the presence of degenerate configurations (e.g., duplicated or dead neurons) is practically important. While we argue that, under standard initialization and training, MLP weights lie outside the exclusion set with high probability, we also explicitly identify regimes where this assumption may fail and degeneracies may naturally arise, most notably in quantized networks (see lines 299–307). In these regimes, we prove that current approaches suffer from expressivity limitations, suggesting the need for more expressive methods. Below, we further support this both empirically and theoretically, and will incorporate these results in the revision.
>
>
> - **Empirical validation.**
> To validate the above, we measure the fraction of models that violate GP as a function of quantization level in the standard MNIST INR dataset [3], where each MLP is trained to represent a single MNIST image (taking coordinates as input and outputting pixel values). The results ([seen here](https://anonymous.4open.science/r/anonymusicml2026-3A06/gp_rate_vs_xbit_bias_quantization.png)) **indicate no violations at high precision**, while degeneracies begin to appear at 12-bit and become significantly more pronounced at 8-bit.
>
>
> - **Alternative definitions of the exclusion set.**
> In our paper, GP is defined as weights with unique bias terms in each layer, however, as noted in Footnote 2 (page 4), our theory extends to a broad class of alternative definitions that do not rely on biases.  For example, for any $p>0$ one may define general position to holds if for each weight matrix all row vectors have distinct $\ell_p$-norms.
>
>
>   This reduces the class of “degenerate” configurations, enabling our theory to cover cases with, e.g., duplicate or dead bias terms. More broadly, for a given weight distribution with continuous PDF, it is highly likely that at least one such definition ensures weights are almost surely in general position, further supporting the robustness of our assumption.
>
>
> **On comparison between different weight-space models:**
> We agree that identifying when one weight-space architecture should be preferred over the others in practice is an important direction. At the same time, we believe our result, showing that existing WS architectures are expressively equivalent, is itself valuable: it enables a unified analysis and suggests that observed performance differences are driven primarily by other factors such as implicit bias, optimization, and generalization. We leave further exploration of these topics for future work.
>
>
> **On providing principled guidance:**
> We agree that the practical guidelines suggested throughout the paper can be highlighted more clearly and expanded upon, and we will improve this in the revision. Here, we focus on what we view as the most impactful guideline; for further details on the experiment below, as well as a broader discussion of additional practical guidelines, we refer to our response to Reviewer HYpN.
>
>
> As discussed in Sec. 7.1, weight-space networks are limited in approximating function-space operators since their  output weights must match the architecture of the input (Prop. 7.1). Our theory shows that this limitation can be mitigated by increasing architecture size (Thm. 7.2), Motivating WS models that map to larger output architectures than their inputs.
>
>
> To validate this, we test a simple and general modification to any weight space network: use output dimension $k>1$ and interpret the output as an ensemble of $k$ MLPs, increasing the output capacity, with predictions averaged across the ensemble. Evaluating DWS [1] and GMN [2] with $k=1,2,4,8$ on the MNIST INR dilation benchmark [3] ([seen here](https://anonymous.4open.science/r/anonymusicml2026-3A06/experiment_results_mnist_inr_dilation.png)) shows consistent and significant improvements as $k$ increases, with up to a 46% MSE reduction. Notably, **GMN with $k=8$ achieves SOTA performance**, even compared to methods using additional signals (e.g., probes or gradients), while **maintaining comparable compute and parameter counts** (with $k=1$ having the most parameters).
>
> We thank the reviewer again for their insightful comments and positive assessment.
>
>
> [1] Navon et al. Equivariant Architectures for Learning in Deep Weight Spaces, ICML 2023.
>
>
> [2] Lim et al. *Graph Metanetworks for Processing Diverse Neural Architectures*, ICLR 2024.
>
>
> [3] Zhou et al. *Permutation equivariant neural functionals*m NeurIPS 2023.

---

> > ### Author Rebuttal · Reviewer_r4xm · 2026-04-01
> >
> > I thank the authors for the rebuttal, which solidifies my belief that the paper should be accepted. I will therefore maintain my positive score.

---

### Official Review · Reviewer_HYpN · 2026-03-11

**Soundness:** 3
**Presentation:** 3
**Significance:** 3
**Originality:** 3
**Overall Recommendation:** 4
**Confidence:** 3

**Summary:**

In this work, the authors study the expressive power of weight-space networks, which are neural networks that operate on the parameters of other neural networks. There are two main theoretical contributions in this paper. First, the authors show that all prominent permutation-equivariant weight-space networks for MLPs are equally expressive. Second, the authors identify four natural and practical settings for weight-space learning, depending on (1) whether the weight-space network maps functions to vectors or weight parameters to vectors, and (2) whether the weight-space network is permutation-invariant or permutation-equivariant. For each of these settings, the authors show that universal approximation is achievable under mild assumptions.

**Compliance With Llm Reviewing Policy:**

Affirmed.

**Final Justification:**

Overall it's a good paper and I would like to see it accepted. Not a higher score due to potentially limited impact.

**Key Questions For Authors:**

In Contribution 3, it is stated that you "identify regimes in which universality does not hold". Is this related to the requirement for general position? Perhaps I missed something, but what regime is this exactly? If this is about low precision, but low precision does not guarantee the loss of general position (it increases the probability of not having general position).

**Limitations:**

yes

**Strengths And Weaknesses:**

Strengths:
- The presentation is generally good and relatively easy to follow. The paper is dense in texts and maths, but I think the authors tried to make sure that they convey the most important messages and results as clear as it can be within the page limit. I only have a few minor complaints for writing which I will describe at the end.
- The claims appear to be technically sound. I did not read the proofs in detail, but the high-level ideas in proof sketches and the stated results are intuitive.
- Characterizing the expressive power of weight-space networks is theoretically interesting, and can provide guidance into practice.

Weakness:
- The analysis of weight-space networks is restricted to those that operate on the parameters of MLPs. This deviates from the practice where MLP, although a building block for many architectures, is rarely used as a stand-alone architecture.
- A thorough discussion on how the theoretical results in this paper can provide additional practical insights is missing. I would love to see the authors discuss more on the practical implications of their analysis.

Minor:
- Use citet and citep as appropriate. E.g., Line 107, col1; Line 115, col1; Line 120, col1; Line 123, col1; Line 267, col1; should use citet.
- Line 144: The notation of continuity is not clear if the sets X and Y are arbitrary. A set can be literally anything, e.g. a set of cats and dogs. If you mean specifically a set consisting of elements from some Euclidean space, so that the notation of equivalent functions can be more conveniently defined, be explicit about that.
- Line 157/163: I don't quite see the reason why the isomorphic symbol is used here. Since you are defining the notation for parameter spaces, shouldn't := be used here?
- Line 208: Psi maps weight parameters (essentially a collection of matrices and vectors) to vectors. Why is it called a functional?

---

> ### Author Rebuttal · Authors · 2026-03-30
>
> We thank the reviewer for their positive and thoughtful assessment, highlighting the relevance, soundness, and clarity of our work. We address their comments and questions below.
>
>
> **On practical implications of our analysis:**
> While our work is primarily theoretical, it leads to several concrete practical insights, which we state throughout the paper, and will emphasize more clearly in the revision. Following the reviewer’s suggestion, we validate some of these empirically, obtaining strong improvements over existing weight-space methods with only minor architectural modifications and **achieving SOTA performance on established weight space benchmarks**.
>
>
> - **Increasing output architecture for function-space operators.**
> As discussed in Sec. 7.1, weight-space networks are limited in approximating function-space operators since their  output weights must match the architecture of the input (Prop. 7.1). Our theory shows that this limitation can be mitigated by increasing architecture size (Thm. 7.2), Motivating WS models that map to larger output architectures than their inputs (no current WS network satisfies this).
>
>
>   To test this, we propose a simple modification applicable to any WS network: use a final feature dimension of $k>1$, and interpret the output as an ensemble of $k$ MLPs, increasing output capacity. The final prediction is obtained by averaging the outputs of these $k$ MLPs.
>
>
>   We evaluate this on both DWS [1] and GMN [2] with $k=1,2,4,8$ on the MNIST INR dilation benchmark [3]. The results ([seen here](https://anonymous.4open.science/r/anonymusicml2026-3A06/experiment_results_mnist_inr_dilation.png))  show consistent and substantial improvements as $k$ increases, with up to a 46% MSE reduction. Notably, **GMN with $k=8$ achieves SOTA performance**, even compared to methods using additional signals (e.g., gradients or probes). Importantly, **all models have comparable computational cost and parameter counts**, with $k=1$ having the most parameters.
>
>
> - **Expressivity limitations under quantization.**
> In Prop. 6.2, we show that expressivity degrades when inputs fall outside general position (GP), and hypothesize this may occur in quantized networks. This suggests that learning over quantized weight spaces may introduce expressivity limitations, motivating more expressive architectures.
>
>
>   To test this, we use the MNIST INR dataset [3] and measure the fraction of models violating GP as a function of quantization level. The results ([seen here](https://anonymous.4open.science/r/anonymusicml2026-3A06/gp_rate_vs_xbit_bias_quantization.png)) show all networks satisfy GP at high precision, but violations begin to appear at 12-bit and become significantly more pronounced in the 8-bit regime. These findings support our prediction and will be included in the revision.
>
>
> - **Additional practical guidelines.**
> We also derive further practical insights. For example, we show that DWS updates can be simplified without loss of expressive power. In addition, we establish that several existing architectures are expressively equivalent, suggesting that performance differences are primarily driven by other factors such as implicit bias, optimization and  generalization.
>
>
> **On extending our analysis beyond MLPs:**
> We thank the reviewer for this important suggestion. We agree that extending our analysis beyond MLP weight space is valuable. Given the already dense presentation (60+ pages with appendix), we focused on fully characterizing the MLP setting, which is currently the most mature and remains highly relevant since, as the reviewer comments, MLPs serve as building blocks in many architectures.
>
>
> Nevertheless, we believe our analysis extends naturally beyond the MLP setting. To support this, we now add an analogue of Thm. 6.3 **, establishing universality of invariant networks under GP for GMNs operating on Transformer weights**, obtained via the same proof strategy. This result will be included in the next revision. Due to space constraints, we omit the proof here and refer to our response to Reviewer Lro2 for a sketch of the main ideas.
>
>
> **On regimes in which universality does not hold:**
> We identify two regimes: (i) inputs not in GP (e.g., quantized networks), and (ii) function-space operators under fixed architecture constraints, where universality may fail requiring increased capacity (Prop. 7.1, Thm. 7.2).
>
>
> **On minor corrections:**
> We thank the reviewer for their careful reading and will address all suggested comments in the next revision.
>
>
> We thank the reviewer again for their insightful comments and positive assessment. We believe our responses address the main concerns and would appreciate reconsideration of the score.
>
>
> [1] Navon et al. *Equivariant Architectures for Learning in Deep Weight Spaces*, ICML 2023.
>
> [2] Lim et al. *Graph Metanetworks for Processing Diverse Neural Architectures*, ICLR 2024.
>
> [3] Zhou et al. *Permutation equivariant neural functionals*m NeurIPS 2023.

---

> > ### Author Rebuttal · Reviewer_HYpN · 2026-04-02
> >
> > I'd like to thank the authors for their response. I will keep my score.

---

### Official Review · Reviewer_Lro2 · 2026-03-12

**Soundness:** 4
**Presentation:** 4
**Significance:** 3
**Originality:** 3
**Overall Recommendation:** 5
**Confidence:** 4

**Summary:**

The paper provides a unified theoretical analysis of the expressive power of weight-space networks that covers most of the known architectures. It first shows that several prominent formulations are equivalent in expressive capacity. It then studies universality across four settings, clarifying when these models are guaranteed to be sufficiently expressive for the target tasks and when expressivity may be a limiting factor.

**Compliance With Llm Reviewing Policy:**

Affirmed.

**Final Justification:**

The authors' rebuttal successfully clarified the technical relationship between Theorem 6.1 and 1-WL expressivity, as well as the nuances of the fixed architecture assumption. I also appreciate the extension of the theory to Transformer architectures, which successfully demonstrates the broader applicability of the framework. These clarifications confirm the robustness of the theoretical contributions, and I am increasing my score to Accept.

**Key Questions For Authors:**

I would consider raising my overall evaluation if the authors can adequately clarify the following points to justify the correctness of their theoretical results.

1. Regarding Theorem 6.1 (universality as function-space functionals): a key step relies on the observation that if a DWS network cannot distinguish two weights, then those weights must induce the same function. In the setting where DWS networks use GNNs, how does this align with the known 1-WL expressiveness limitations of GNNs? Concretely, it seems to imply that if two MLP computation graphs are indistinguishable by 1-WL (e.g., as suggested by Figure 3), then the corresponding MLPs must implement the same function. Is this implication correct? Or does the result hinge on the assumption of a fixed MLP architecture, in which case the underlying computational graph is fixed and the 1-WL comparison across different graphs is not directly applicable?

2. Relatedly, if the paper indeed focuses on a fixed MLP architecture so that the computational graph is fixed, I am not fully convinced by the proof of Theorem 6.1. By Definition D.11, the NG-GNN uses a fixed graph structure (i.e., even when a weight is zero, the corresponding edge exists; the initial edge feature is simply initialized to zero). This appears to be in tension with the argument sketched in Figure 3, which seems to rely on structural distinctions. Could the authors clarify how these points are reconciled in the proof, and whether the indistinguishability argument depends on learned features rather than graph topology per se?

**Limitations:**

Yes

**Strengths And Weaknesses:**

Strengths:
- The paper is clearly structured and well organized, with effective figures that aid understanding. It offers a thorough review of prior work on weight-space networks, categorizes the tasks considered in the literature, and delivers a comprehensive expressivity analysis for each case.
- The theoretical insights are nontrivial and practically informative. The unifying view presented is a meaningful contribution to the study of weight-space models.

Weaknesses:
I do not find any big weakness of this work. Here are some potential areas of improvement, but I think the paper is good as it is (provided the author could clarify some of the key questions raised in the next part).
  1. The analysis centers on a fixed MLP architecture. It would be valuable to discuss how the results might possibly extend to more complex architectures or varying network sizes. What are the key challenges there, and whether we expect similar, or totally different phenomenon?
  2. The proof sketches in the main text are quite dense which undermines its readability. I'd suggest hiding some of the technical details, and to highlight the key intuition and structure of the proofs.

---

> ### Author Rebuttal · Authors · 2026-03-30
>
> We thank the reviewer for their positive assessment, noting our “theoretical insights are nontrivial and practically informative”, a “meaningful contribution,” and the paper is “clearly structured and well organized.” We address their comments below.
>
> **On the connection between Theorem 6.1 and 1-WL:**
> We thank the reviewer for this insightful point. As suggested by the reviewer, our results imply that if two MLP computation graphs (with node and edge features) are indistinguishable by 1-WL, then they represent the same function.  This follows since graph-based weight-space models such as GMN [1] can approximate a forward pass on the input MLP, yet are MPNN-based and thus limited by 1-WL expressivity. Therefore, if two networks compute different functions, such models can distinguish them, implying they must be 1-WL distinguishable. Note that, any pair of computational graphs corresponding to different architectures is 1-WL distinguishable due to the positional encoding used by GMNs.
>
> **Regarding Figure 3 and fixed architecture:**
> We agree that this point requires clarification and will do so in the next revision. Throughout, we apply 1-WL to attributed graphs with both node and edge features (treating weights as edge features), using the standard feature-aware refinement procedure. Thus, even with fixed connectivity, different weight assignments may yield non-isomorphic computational graphs that can still be 1-WL indistinguishable. Figure 3 illustrates this: we consider binarized MLPs (weights in 0,1), and for visualization, edges with weight 0 are shown as absent, although formally they remain with zero-valued features.
>
> **Regarding the density of the proof sketches:**
> We agree and will improve readability by moving technical details to the appendix and emphasizing key ideas.
>
> **On extending our results to more complex architectures:**
> We thank the reviewer for this important suggestion. We agree that extending beyond fixed MLP architectures is valuable. Given the already dense presentation (60+ pages with appendix), we focused on fully characterizing the fixed-architecture MLP setting. Nevertheless, we believe our framework extends more broadly and will discuss this in the revision. To demonstrate this, we provide an analogue of Theorem 6.3 for networks processing transformers.
>
> **Thm:** Under the representation of Transformer weights as parameter graphs described in [1], the GMN architectures introduced in [1] are universal approximators of graph-invariant functions on any compact set $K$ satisfying the general position assumption (defined below).
>
> **Sketch**: The key idea in Theorem 6.3 is to define, outside a suitable exclusion set $\mathcal{E}$, a continuous canonization map $\mathrm{canon}$ that WS networks can approximate. Universality follows by composing $\mathrm{canon}$ with an MLP head.
>
> A Transformer consists of MLP blocks, attention blocks, layer normalization, and residual connections. GMNs represent weights as edges in a parameter graph, augmented with features encoding component type and depth.
>
> Define $\mathcal{E} = \bigcup_l \mathcal{E}_l$, where each $\mathcal{E}_l$ corresponds to a single MLP/attention block $l$. For MLP blocks, $\mathcal{E}_l$ is as in the paper. For attention blocks, $\mathcal{E}_l$ consists of weight configurations where two rows (in $W_v$ or $W_o$) have identical sums. Thus, $\mathcal{E}$ remains a finite union of lower-dimensional sets, and generic weights lie outside it with high probability.
>
> For $K \cap \mathcal{E} = \emptyset$, define $\mathrm{canon}: K \to V$ as follows. For MLP blocks, neurons (and their associated weights) are ordered by sorting their bias values, as in Theorem 6.3.  For attention blocks, neurons are ordered according to the sum of the corresponding row in $W_v$ or $W_o$, which uniquely determines an ordering under the exclusion assumption. The weight parameters are then ordered lexicographically according to the induced ordering of their source and target neurons. Residual and layer-normalization parameters are ordered analogously.
>
> Finally, GMNs can approximate $\mathrm{canon}$. For MLP blocks, this follows from Theorem 6.3. For attention layers, GMNs can compute row-wise sums of $W_v$ and $W_o$ via message passing, then apply a set-equivariant network to implement sorting and reordering. Thus, the same canonization argument applies, yielding universality for transformer weight spaces under a suitable general position assumption. While we omit full details due to space constraints, the discussion above illustrates that our framework extends beyond MLPs; we will expand on this and provide full and formal proofs in the revision.
>
> We thank the reviewer again for their insightful comments and positive assessment. We believe our responses address the main concerns raised and would appreciate a reconsideration of the score.
>
> [1] Lim et al. *Graph Metanetworks for Processing Diverse Neural Architectures*, ICLR 2024.

---

> > ### Author Rebuttal · Reviewer_Lro2 · 2026-04-02
> >
> > I appreciate the authors for their detailed replies, which have addressed my concerns. I will increase my score accordingly.

---

### Decision · Program_Chairs · 2026-04-30

**Decision:**

Accept (spotlight)

**Comment:**

Reviewers assessed this to be a strong theoretical contribution on an underexplored topic. The authors also derive practical implications of their framework on applied machine learning (transformers), which demonstrates breadth. A clear accept.